# Reconstitution of pluripotency from mouse fibroblast through *Sall4* overexpression

Lizhan Xiao [1,2,8], Zifen Huang[1,2,8], Zixuan Wu[1,2,8], Yongzheng Yang[3], Zhen Zhang[1,2], Manish Kumar [1,2], Haokaifeng Wu [1,2], Huiping Mao[1,2], Lihui Lin[1,2], Runxia Lin[1,2], Jingxian Long[1,2], Lihua Zeng[1,2], Jing Guo[1,2], Rongping Luo[1,2], Yi Li[1,4], Ping Zhu [3,4], Baojian Liao [5] ✉, Luqin Wang [1,2] ✉ & Jing Liu [1,2,6,7] ✉

Somatic cells can be reprogrammed into pluripotent stem cells (iPSCs) by overexpressing defined transcription factors. Specifically, overexpression of OCT4 alone has been demonstrated to reprogram mouse fibroblasts into iPSCs. However, it remains unclear whether any other single factor can induce iPSCs formation. Here, we report that SALL4 alone, under an optimized reprogramming medium iCD4, is capable of reprogramming mouse fibroblasts into iPSCs. Mechanistically, SALL4 facilitates reprogramming by inhibiting somatic genes and activating pluripotent genes, such as *Esrrb* and *Tfap2c*. Furthermore, we demonstrate that co-overexpressing SALL4 and OCT4 synergistically enhances reprogramming efficiency. Specifically, the activation of *Rsk1/Esrrb/Tfap2c* by SALL4, alongside OCT4's activation of *Sox2* and the suppression of *Mndal* by SALL4 and *Sbsn* by OCT4, cooperate to facilitate SALL4+OCT4-mediated reprogramming. Overall, our study not only establishes an efficient method for iPSCs induction using the SALL4 single factor but also provides insights into the synergistic effects of SALL4 and OCT4 in reprogramming.

The development of induced pluripotent stem cell (iPSCs) technology has provided valuable insights into the mechanisms underlying cell fate decisions. iPSCs can be generated using different approaches, with the overexpression of the classical four Yamanaka factors, OCT4, SOX2, KLF4, and c-MYC (OSKM) or their simplified combination, such as OKS, OK, or OM[1–4]. OCT4 has been recognized as a central reprogramming factor that is required for iPSCs induction[3,5,6]. Previous studies have reported on methods utilizing single OCT4-mediated iPSCs generation. In 2009, Kim et al. found that single OCT4 alone can induce the conversion of adult neural stem cells into iPSCs[7,8]. With the help of chemical cocktails such as AMI-5 and A8301, mouse fibroblasts can be reprogrammed into iPSCs within 30–40 days after OCT4 overexpression[9]. In addition, under the iCD1 + BMP4 culture condition, mouse fibroblasts can be reprogrammed into iPSCs with ~0.05%

[1]Center for Development and Regeneration, Guangzhou Institutes of Biomedicine and Health, Chinese Academy of Sciences, Guangzhou, China. [2]Guangdong Provincial Key Laboratory of Stem Cell and Regenerative Medicine, Guangdong-Hong Kong Joint Laboratory for Stem Cell and Regenerative Medicine, Guangzhou Institutes of Biomedicine and Health, Chinese Academy of Sciences, Guangzhou, China. [3]Guangdong Cardiovascular Institute, Guangdong Provincial People's Hospital (Guangdong Academy of Medical Sciences), Southern Medical University, Guangzhou, Guangdong, China. [4]Guangdong Provincial Key Laboratory of Pathogenesis, Targeted Prevention and Treatment of Heart Disease, Guangzhou Key Laboratory of Cardiac Pathogenesis and Prevention, Guangzhou, Guangdong, China. [5]School of Basic Medical Sciences, Key Laboratory of Biological Targeting Diagnosis, Therapy and Rehabilitation of Guangdong Higher Education Institutes, the Fifth Affiliated Hospital of Guangzhou Medical University, Guangzhou, China. [6]Joint School of Life Sciences, Guangzhou Institutes of Biomedicine and Health, Chinese Academy of Sciences, Guangzhou Medical University, Guangzhou, China. [7]Centre for Regenerative Medicine and Health, Hong Kong Institute of Science & Innovation, Chinese Academy of Sciences, Hong Kong SAR, PR China. [8]These authors contributed equally: Lizhan Xiao, Zifen Huang, Zixuan Wu. ✉e-mail: liaobaojian@gzhmu.edu.cn; wang_luqin@gibh.ac.cn; liu_jing@gibh.ac.cn

efficiency after 24 days of OCT4 overexpression[10]. These approaches have shown limitations in terms of efficiency and duration. It has been observed that OCT4 aberrantly activates genes unrelated to pluripotency and negatively impacts the expression of imprinted genes, which could be a potential underlying cause for the low-efficiency observed[11]. As a pioneer transcription factor, OCT4 can interact with other proteins to bind to unoccupied chromatin sites, modify chromatin status, and initiate pluripotent-related gene expression[12–16]. The crucial roles of OCT4 in both pluripotency maintenance[17] and reprogramming highlight the significance of this gene. Furthermore, it is still unknow whether there exist other factors, apart from OCT4 that can individually mediate somatic cell reprogramming.

Recently, a set of alternative reprogramming factors cocktails, encompassing *Nanog, Esrrb, Glis1, Jdp2, Kdm2b, Sall4,* and *Mkk6*, has been identified for their demonstrated capacity to efficiently and effectively reprogram mouse embryonic fibroblasts (MEFs) into high-quality iPSCs[18]. Remarkably, dropout experiments show that SALL4 stands out as the most pivotal factor within the reprogramming cocktails. Moreover, there is consistent evidence demonstrating that SALL4 significantly boosts the efficiency of reprogramming in the context of OKS (OCT4, KLF4, and SOX2)[19,20]. The overexpression of *Sall4, Nanog, Esrrb,* and *Lin28* in MEFs could also be sufficient to produce iPSCs[21]. However, it is not known whether SALL4 alone can induce iPSCs generation. Maternal SALL4 is detectable as early as the embryonic two-cell stage, and its expression commences during the early cleavage period following zygotic genome activation[22–24]. Studies have also reported the requirement of SALL4 for the proliferation, self-renewal, and pluripotency of embryonic stem cells (ESCs)[25,26]. Given the critical role of SALL4 in somatic cell reprogramming and embryonic development, we hypothesize that SALL4 alone is capable of reprogramming somatic cells into a pluripotent state, similar to OCT4.

In this study, we successfully established single-factor-mediated somatic reprogramming systems through the overexpression of SALL4 or OCT4, respectively. Furthermore, we discovered that SALL4 and OCT4 can synergistically work together to significantly enhance reprogramming efficiency. We also investigated the individual roles of SALL4 and OCT4 in promoting reprogramming and explored the molecular mechanisms underlying their synergistic interaction, which markedly facilitates the reprogramming process.

## Results

### Establishment of SALL4-induced reprogramming system

Previously, we developed a medium known as iCD1, which demonstrated remarkable efficiency in supporting iPSCs reprogramming[27]. Notably, the addition of BMP4 to iCD1 further supported the OCT4-induced reprogramming[10]. Building upon these findings and considering the potent role of SALL4 in reprogramming, our hypothesis was that SALL4 alone could reprogram somatic cells into iPSCs when cultivated in a suitable medium. To test this hypothesis, we conducted a compound screening based on iCD1 medium (iCDx) and identified eight molecules that exhibited the capability to drive the reprogramming of MEFs into iPSCs by overexpressing SALL4 through retrovirus infection (Supplementary Fig. 1a–c). Among those compounds, RepSox, an inhibitor of TGF-βR/ALK5, exhibited the most significant effect at 5 μM in concentration (Supplementary Fig.1d). After further optimization, we finally developed a medium, iCD4, which demonstrated effective support for SALL4-mediated iPSCs generation (Fig. 1a, b). During the process of SALL4-induced reprogramming, we observed significant epithelialization on day 4, followed by the appearance of OCT4-GFP⁺ cells on day 7. By day 10, typical iPSCs colonies were formed at a frequency of approximately 20 colonies per 30,000 cells (Fig. 1b, c). Subsequently, we selected these colonies and maintained them in the KSR-2iLIF medium, where the derived iPSCs exhibited stable passaging and maintained a normal karyotype

(Fig. 1d, e). The SALL4-iPSCs demonstrate comparable patterns of pluripotent gene expression to ESCs at both RNA and protein levels (Fig. 1f, g). In addition, transcriptome profiling analysis (Fig. 1h and Supplementary Fig. 1e) confirmed the resemblance of SALL4-iPSCs to ESCs. Subsequent experiments involving teratoma formation (Fig. 1i) and chimeric mouse generation with germline transmission capability (Fig. 1j) further validated the pluripotent nature of SALL4-iPSCs. Furthermore, we obtained OCT4-GFP⁺ cells using mouse tail tip fibroblasts (TTFs) as starting cells (these cells failed to develop into stable iPSCs lines) (Supplementary Fig. 1f). These findings collectively demonstrate that SALL4 alone has the capability to induce the generation of iPSCs under iCD4 conditions.

To investigate the contribution of the main components of iCD4 in SALL4-induced iPSCs generation, we performed dropout experiments and measured the effect of indicated components. The result revealed that all the components were required for successful SALL4-iPSCs induction. Significantly, within the iCD4 medium, the components Vc, Chir99021, SGC0946, RepSox, and the cytokine bFGF (the absence of bFGF leads a low cytoactivity for MEFs) emerge as particularly crucial (Fig. 1k and Supplementary Fig. 1g). Moreover, in our investigation to validate the impact of *Sall4*-related reprogramming-enhancing compounds in the OKS-reprogramming process, we conducted OKS-induced reprogramming using iCD4-remove-RepSox medium supplemented with eight molecules respectively. Results revealed that while RepSox slightly inhibits OKS-reprogramming, the other compounds showed no significant effects (Supplementary Fig. 1h, i). Notably, an inhibitory effect on reprogramming was observed using OKS + SALL4 under iCD4 conditions (Supplementary Fig. 1j, k). This suggests the diverse roles of these compounds and genes in various reprogramming methodologies.

SALL4 possesses both the DNA-binding domain and NuRD recruitment domain, which may be critical for its functions. To assess the significance of SALL4's DNA-binding ability in reprogramming, we created three distinct mutants of SALL4, namely ΔZFC1 (deletion of zinc finger domains cluster 1), ΔZFC2 (deletion of ZFC2), and ΔZFC3 (deletion of ZFC3) (Supplementary Fig. 2a, b). Functional experiments conducted with these mutants revealed their inability to generate iPSCs colonies (Supplementary Fig. 2c). These findings suggest that SALL4's DNA-binding ability may play a crucial role in mediating the reprogramming process. In addition, earlier studies have indicated that SALL4 recruits a transcriptional repressor, the NuRD complex, to facilitate JGES (*Jdp2, Glis1, Esrrb,* and *Sall4*)-mediated reprogramming by targeting specific somatic loci[28]. However, whether this function is also pertinent in SALL4-driven reprogramming alone remains unknown. To explore this role within the process, we disrupted the NuRD recruitment function by deleting the N-terminal NuRD recruitment domain(ΔN12) of SALL4 (Supplementary Fig. 2a, b). The IP-MS experiment confirmed the defect in the NuRD recruitment ability of the SALL4-ΔN12 mutant (Supplementary Fig. 2d–f). In the reprogramming experiment, there was an acceleration in the emergence of OCT4-GFP-positive cells during SALL4-ΔN12-driven reprogramming (Supplementary Fig. 2c). However, our further experiments revealed defects in the ability to generate a stable iPSCs cell line with these OCT4-GFP-positive cells, as most of the picked GFP-positive cells failed to grow and passage (Supplementary Fig. 2g–j). These results suggest that the NuRD recruitment function of SALL4 may be important for iPSCs formation.

Our subsequent aim was to identify the reprogramming intermediates during SALL4-driven reprogramming. For this, we conducted a time-course FACS analysis utilizing previously reported cell surface markers (*Thy1* and *Epcam*) associated with OKSM-reprogramming intermediates[29]. The results unveiled a gradual rise in a cluster of

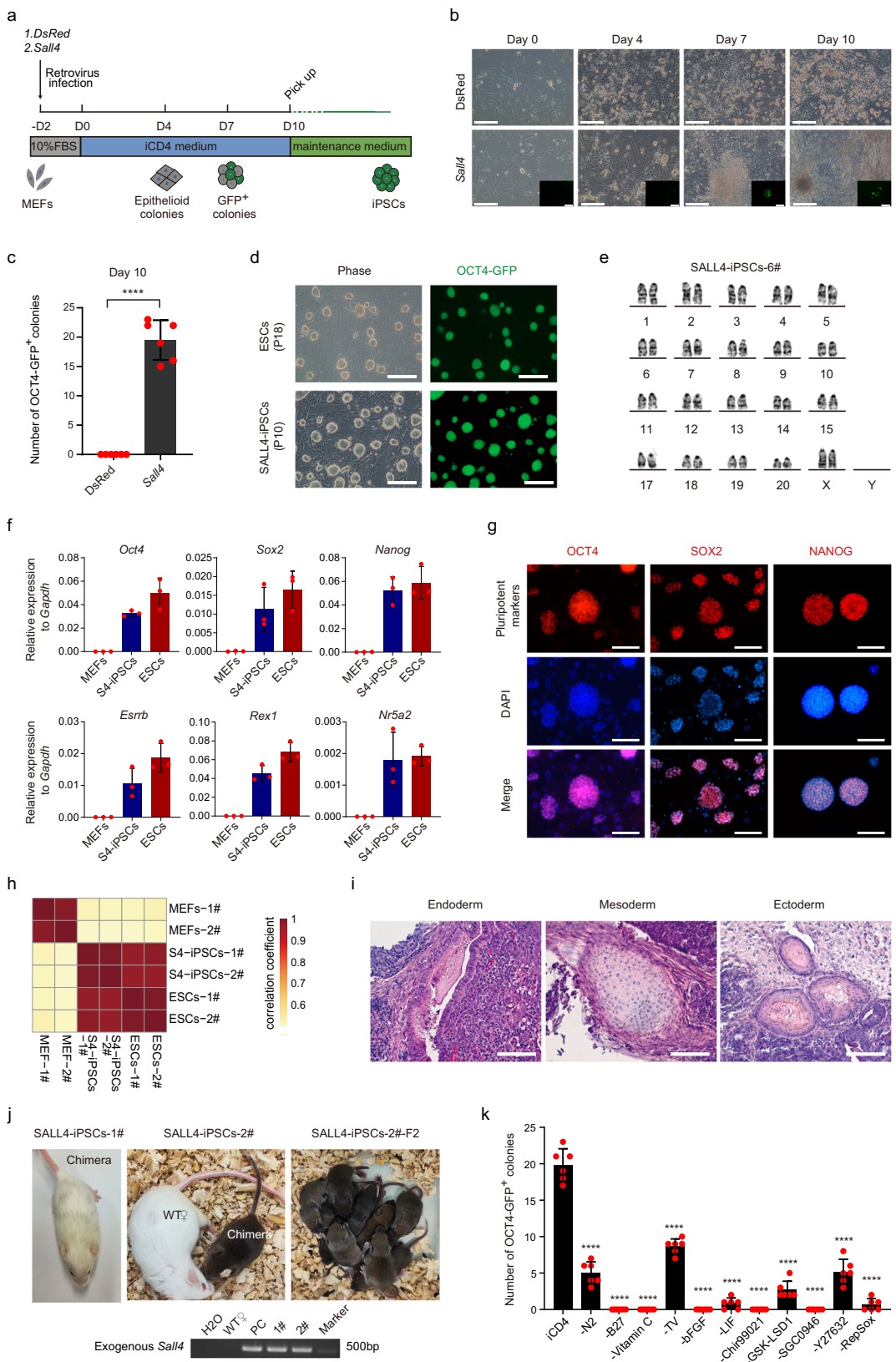

THY1⁻/EPCAM⁺ cells during reprogramming, with nearly all OCT4-GFP⁺ cells being EPCAM positive (Supplementary Fig. 3a, b). Subsequently, we isolated these cells at day 7 and induced them using iCD4 medium while keeping unsorted cells as the control group. After a 4-day induction, we observed OCT4-GFP positive cells in both the control group and the THY1⁻/EPCAM⁺ cluster. Notably, the THY1⁻/EPCAM⁺ cluster demonstrates a relatively higher efficiency in inducing OCT4-GFP⁺ cells compared to other clusters, although these OCT4-GFP⁺ cells exhibit limited proliferative capacity in KSR-2iLIF medium (Supplementary Fig. 3c, d). This suggests that the THY1⁻/EPCAM⁺ cluster represents the reprogramming intermediates in SALL4-driven reprogramming.

**Fig. 1 | Generation of iPSCs from MEFs induced by SALL4 and small molecules.**
**a** Schematic of iPSCs induction from mouse fibroblasts using exogenous genes and small molecules. **b** Morphological diagram for the SALL4-iPSCs induction process. Scale bars, 200 μm. The experiments were repeated independently three times with similar results. **c** Number of OCT4-GFP⁺ colonies from $3 \times 10^4$ MEFs infected with *Sall4* or DsRed in iCD4 on Day 10. MEFs infected with DsRed as control are shown. Data are mean ± SD. Statistical analysis was performed using a two-tailed, unpaired *t* test; *n* = 6 well from 3 independent experiments. ****$p$ = 0.0000325. **d** The morphology of Passage10 iPSCs colonies derived from MEFs by over-expressed *Sall4* in iCD4. ESCs as control are shown. Scale bars, 200 μm. The experiments were repeated independently three times with similar results. **e** The normal karyotype for SALL4-iPSCs. **f** qRT-PCR analysis of pluripotency markers in SALL4-iPSCs. Data are mean ± SD; *n* = 3 biological replicates. S4, SALL4. **g** Immunofluorescence analysis of pluripotency markers in SALL4-iPSCs. Scale bars, 200 μm. The experiments were repeated independently three times with similar results. **h** Correlation analysis for RNA-seq from SALL4-iPSCs, MEFs, and ESCs. *n* = 2

biological replicates. S4, SALL4. **i** The three germ layers of a teratoma from SALL4-iPSCs. Entoderm, Glandular duct tissue. Mesoderm, chondrocyte. Ectoderm, skin tissue. Scale bars, 100 μm. The experiments were repeated independently three times with similar results. **j** Chimera mice with Germline transmission from SALL4-iPSCs (top) and genotype identification for exogenous *Sall4* in chimeras by PCR (bottom). PCR are using the pMXs-*Sall4* plasmid as positive control (PC). The experiments were repeated independently three times with similar results. **k** Drop-out of individual components during SALL4-induced iPSCs reprogramming. Data are mean ± SD. Statistical analysis was performed using a two-tailed, unpaired *t* test; *n* = 6 well from 3 independent experiments. iCD4 versus -N2, ****$p$ = 0.0000003; iCD4 versus -B27, ****$p$ = 0.0000038; iCD4 versus -Vitamin C, ****$p$ = 0.0000038; iCD4 versus -TV, ****$p$ = 0.0000099; iCD4 versus -bFGF, ****$p$ = 0.0000038; iCD4 versus -LIF, ****$p$ = 0.0000009; iCD4 versus -Chir99021, ****$p$ = 0.0000038; iCD4 versus -GSK-LSD1, ****$p$ = 0.0000003; iCD4 versus -SGC0946, ****$p$ = 0.0000038; versus -Y27632, ****$p$ = 0.0000003; iCD4 versus -RepSox, ****$p$ = 0.0000006. TV, Thiamine HCl + Vitamin B12.

## The transcriptome dynamics for SALL4-induced reprogramming

To further investigate the molecular mechanism underlying SALL4-induced reprogramming, we conducted RNA-seq analysis at four time points (Day0, Day4, Day7, Day10) during the reprogramming process mediated by SALL4 (referred to as the SALL4 system) or DsRed (referred to as DsRed system). We included RNA-seq data from ESCs, MEFs, and SALL4-iPSCs as controls (Supplementary Fig. 4a). The PCA plot shows that the reprogramming path in the DsRed system diverged from ESCs, whereas the SALL4 system gradually approached ESCs (Supplementary Fig. 4b). To identify the genes regulated by SALL4 during iPSCs induction, we analyzed differentially expressed gene in the SALL4 system. Using the DsRed system as a reference, we categorized gene changes into two major groups: genes specifically upregulated by SALL4 (C1-C3) and genes specifically downregulated by SALL4 (C4-C6) (Supplementary Fig. 4c). The gradual convergence of gene expression levels in the SALL4 system towards those of ESCs suggests that these genes may play a role in promoting SALL4-mediated reprogramming (Supplementary Fig. 4c). We then conducted GO analysis for the C1 and C6 subgroups to gain insights into the biological processes involving these genes. The C1 subgroup, notably upregulated by SALL4 in comparison to the DsRed system, is linked with biological processes crucial in reprogramming. These processes include epithelial cell morphogenesis, maintenance of stem cell populations, and specification of embryonic patterns (Supplementary Fig. 4d). In contrast, the C6 subgroup is associated with processes related to organ differentiation, including the inflammatory response, nervous system development, lung development, and heart development (Supplementary Fig. 4e). The proper regulation of these biological processes is likely crucial for the transformation of pluripotency during SALL4-induced iPSCs generation. We further conducted GO analysis on subgroups C2, C3, C4, and C5, revealing their enrichment in processes related to the cell cycle and immune system development (Supplementary Fig. 4f, g).

## The chromatin binding dynamics of SALL4 during SALL4-mediated reprogramming

SALL4 functions as a nuclear transcription factor that interacts with enhancers and promoters, to regulate transcriptional changes during early embryonic development[13,30–32]. Our mutant experiments demonstrated the importance of SALL4's zinc finger domain in inducing iPSCs reprogramming (Supplementary Fig. 2c). This suggests that the DNA-binding ability of SALL4 may have effects on reprogramming. Therefore, we aimed to investigate how SALL4 regulates reprogramming by binding to specific genomic loci. To obtain DNA binding data of exogenous SALL4, we performed Cut&Tag using the Flag-tagged SALL4 or SALL4-mutants overexpressed cells (overexpressed by retroviral infection) during the iPSCs induction process, respectively (Fig. 2a and Supplementary Fig. 5a, b). We initially analyzed the genomic

distribution of SALL4 binding peaks and observed that only 25% of these peaks were located in promoter regions, while the majority were found in distal intergenic regions and introns (Fig. 2b). This suggests that SALL4 may regulate gene expression not only by binding to gene promoter regions but also by binding to enhancers or silencers in distal intergenic regions and introns. Subsequently, we conducted a Gene Ontology (GO) analysis for these peaks. The outcomes revealed that the genes bound by SALL4 encompass reprogramming-related biological processes, including chromatin remodeling, epithelial cell proliferation, and the maintenance of stem cell populations (Fig. 2c). In addition, we performed a comparison of the binding peaks between SALL4-WT and the mutants, defining the alterations in binding peaks caused by the SALL4 mutants (Supplementary Fig. 5b, c). To identify the genes regulated by SALL4, we compared the genes annotated by SALL4 binding peaks with the genes specifically up / down-regulated in C1 and C6 subgroups. The Venn diagram revealed that 1485 genes were both occupied by SALL4 and exhibited changes in transcription levels, with 507 genes upregulated and 978 genes downregulated (Fig. 2d). GO analysis of these genes showed that upregulated genes were involved in functions such as stem cell population maintenance and epithelial cell development (Fig. 2d). Downregulated genes, on the other hand, were associated with angiogenesis and synapse organization(Fig. 2d).

To further characterize the binding site of SALL4, we performed motif enrichment analysis, revealing that the top seven enriched motifs shared a common TGACTCA sequence (Fig. 2e). These putative SALL4 binding sites were similar to those recognized by transcription factors such as FOS, BATF, FRA1, AP-1, JUNB, ATF3, and FRA2, suggesting potential shared downstream target genes between these factors and SALL4. De novo motif analysis also showed enrichment of the TGACTCA sequence in this process (Supplementary Fig. 5d). To understand the role of these motif-related transcription factors (TFs), we initially examined the RNA-seq data during SALL4-driven iPSCs induction. Our findings revealed that *Batf, Fos*, and *Atf3* showed a slight upregulation during this process, although their overall expression levels remained relatively low. In contrast, *Junb, Jun, Fosl1* (*Fra1*), and *Fosl2* (*Fra2*) exhibited high expression levels and were subsequently downregulated by SALL4 during the early stages (D0-D7) of iPSCs induction (Supplementary Fig. 5e). Based on these findings, we hypothesized that the inhibition of these transcription factors (TFs) by SALL4 might promote reprogramming. To test this hypothesis, we conducted overexpression experiments by retroviral infection to counteract the downregulated expression induced by SALL4. The results demonstrated that the overexpression of *Junb, Jun, Fosl1/2, Atf3*, and *Fos* suppressed iPSCs generation, aligning with our expectations (Supplementary Fig. 5f). Interestingly, the overexpression of *Batf* alongside *Sall4* improved reprogramming (Supplementary Fig. 5f, g). These results suggest a potential interaction between SALL4 and BATF in facilitating reprogramming.

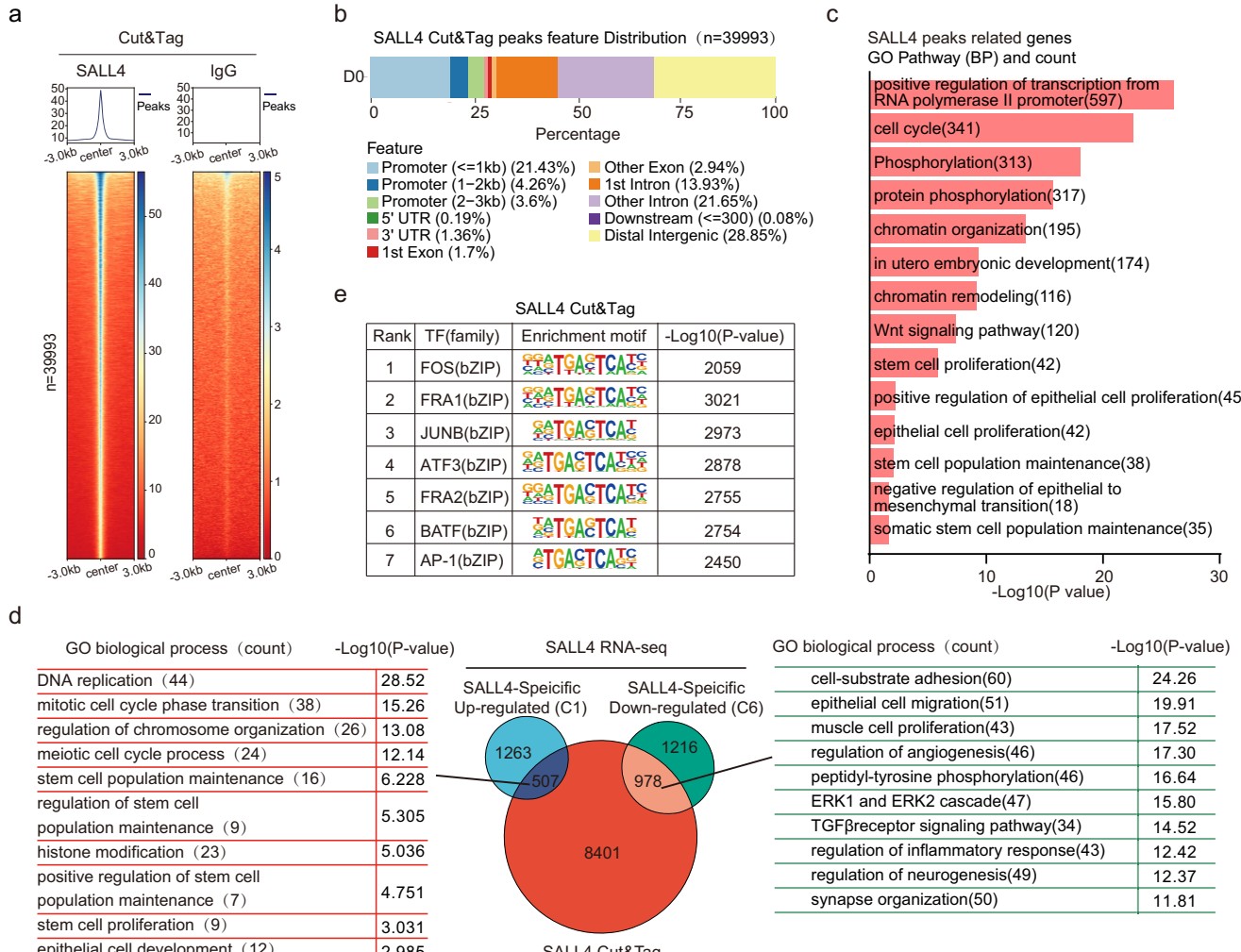

**Fig. 2 | SALL4 binding the genomic locus to activating and silencing reprogramming-related genes. a** Heatmap of Cut&Tag data at D0 from IgG and SALL4, respectively, showing all binding peaks centered on the peak region within a 3 kb window around the peak. **b** Genome distribution of the location for SALL4-occupied peaks relative to the nearest annotated gene. **c** GO biological processes analysis for genes near the SALL4-TSS loci binding peaks. Statistical analysis was performed using Fisher's exact test, the -Log10(p-value) for each term is shown. **d** Left, representative GO biological processes analyzed by the up-regulated overlapping genes. Right, representative GO biological processes are analyzed by the down-regulated overlapping genes. Middle, overlapping genes compared by CUT&Tag and RNA-seq data. Statistical analysis for GO was performed using Fisher's exact test, the -Log10(p-value) for each term is shown. **e** Transcription factor motif enrichment of SALL4-binding peaks. Statistical analysis was performed using Karlin/Altschul statistics, and the -Log10(p-value) for the motif is shown.

Despite the RNA-seq data indicating the up-regulation of *Batf*, we were unable to detect this protein during SALL4-driven reprogramming using western blot analysis (Supplementary Fig. 5h). Based on these findings, we hypothesize that SALL4 binds to the BATF-related loci and regulates these genes to influence reprogramming efficiency. To explore this, we introduced the BATF-DNA binding region into WT SALL4 (SALL4-BATF-B) and created a variant with zinc finger domain deletions (SALL4 mut1/2/3-BATF-B) to enhance the binding affinity to the BATF motif region (Supplementary Fig. 5i). Reprogramming experiments revealed improved induction efficiency using the SALL4-BATF-B fusion protein. Furthermore, the deficiencies resulting from the deletion of SALL4 ZFC1 or ZFC2 were restored by the addition of the BATF-DNA binding region (Supplementary Fig. 5j, k). These results suggest that SALL4 may bind to BATF motif-related genes to promote reprogramming.

**SALL4 binds and regulates chromatin accessibility dynamics through direct and indirect effects to promote iPSCs induction**
Chromatin remodeling is an essential event during reprogramming. To explore the chromatin accessibility dynamics (CADs) during SALL4-driven reprogramming, we collected ATAC-seq data from the aforementioned four-time points of DsRed and SALL4 systems (Fig. 3a and Supplementary Fig. 4a). We conducted a comparison of peaks at each locus between MEFs and ESCs, categorizing the peaks into three main groups: closed in MEFs but open in ESCs (CO), open in MEFs but closed in ESCs (OC), and open in both MEFs and ESCs (PO). Following this classification, the CO and OC peaks were further segmented into distinct subgroups (OC1-OC5 and CO1-CO5) based on the timing of transition, effectively illustrating the progression of dynamics in chromatin opening and closing (Fig. 3a). We found that the number of peaks in OC1, OC3, OC4, OC5, and CO5 subgroups were different between the SALL4 and DsRed systems (Fig. 3b). The higher number of OC1-4 and CO1-4 peaks and the lower number of OC5 and CO5 peaks in the SALL4 system compared to the DsRed system suggests that the addition of SALL4 increase the transition numbers of ESCs-CADs-related-peaks during reprogramming (Fig. 3b and Supplementary Fig. 6a). Moreover, the Venn diagram analysis of CO1-4, OC1-4, and PO peaks between the DsRed and SALL4 systems revealed 39,959 specific OC peaks and 5028 specific CO peaks induced by SALL4 (Fig. 3c). We conducted statistical analysis on the distribution of peaks in genomic

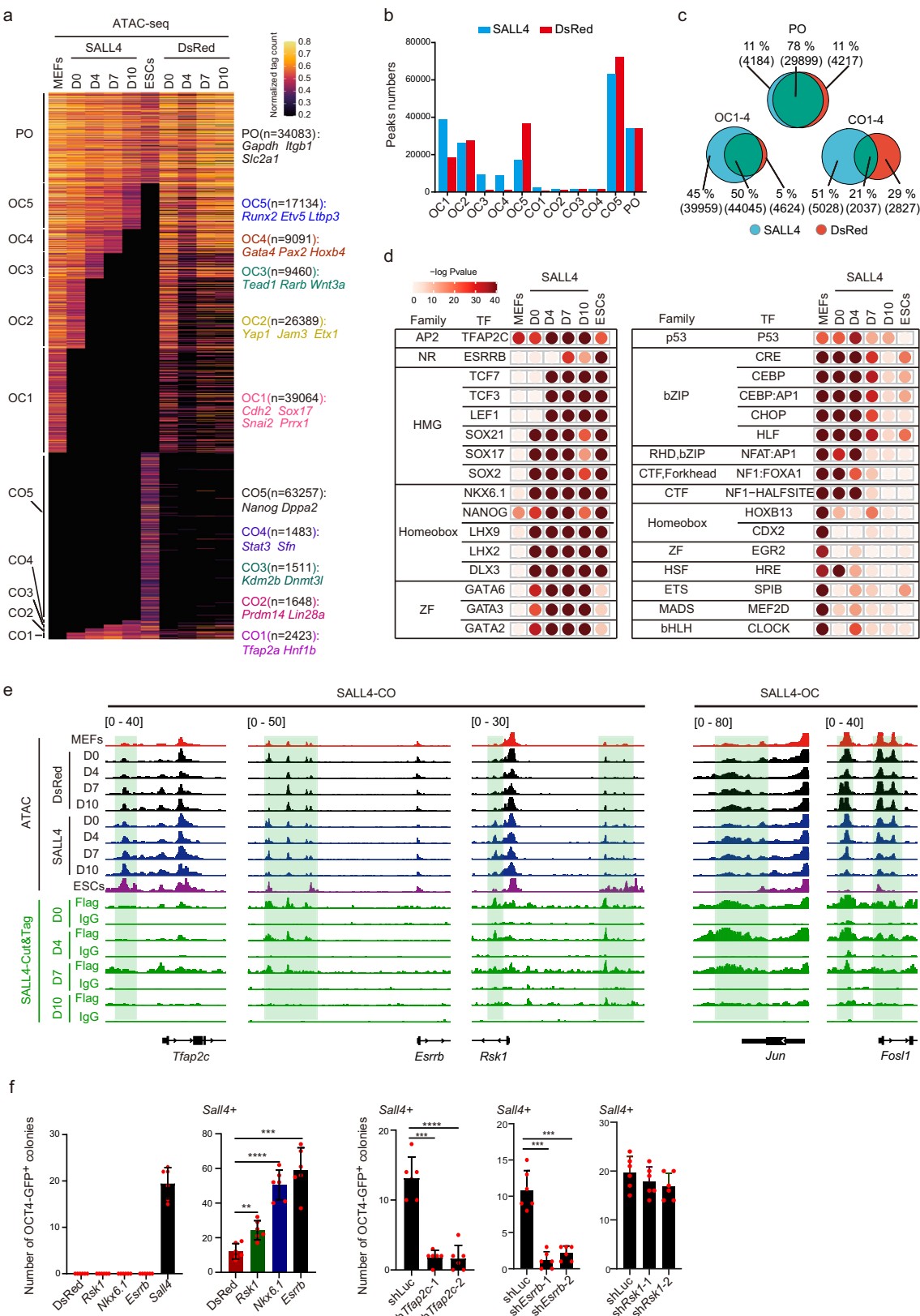

loci and carried out a Gene Ontology (GO) analysis for each CO and OC subgroup. Remarkably, the OC subgroups displayed enrichment in somatic-related processes, such as synapse organization (Supplementary Fig. 6b–d). This finding suggests that SALL4 primarily regulates the dynamics of open-to-close chromatin accessibility to promote reprogramming. To comprehend the regulatory landscape

within the SALL4 system, we conducted motif enrichment analysis for the peaks influenced by SALL4. Notably, the peaks observed in the DsRed system were excluded from the SALL4 system. The results showed a significant enrichment of key reprogramming factors in the SALL4 system, including ESRRB, TFAP2C, SOX2, and NKX6.1. Conversely, factors associated with somatic cell characteristics, such as

**Fig. 3 | Chromatin accessibility dynamics during SALL4-iPSCs induction. a** CADs for SALL4 system and DsRed system. PO, permanently open. CO, close to open. OC, open to close. Take the DsRed system as a reference is shown. **b** The histogram shows the number of peaks for CO, OC, and PO subgroups of the SALL4 system and DsRed system. SALL4, SALL4 system. DsRed, DsRed system. **c** Venn diagrams show the overlapping number and factor-specific number of CO1-4, OC1-4, and PO peaks between the SALL4 system and the DsRed system. SALL4, SALL4 system. DsRed, DsRed system. **d** Motif analysis of the SALL4-specific enrichment peaks at D0, D4, D7, and D10 during iPSCs induction, the peaks for each day in the DsRed system as a background was removed from the peaks at D0, D4, D7, and D10 in SALL4 system. Statistical analysis was performed using Karlin/Altschul statistics, and the -Log10(*p*-value) for the motif is shown. SALL4, SALL4 system. **e** Representative Genomic loci and genes for SALL4-specific close-to-open (SALL4-CO) peaks, SALL4-specific open-to-close (SALL4-OC) and SALL4-binding peaks (Cut&Tag data). **f** The iPSCs induction efficiency using SALL4 overepressing with representative SALL4-occupied gene *Esrrb, Rsk1*, and ATAC-seq motif-enriched gene *Nkx6.1*. The reprogramming efficiency induced by *Sall4* when knocking down *Tfap2c* or *Esrrb*(right). Data are mean ± SD. Statistical analysis was performed using a two-tailed, unpaired *t* test; *n* = 6 well from 3 independent experiments. *Sall4* + DsRed versus *Sall4* + *Nkx6.1*, ****p = 0.0000162*; *Sall4* + DsRed versus *Sall4* + *Esrrb*, ***p = 0.0001515*; *Sall4* + DsRed versus *Sall4* + *Rsk1*, **p = 0.0019697*; *Sall4* + sh*Tfap2c-1* versus *Sall4* + shLuc, ***p = 0.0001119*; *Sall4* + sh*Tfap2c-2* versus *Sall4* + shLuc, ****p = 0.0000343*; *Sall4* + sh*Esrrb-1* versus *Sall4* + shLuc, ***p = 0.0001059*; *Sall4* + sh*Esrrb-2* versus *Sall4* + shLuc, ***p = 0.0002597*. shLuc, shLuciferase.

P53, HLF, MEF2D, and HRE, were progressively depleted during the reprogramming process (Fig. 3d). These observations suggest that these factors may play a crucial role in the generation of iPSCs induced by SALL4. Taken together, these findings imply that SALL4 orchestrates the transition of Chromatin Accessibility Dynamics (CADs) toward an embryonic stem cell (ESCs) state during the induction of iPSCs.

For a deeper exploration of the relationship between SALL4 binding and chromatin accessibility dynamics, we compared the SALL4 Cut&Tag peaks and ATAC-seq peaks on day 0 of reprogramming. Approximately 35779 of Cut&Tag peaks were found to overlap with ATAC-seq peaks (C2), with some of these regions (belonging to the OC2-4 subgroups) displaying closure during reprogramming (Supplementary Fig. 7a, b). The distribution of ATAC-seq peaks that overlapped without Cut&Tag peaks (C1) was predominantly observed in OC1, OC2, and CO1 (Supplementary Fig. 7c). Furthermore, we conducted a Gene Ontology (GO) analysis to further elucidate the function of these peaks (Supplementary Fig. 7d). In addition, we also examined the direct or indirect effects of SALL4 on the chromatin regions at various time points. Our findings further indicate that the majority of SALL4 binding congregates within the Open-Close (OC) subclusters. In OC2-4, these chromatin regions are initially occupied by SALL4 and eventually close, suggesting a correlation between CADs and SALL4's direct binding. Conversely, the OC1 subcluster demonstrates a low level of SALL4 binding, indicating the indirect effects of SALL4 regulation. Regarding Close-Open dynamics, the chromatin regions in CO1 exhibit relatively higher direct SALL4 binding and become open at later stages. (Supplementary Fig. 7e). These results highlight both the direct and indirect effects of SALL4 in reprogramming.

Combining the analysis of these omics data, we observed that the reprogramming-promoting genes such as *Oct4, Esrrb, Tfap2c, Rsk1, Lin28a, Tbx3, Kdm2b, Cdh1, Cldn7* and *Rcor2*[21,33–37] were opened in SALL4 system (Fig. 3e and Supplementary Fig. 8a). Conversely, genes associated with somatic cell characteristics and known reprogramming suppressors, including *Elk3, Cdkn2a, Cdkn2b, Fosl1* and *Jun*[38–40] were closed in SALL4 system (Fig. 3e and Supplementary Fig. 8b). Corresponding to these chromatin accessibility changes, the RNA expression levels of the reprogramming-promoting genes were upregulated, while those of the somatic cell-related and reprogramming suppressor genes were downregulated (Supplementary Fig. 8c, d). To further investigate the function of SALL4 downstream target genes (*Rsk1, Esrrb, Tfap2c*) and the ATAC-motif-related genes (*Nkx6.1, Esrrb, Tfap2c*) in reprogramming (Fig. 3d, e), we performed overexpression and knockdown experiments by retroviral infection and showed that when *Rsk1, Esrrb* or *Nkx6.1* was co-overexpressed with *Sall4*, a significant improvement in reprogramming efficiency was observed, and knockdown of *Tfap2c* and *Esrrb* defects the reprogramming (Fig. 3f and Supplementary Fig. 8e). Knockdown of *Rsk1* also slightly inhibit the reprogramming (Fig. 3f and Supplementary Fig. 8e). These results suggest that the presence of RSK1, ESRRB, TFAP2C and NKX6.1 in SALL4-driven reprogramming could facilitate iPSCs generation. In summary, SALL4 regulates gene expression through direct binding to target genes and indirect regulation, thereby promoting the reprogramming.

## SALL4 cooperating with OCT4 improves the iPSCs induction efficiency

OCT4 alone has been identified as a mediator for reprogramming[9,10,41], and the activation of endogenous OCT4 is vital for achieving pluripotency[42–44]. Thus, OCT4 plays a critical role in reprogramming. In our study, we investigated the capacity of OCT4 alone and in combination with SALL4 to induce iPSCs under iCD4 medium (Fig. 4a). Our results showed that OCT4 alone could generate iPSCs in iCD4 medium (Fig. 4b, c). Epithelialization was observed on the fourth day of induction in all three conditions (overexpression of *Sall4, Oct4,* or *Sall4* + *Oct4* through retrovirus infection). OCT4-GFP⁺ cells were generated on Day 6 in the *Sall4* + *Oct4* group, and *Sall4* or *Oct4* group induced the appearance of OCT4-GFP⁺ cells on Day7 or Day9, respectively (Fig. 4b). On Day 10, we counted the number of OCT4-GFP⁺ colonies and found that the efficiency of OCT4-induced reprogramming was lower than that of SALL4 under iCD4 conditions. However, co-overexpression of SALL4 and OCT4 significantly increased the efficiency of somatic cell reprogramming (Fig. 4c). Furthermore, iPSCs obtained from the three methods exhibited stable passaging with pluripotency characteristics (Fig. 4d and Supplementary Fig. 9a–e). Exogenous genes were silenced in OCT4-GFP⁺ cells, and the transgene genomic integration was confirmed by PCR during iPSCs induction (Supplementary Fig. 9f–h). In addition, using mouse tail fibroblasts as the starting cells, O + S and OCT4 could induce them into OCT4-GFP⁺ cells (Supplementary Fig. 9i). In summary, our findings indicate that iCD4 can also induce OCT4-mediated reprogramming despite the low efficiency. The co-overexpression of SALL4 and OCT4 significantly improves the efficiency of reprogramming. This suggests that SALL4 can synergistically promote reprogramming with OCT4.

To investigate the molecular mechanism of iPSCs reprogramming mediated by OCT4 and SALL4, we collected transcriptomic data for SALL4, OCT4 and O + S systems, using the DsRed system as a control (Supplementary Fig. 10a). We first performed PCA analysis and showed that each system displayed a unique reprogramming pathway, with the O + S system positioned between the OCT4 and SALL4 systems (Supplementary Fig. 10b). Next we performed gene expression clustering analysis for this three systems. The findings indicated that the activation and silencing of genes during O + S-driven iPSCs generation were relatively more similar to the expression profile of ESCs than those of the other systems. The SALL4 system showed relatively higher similarity to the O + S system than to the OCT4 system (Fig. 4e, f). We also conducted a GO analysis on the upregulated and downregulated genes in each system. Notably, Genes in the UC7 cluster, which were upregulated by both SALL4 and OCT4 + SALL4, were found to be involved in biological processes such as epithelial cell morphogenesis, spermatogenesis, and stem cell proliferation. The genes within the UC10 cluster, commonly upregulated in both the OCT4 and O + S systems, are associated with biological processes related to neuronal differentiation and development. Conversely, genes in the UC19 cluster,

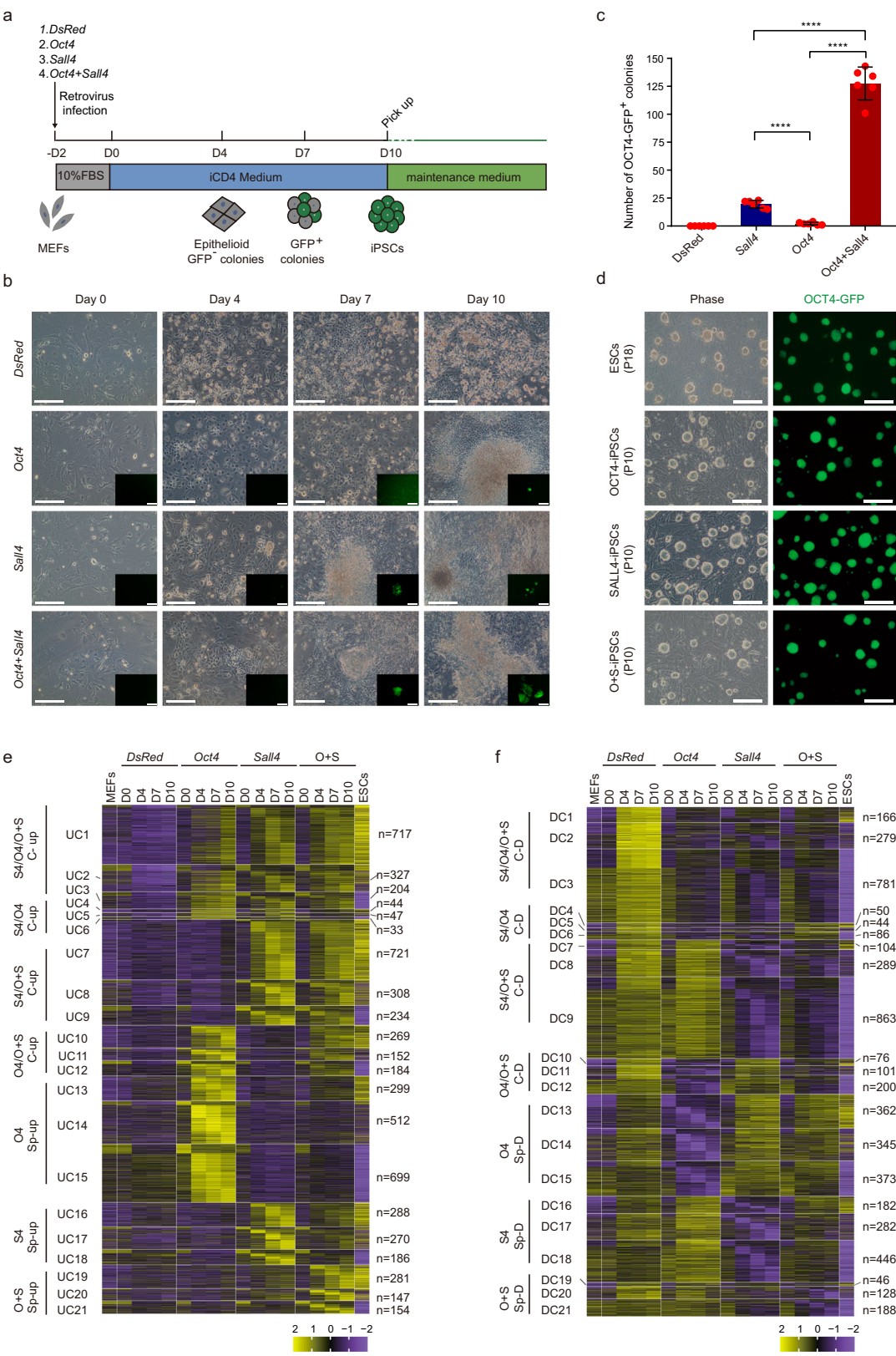

specifically upregulated in the O + S system, are linked to biological processes such as DNA methylation involved in gamete generation, spermatogenesis, and maintenance of stem cell populations (Supplementary Fig. 10c). On the other hand, genes in the DC9 cluster, commonly downregulated in both the SALL4 and O + S systems, were found to be involved in biological processes such as positive

regulation of neuronal projection development and nervous system development. The genes within the DC12 cluster, commonly downregulated in both the OCT4 and O + S systems, are associated with biological processes related to the negative regulation of neuron apoptotic processes and inflammatory responses. Genes in the DC21 cluster, specifically downregulated in the O + S system, are linked to

**Fig. 4 | Co-overexpressing of SALL4 with OCT4 in MEFs facilitates the iPSCs reprogramming efficiency. a** Schematic of iPSCs induction from mouse fibroblasts using exogenous genes and small molecules. **b** Morphological diagram for the iPSCs induction process. Scale bars, 200 µm. The experiments were repeated independently three times with similar results. **c** Number of OCT4-GFP$^+$ colonies from $3 \times 10^4$ MEFs infected with *Sall4*, *Oct4*, or *Sall4 + Oct4* induced by iCD4 on Day 10. MEFs infected with DsRed as control are shown. Data are mean ± SD. Statistical analysis was performed using a two-tailed, unpaired *t* test; *n* = 6 well from 3 independent experiments. *Sall4* versus *Oct4*, ****$p$ = 0.0000179; *Sall4* versus *Oct4 + Sall4*, ****$p$ = 0.0000048; *Oct4* versus *Oct4 + Sall4*, ****$p$ = 0.0000043. **d** The morphology of P10 iPSCs colonies derived from MEFs induced by SALL4, OCT4, or SALL4 + OCT4 in iCD4. ESCs as control are shown. Scale bars, 200 µm. O + S, OCT4 + SALL4. The experiments were repeated independently three times with similar results. **e** Heatmaps show the RNA-seq data classification of SALL4, OCT4, and O + S-relative up-regulated gene expression compared with DsRed. MEFs and ESCs as controls are shown. The subgroups(UC1-UC21) were based on the fold change of gene expression between MEFs and ESCs: e.g., UC1 of S4/O4/O + S C-up category represents GCN(gene counts number) in ESCs / GCN in MEFs ≥ 2; UC2 represents 0.5 < GCN in ESCs / GCN in MEFs < 2; and UC3 represents GCN in ESCs / GCN in MEFs ≤ 0.5. Other major categories were classified similarly. DR, DsRed system. O4, OCT4 system. S4, SALL4 system. O + S, OCT4 + SALL4 system. C-up, common upregulated. sp-up, specifically upregulated. **f** Heatmaps show the RNA-seq data classification of SALL4, OCT4, and O + S-relative down-regulated gene expression compared with DsRed. MEFs and ESCs as controls are shown. The subgroups(DC1-DC21) were based on the fold change of gene expression between MEFs and ESCs: e.g., DC1 represents GCN in ESCs / GCN in MEFs ≤ 2; DC2 represents 0.5 > GCN in ESCs / GCN in MEFs > 2; DC2 represents GCN in ESCs / GCN in MEFs ≥ 0.5. Other major categories were classified similarly. DR, DsRed system. O4, OCT4 system. S4, SALL4 system. O + S, OCT4 + SALL4 system. C-D, common downregulated. sp-D, specifically downregulated.

biological processes such as raft assembly, endocytosis, and integrin-mediated signaling pathway (Supplementary Fig. 10d). In addition, the genes within the UC15 cluster, specifically upregulated in the OCT4 system, are associated with biological processes related to synapse organization. These genes might impede the reprogramming process and can be suppressed by the addition of SALL4 (Fig. 4e and Supplementary Fig. 10e). Overall, our GO analysis suggests that SALL4 could promote epithelial cell formation and inhibit neuron development related genes in O + S system to facilitates iPSCs induction.

## Mapping the cell fate transition during OCT4 + SALL4-induced reprogramming by Single-Cell RNA Sequencing

To obtain a more comprehensive understanding of the molecular roadmap associated with SALL4 and SALL4 + OCT4-mediated reprogramming, we performed single-cell RNA sequencing at various time points throughout the SALL4 and SALL4 + OCT4 reprogramming process, specifically collecting samples on days 0, 4, 7, and 10.

We utilized UMAP plots to visualize cell fate transitions in both reprogramming systems, revealing significant changes from day 0 to day 4 (as depicted in Supplementary Fig. 11a, b). The observation of a relatively lower number of Nanog-positive cells in the SALL4 system compared to the O + S system on day 10 further substantiated the cooperative effect of SALL4 and OCT4 during reprogramming. Notably, iPSCs exhibited a closer clustering with ESCs than D10-*Nanog* positive cells in both systems (as demonstrated in Supplementary Fig. 11a–d), suggests that while most *Nanog* positive cells emerged at day10 may not fully mature into iPSCs, these cells can achieve maturation when cultured with ESCs maintenance medium. Furthermore, we have also observed the upregulation or downregulation of SALL4-regulated reprogramming-promoting and barrier genes in distinct cell subpopulations during both reprogramming processes (Supplementary Fig. 11c, d).

To further elucidate the trajectory of cellular differentiation during reprogramming, we performed monocle trajectory analysis on days 0, 4, 7, 10 and iPSCs in both systems (Supplementary Fig. 11e, f). The results revealed the emergence of two distinct developmental branches during the reprogramming process, which were not readily discernible in UMAP plotting. We characterized one branch as likely to achieve pluripotency potential (pluripotency branch), based on its alignment with iPSCs-reprogramming directions (Supplementary Fig. 11e, f). Importantly, cells within the pluripotency branch in the O + S system exhibited a more uniform distribution compared to those in the SALL4 system, suggesting that SALL4 and OCT4 collaboratively induce a state of cellular plasticity conducive to acquiring pluripotency more efficiently than SALL4 alone.

In addition, in order to distinguish differences in reprogramming intermediates across various systems, we conducted a comparative analysis of the differential gene expression for THY1-/EPCAM + cells within the SALL4, O + S, and OKS systems. This analysis revealed variations in transcriptional regulations across different reprogramming processes, as indicated by both gene quantity and the functions annotated through GO analysis of differential expression genes (Supplementary Fig. 12a–g).

## SALL4 activates *Esrrb*, *Rsk1*, and *Tfap2c* in O + S-mediated iPSCs reprogramming to facilitate induction efficiency

To gain insights into the changes in chromatin accessibility mediated by OCT4 and SALL4 during reprogramming, we compared the ATAC-seq data of the SALL4 system, OCT4 system, and SALL4 + OCT4 system (O + S system) at Day 0, 4, 7, and 10 (Supplementary Fig. 10a). Using CO-OC analysis, we categorized all peaks of the O + S system into 11 groups based on the transition from open to close or from close to open. These groups included persistent open (PO), open to closed (OC1-5), and closed to open (CO1-5) (Fig. 5a). Comparing these systems, we found that the O + S system is more effective than the SALL4 and OCT4 systems in regulating correct chromatin opening and closing, with the SALL4 system being superior performance over the OCT4 system (Fig. 5a). This is consistent with the comparison of their induction efficiency. Subsequently, we conducted motif enrichment analysis in the three reprogramming systems. The results revealed an increase in the abundance of binding motifs for transcription factors known to promote reprogramming, such as OCT4, SOX2, NANOG, TFAP2C, and ESRRB, in the O + S system. Conversely, motifs associated with inhibiting reprogramming, such as CLOCK, and AP1, showed a decrease in abundance (Fig. 5b).

Combining the analyses of the SALL4, OCT4, and O + S systems, we proposed six patterns in which OCT4 and SALL4 cooperatively regulate CADs in the O + S system and defined the peak numbers of each pattern (Supplementary Fig. 13a). These patterns are reflected in chromatin accessibility as follows: (1) Common open in O + S system and SALL4 system (O + S/S4-C-O), (2) Common close in O + S system and SALL4 system (O + S/S4-C-C), (3) Common open in O + S system and OCT4 system (O + S/O4-C-O), (4) Common close in O + S system and OCT4 system (O + S/O4-C-C), (5) only open in O + S system (O + S-S-O), and (6) only close in O + S system (O + S-S-C) (Supplementary Fig. 13a). We further categorized the genes associated with these CADs patterns based on transcription levels using RNA-seq data and revealed numerous changes in gene expression that fit these types of synergistic modes (Fig. 5c, d and Supplementary Fig. 13b–e). We hypothesize that OCT4 and SALL4 improve iPSCs induction efficiency through their regulation of genes within these patterns, where patterns 1, 3, and 5 likely contain genes that promote reprogramming in the O + S system, while patterns 2, 4, and 6 may contain genes that hinder reprogramming. To support this hypothesis, we selected genes based on expression levels, peak enrichment, and functional relevance. Representative potential reprogramming-promoting genes that have been identified include *Esrrb*, *Tfap2c*, *Rsk1*, and *Sox2* for their roles in facilitating the induction of iPSCs in the SALL4 and OKS systems.

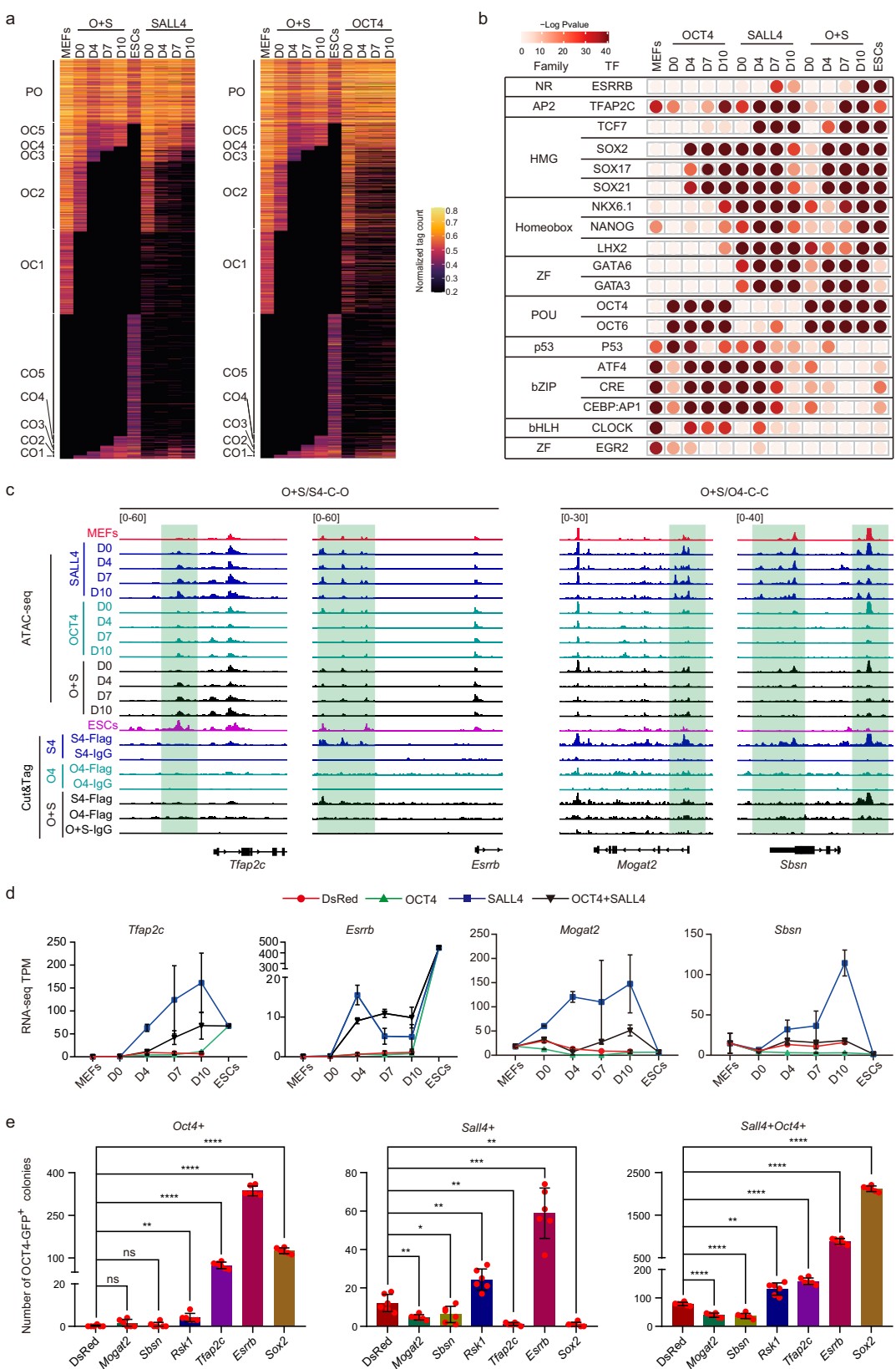

Conversely, genes such as *Mndal*, *Mogat2*, and *Sbsn* were identified as potential barriers to reprogramming due to their somatic-related functions and high levels of expression and peak enrichment (Fig. 5c, d and Supplementary Fig. 13b–e).

We next explore and compare the reprogramming abilities of these representative genes during iPSCs induction. Overexpression

of *Esrrb* and *Rsk1* through retroviral infection significantly promotes the generation efficiency of OCT4-GFP+ colonies in the OCT4, SALL4, or O + S systems, Conversely, *Mogat2* and *Sbsn* impair the iPSCs induction for these three systems, while *Mndal* exerts an inhibitory effect in the O + S systems (Fig. 5e and Supplementary Fig. 13f). Overexpression of *Tfap2c* or *Sox2* by retroviral infection promotes

**Fig. 5 | The dynamics of chromatin accessibility during SALL4, OCT4, or O + S-iPSCs induction. a** CADs for O + S system, SALL4 system and OCT4 system. PO, permanently open. CO, close to open. OC, open to close. Take O + S system as reference is shown. OCT4, OCT4 system. SALL4, SALL4 system. O + S, OCT4 + SALL4 system. **b** Motif analysis of the SALL4, OCT4, or O + S-specific enrichment peaks at D0, D4, D7, and D10 during iPSCs induction, the peaks for each day in DsRed system as a background was removed from the peaks at D0, D4, D7 and D10 in SALL4, OCT4, and O + S system. Statistical analysis was performed using Karlin/Altschul statistics, and the -Log10(*p*-value) for motif is shown. OCT4, OCT4 system. SALL4, SALL4 system. O + S, OCT4 + SALL4 system. **c** Representative Genomic loci and genes for OCT4 + SALL4 / SALL4-Common-open peaks and OCT4 + SALL4 / OCT4-Common-close peaks. O4, OCT4 system. S4, SALL4 system. O + S, OCT4 + SALL4 system. O + S/S4-C-O, OCT4 + SALL4 / SALL4-Common-open; O + S/O4-C-C, OCT4 + SALL4 / OCT4-Common-close. **d** RNA-seq data shows the expression of representative genes in Fig. 5c. Data are mean ± SD. *n* = 2 samples for each time point from 2 independent experiments. **e** The iPSCs induction efficiency using *Sall4, Oct4,* or *Oct4 + Sall4* overexpressing with representative S4/O + S-common open genes (*Rsk1, Esrrb* and *Tfap2c*), O4/

O + S-common open genes (*Sox2*) and O + S/OCT4-Common-close genes (*Sbsn* and *Mogat2*). DsRed, *Sall4, Oct4,* and *Oct4 + Sall4* as a control are shown. Data are mean ± SD. Statistical analysis was performed using a two-tailed, unpaired *t* test; *n* = 6 well from 3 independent experiments. *Oct4* + DsRed versus *Oct4* + *Mogat2*, n.s. *p* = 0.1248925; *Oct4* + DsRed versus *Oct4* + *Sbsn*, n.s. *p* = 0.3742057; *Oct4* + DsRed versus *Oct4* + *Rsk1*, \*\**p* = 0.0034229; *Oct4* + DsRed versus *Oct4* + *Tfap2c*, \*\*\*\**p* = 0.0000169; *Oct4* + DsRed versus *Oct4* + *Esrrb*, \*\*\*\**p* = 0.0000001; *Oct4* + DsRed versus *Oct4* + *Sox2*, \*\*\*\**p* = 0.0000007; *Sall4* + DsRed versus *Sall4* + *Mogat2*, \*\**p* = 0.0077033; *Sall4* + DsRed versus *Sall4+Sbsn*, \**p* = 0.0382438; *Sall4* + DsRed versus *Sall4* + *Rsk1*, \*\**p* = 0.0019697; *Sall4* + DsRed versus *Sall4* + *Tfap2c*, \*\**p* = 0.0015495; *Sall4* + DsRed versus *Sall4* + *Esrrb*, \*\*\**p* = 0.0001515; *Sall4* + DsRed versus *Sall4* + *Sox2*, \*\**p* = 0.0012618; *Sall4* + *Oct4* + DsRed versus *Sall4* + *Oct4* + *Mogat2*, \*\*\*\**p* = 0.0000019; *Sall4* + *Oct4* + DsRed versus *Sall4* + *Oct4* + *Sbsn*, \*\*\*\**p* = 0.0000092; *Sall4* + *Oct4* + DsRed versus *Sall4* + *Oct4* + *Rsk1*, \*\**p* = 0.0012754; *Sall4* + *Oct4* + DsRed versus *Sall4* + *Oct4* + *Tfap2c*, \*\*\*\**p* = 0.0000006; *Sall4* + *Oct4* + DsRed versus *Sall4* + *Oct4* + *Esrrb*, \*\*\*\**p* = 0.0000008; *Sall4* + *Oct4* + DsRed versus *Sall4* + *Oct4* + *Sox2*, \*\*\*\**p* = 0.00000001.

---

the generation efficiency of OCT4-GFP⁺ colonies in the OCT4 or O + S systems but has an inhibitory effect on the induction efficiency of the SALL4 system (Fig. 5e).

In summary, OCT4 and SALL4 collaboratively activate genes like *Esrrb*, *Rsk1*, *Tfap2c*, and *Sox2* to enhance iPSCs induction and synergistically repress genes such as *Mndal*, *Mogat2*, and *Sbsn* that act as reprogramming barriers within the O + S system. This dual regulation of promotive and inhibitory genes by OCT4 and SALL4 effectively boosts reprogramming efficiency in the O + S system (Fig. 6a).

### The chromatin binding dynamics for OCT4 and SALL4 during O + S-induced reprogramming
To explore the cooperative role of SALL4 and OCT4, we conducted Cut&Tag assays for OCT4 during OCT4/O + S-driven reprogramming and for SALL4 during O + S-driven reprogramming on day 0 (Supplementary Fig. 14a). We initially analyzed the genomic distribution of these Cut&Tag peaks and subsequently performed a Gene Ontology (GO) analysis for genes associated with these peaks. The results demonstrated diverse biological processes regulated by SALL4 and OCT4 (Supplementary Fig. 14b, c). We proceeded to compare the SALL4-binding peaks between the SALL4 system and the O + S system, revealing 19,820 new peaks (C2) and 13,466 disappeared peaks (C1) in the O + S system (Supplementary Fig. 14d). Gene Ontology (GO) analysis identified functional roles of genes within these clusters (Supplementary Fig. 14e). To further study the change of the binding pattern in O + S system, we conducted a comparison of the SALL4-binding peaks and OCT4-binding peaks between the SALL4 system and OCT4 system, revealing approximately 6180 peaks that were commonly bound by both SALL4 and OCT4 (C3) (Supplementary Fig. 14f). SALL4 and OCT4 are theoretically capable of commonly binding to these predicted sites to regulate these regions. In actuality, the number of common binding peaks between SALL4 and OCT4 in O + S-driven reprogramming increased to 9769 peaks (C4). When comparing the predicted sites (C3) with the actual binding sites (C4), the results revealed approximately 2418 predicted peaks disappeared (C5), while 6053 peaks emerged (C6) in O + S-driven reprogramming (Supplementary Fig. 14f). Further Gene Ontology (GO) analysis identified the functional roles of genes annotated by these clusters(C5 and C6) (Supplementary Fig. 14g). In addition, the analysis revealed that the proportion of sites near the promoter (< = 1kb) that increased (C6, 32.62%) and decreased (C5, 41.89%) in the O + S system was notably higher than that observed in the single-factor systems (SALL4 system, 21.43%; OCT4 system, 25.06%) (Fig. 2b and Supplementary Fig. 14b, h). This suggests that the synergistic effect of SALL4 and OCT4 in the O + S system is more inclined towards the regulation of promoter regions.

In addition, we investigated the relationship between factors binding and chromatin accessibility dynamics, we compared the SALL4 and OCT4 Cut&Tag peaks with ATAC-seq peaks in the O + S-system at day 0 (Supplementary Fig. 14i). The results show that most of SALL4 and OCT4 Cut&Tag peaks were overlapped with ATAC-seq peaks in O + S-system, suggest the important role of these DNA binding. The broader binding regions, in cooperation with SALL4 and OCT4, may also regulate more chromatin accessibility dynamics to promote reprogramming (Supplementary Fig. 14i).

To further investigate the regulatory dynamics of SALL4 and OCT4, we analyzed their binding patterns within the O + S system by comparing the genes associated with binding peaks, SALL4 (or OCT4)-ATAC-CO peaks-related genes, and genes specifically upregulated by SALL4 (or OCT4). We identified gene sets that are specifically elevated and exhibit open chromatin due to SALL4 (C1, comprising 26 genes) or OCT4 (C2, comprising 56 genes) influence (Supplementary Fig. 15a, c). Subsequently, we compared these gene sets with the genes associated with binding peaks in the SALL4, OCT4, and O + S systems, respectively. The results illustrate a reduction in the number of SALL4-binding genes within the C1 gene set in the O + S system compared to the SALL4 system alone, suggesting that OCT4's addition may alter SALL4's occupancy landscape within the O + S system (Supplementary Fig. 15a, b). In addition, the number of OCT4-binding genes in the C2 gene set also shows a reduction pattern in O + S systems, whereas the number of SALL4-binding genes was significantly larger, suggesting that SALL4's binding might be related to the downregulation of these genes (Supplementary Fig. 15c, d). This analysis highlights the complex regulatory interplay between SALL4 and OCT4 in modulating gene expression and chromatin accessibility during cellular reprogramming.

## Discussion
SALL4 is an important gene in early embryonic development and somatic cell reprogramming[13,18,20,26]. However, the regulatory mechanisms of SALL4 in iPSCs reprogramming are still largely unknown and require further exploration. In this study, we demonstrated that SALL4 alone could mediate reprogramming (Fig. 1a–c). In the process, we found that RepSox[45–47] could facilitate iPSCs induction, but its specific mechanism still needs further investigation (Supplementary Fig. 1a, d). The induction efficiency remains lower than that of multi-factor reprogramming. This suggests the need for further optimization of the induction medium for SALL4-iPSCs induction. In addition, we revealed the dynamic changes of transcriptional levels and chromatin accessibility induced by SALL4 (Fig. 3a and Supplementary Fig. 4c). We also found an enrichment of the AP-1-related motif TGACTCA in SALL4 binding sites(Fig. 2e).

**Fig. 6 | The model for SALL4-drived iPSCs induction and SALL4 cooperated with OCT4 to enhance iPSCs generation. a** A model for SALL4-drived iPSCs induction and SALL4 cooperated with OCT4 to enhance iPSCs generation.

Transcription factors with similar binding motifs are reprogramming barriers[48,49], such as JUN. But our data shows a reprogramming promoting role of *Batf*. The incorporation of the BATF-DNA binding domain into WT-SALL4 also enhances reprogramming. In addition, the deficiencies resulting from the deletion of SALL4 ZFC1 or ZFC2 are restored by the addition of the BATF-DNA binding region (Supplementary Fig. 5j). Previous studies have shown that BATF can form heterodimers with the reprogramming barrier gene C-JUN through its bZIP domain, functioning as a component of the AP-1 complex and acting as a negative modulator of the complex's transcription potential[40,50,51]. The formation of the C-JUN/BATF complex may alter the transcription activity of AP-1-regulated genes and reduce the inhibitory role of C-JUN during reprogramming. Thus, SALL4 may bind and regulate the downstream genes of related reprogramming-promoting factors and reprogramming barrier factors to regulate reprogramming.

We showed that the reprogramming factor ESRRB[52,53] and a Ser-ine/threonine-protein kinase RSK1 were specifically occupied and activated by SALL4 to promote reprogramming(Fig. 3e, f and Supplementary Fig. 8c). Our findings indicate that the knockdown of *Tfap2c* within the SALL4 system hampers reprogramming, highlighting the critical role of TFAP2C activation in this process. However, contrary to expectations, overexpressing *Tfap2c* actually inhibits reprogramming. Intriguingly, this inhibitory effect seems to intensify with higher doses of the *Tfap2c* virus (Supplementary Fig. 8f). This paradoxical observation leads us to speculate that premature activation of *Tfap2c* may obstruct reprogramming, suggesting that the timing of its activation might be crucial for realizing its positive impact on the reprogramming process. In addition, we found that *Nkx6.1*, which is involved in transcriptional regulation of the insulin gene[54,55], could specifically promote SALL4-mediated reprogramming, although this gene is not expressed in this process (Fig. 3f). ATAC motif enrichment analysis

revealed the NKX6.1 binding sites were enriched in the genome, suggests that NKX6.1 may promote reprogramming by binding to the relevant motifs (Fig. 3d).

The induction efficiency of the O + S system is significantly higher than that of the SALL4 or OCT4 systems (Fig. 4c). SALL4 and OCT4 can interact with chromatin modification complexes and cause changes in chromatin accessibility, which affect the maintenance of pluripotency[13,56–59]. In addition, SALL4 and OCT4 each have "independent" regulatory regions in the pluripotency regulatory network[59], suggesting that the high efficiency of the O + S system may be the result of mutual compensation and synergy between the two factors. At the transcriptional level, both the SALL4 and OCT4 systems exhibit gene groups with expression levels differing from the ESCs state, potentially impeding the induction efficiency of each system. However, these genes can be rescued by the collaborative action of SALL4 and OCT4 in the O + S system, leading to a more 'neutral' outcome. For instance, our investigation revealed that SALL4 specifically activates genes like *Rsk1, Esrrb*, and *Tfap2c*[60]. Notably, these genes promote the reprogramming within the SALL4 system. It's surprising to note that *Rsk1, Esrrb*, and *Tfap2c* demonstrate a consistent promoting effect on reprogramming in both the O + S and OCT4 systems, suggesting their significance as drivers in O + S-mediated reprogramming(Fig. 5e). However, OCT4 alone is incapable of activating the expression of *Esrrb* or *Tfap2c*(Fig. 5d). Therefore, in the O + S system, SALL4 takes charge of activating the expression of *Rsk1, Esrrb*, and *Tfap2c*, thereby promoting the O + S-mediated reprogramming process.

Both the OCT4 and O + S systems exhibit specific expression of *Sox2*. Overexpression experiments have demonstrated that SOX2 significantly contributes to the promotion of reprogramming in both the OCT4 and O + S systems. However, *Sox2* is expressed at low levels and acts as a barrier in the SALL4 system (Fig. 5e and Supplementary Fig. 13e). Introducing *Sox2* into the SALL4 system led to the downregulation of a cluster of genes that promote reprogramming and the upregulation of a set of genes associated with ectodermal development, ultimately directing the cell towards an alternative fate, as illustrated in Supplementary Fig. 16a–e. Moreover, the reprogramming barrier genes within the O + S system can be suppressed through the interaction of SALL4 and OCT4. For instance, *Mndal*, which can be activated by OCT4, plays a role in inhibiting the O + S-mediated reprogramming (Supplementary Fig. 13e, f). However, the expression of *Mndal* is suppressed by the addition of SALL4. Similarly, reprogramming barrier genes activated by SALL4 in the SALL4 system, such as *Mogat2* (2-acylglycerol O-acyltransferase2)[61] and *Sbsn* (associated with the differentiation of keratinocytes)[62,63], can be downregulated by OCT4 in O + S system (Figs. 5d, e and 6a) and these genes may inhibit reprogramming through regulating the changes of somatic cell differentiation-related genes(Supplementary Fig. 15e, f). Furthermore, there exists a subset of genes uniquely up- or down-regulated in the O + S system, suggesting their expression relies on the concurrent regulation by both OCT4 and SALL4(Fig. 4e, f). In addition, transcription factors such as TCF7 and LHX2, may also exert regulatory influence in the O + S system for their motif enrichment pattern and gene expression level in our analysis (Fig. 5b and Supplementary Fig. 13g). However, overexpression of these genes causes an inhibitory effect in the O + S system (Supplementary Fig. 13f, g). These findings imply the existence of additional reprogramming-promoting and inhibiting genes that may not be appropriately regulated by OCT4 and SALL4 but could potentially be controlled by other transcription factors.

Hence, the collaboration between SALL4 and OCT4 not only amplifies their individual reprogramming capabilities but also combats elements that hinder the process. This collaboration broadens the scope of gene regulation, fostering the activation of more reprogramming-associated genes and expediting the suppression of somatic cell-related genes' expression. As a result, it significantly improves the efficiency of induced pluripotent stem cell generation.

Our research delves into the complex interactions between OCT4 and SALL4, primarily exploring gene expression regulation. However, this interaction's complete mechanisms and contributions are yet to be fully uncovered. The diverse roles of transcription factors in different reprogramming systems implied the complexity of interactions and the diverse impact of specific transcription factors in various reprogramming contexts. We believe that further research is necessary to uncover these complex regulatory effects in reprogramming. In addition, Epigenetic regulation, especially SALL4's roles in chromatin remodeling and the regulation of chromatin 3D structure stands as a critical area for future investigation, potentially offering alternative principles for cell fate control during reprogramming.

## Methods

### Mice
NCG and ICR mice were purchased from GemPharmatech. OCT4-GFP(OG2) transgenic mice (CBA/CaJ X C57BL/6 J) were purchased from The Jackson Laboratory. All mice were housed in the specific pathogen-free animal facility under a 12 hr light/12 hr dark cycle and provided food and water in abundance. All experiments related to animals were performed on the basis of the Animal Protection Guidelines of Guangzhou Institutes of Biomedicine and Health (GIBH), and approved by the Committee on the Ethics of Animal Experiments at GIBH.

### MEFs preparation and cell culture
129S4/SvJaeJ female mice were crossed with male OCT4-GFP transgenic allele-carrying mice (CBA/CaJ 3 C57BL/6 J), picked out the mice with vaginal plug the next day, and marked as day 0, after 13.5 days, mouse embryos were isolated from pregnant mice for MEFs preparation. mouse embryos that separated the integral organs, head, limbs, and tail were dissociated as a single cell suspension using 0.25% trypsin (#25200114,Gibco), then seeded onto a 0.1% gelatin-coated culture dish.

Plat-E and MEFs were cultured in 10%FBS medium(high glucose DMEM (#SH30022.01,HyClone) supplemented with 1% NEAA (#11140076,Gibco), 1% GlutaMax (#17504044,Gibco) and 10% FBS(#BVS500,Biovision)). ESCs/iPSCs were cultured in KSR-2iLIF medium (Knockout DMEM supplemented with 1% Non-essential Amino Acids, 1% GlutaMax, 1% Sodium Pyruvate, 0.1 mM β-mercaptoethanol, 15% KSR (#10828028,Gibco), 1 μM PD0325901, 3 μM Chir99021, 1000 units/ml mLIF).

### Immunofluorescence analysis
Cultured the iPSCs colonies in 24 well plates. When the state and size of iPSCs colonies are well enough, wash the iPSCs colonies with PBS three times. Add 150 μl 4% PFA to cells for 30 min at room temperature to fix it. Discard 4% PFA and wash the cells with PBS three times. Then add an equal volume of 0.1% Triton X-100 and 3% BSA to penetrate and block at room temperature. Subsequently, the cells were treated with PBS three times and incubated with primary antibody at 4°C overnight. After washing in PBS five times, two hours of incubation in the second antibody at room temperature away from light. After that, cells were washed in PBS three times and incubated in DAPI for 1 min. Finally, add 500 μl PBS and observe by inverted fluorescence microscope. The following primary antibodies were used: anti-OCT4 (#ab19857,Abcam,1:1000), anti-Sox2 (#3579,Cell Signaling Technology, 1:200), anti-Nanog (#8822S,Cell Signaling Technology, 1:1000).

### Western blot
After being dissociated and counted, $1 \times 10^6$ cells were collected and lysed using RIPA Lysis Buffer (Beyotime, Shanghai, China) with $100 \times$ PMSF (Beyotime) and $100 \times$ protease inhibitor cocktail (Roche) on ice for 5 min and boiled for 10 min at 100 °C. The samples were separated using 10% SDS-PAGE, transferred onto a polyvinylidene difluoride membrane (Millipore) using a wet transfer system, and then

incubated with the primary antibodies and secondary antibodies. The following primary antibodies were used: anti-FLAG (#F1804, Sigma,1:1000), anti-BATF(#sc-100974, Santa Cruz, 1:500) and anti-GAPDH (#60004-1-lg, ProteinTech, 1:1000).

## Flow cytometry
Cells were dissociated into a single cell using 0.25% trypsin and collected by centrifugation. The cell pellet was resuspended with PBS. Added the Thy1.2 antibody (#48-0902-82, Invitrogen, 1:200) and Epcam antibody (#12-5791-81, Invitrogen, 1:200) in cell suspension and incubated for 30 min. Washed twice with PBS followed by filtration using a cell strainer (BD Biosciences) to remove large clumps of cells. The cells were then analyzed and sorted on the FACSAria II flow cytometer (BD Biosciences). The sorted cells were used for reprogramming. Data analysis was done using FlowJo software.

## Teratoma formation
iPSCs were dissociated as single cells and resuspended to $1 \times 10^6$ using 0.1 mL 10% FBS medium. Then injected subcutaneously into 6 week-old immunodeficiency NCG mice. After 4 weeks, teratomas were dissected and fixed with 4% paraformaldehyde (PFA) for 24 hours at room temperature. The fixed teratomas were embedded in paraffin and section for HE staining.

## Plasmid construction and retrovirus production
The full-length CDS of individual factors (*Sall4, Oct4, Sox2, Rsk1, Tfap2c, Esrrb, Nkx6.1, Mogat2, Sbsn, Batf, Jun, Junb, Fos, Fosl1, Fosl2, Atf3*) were amplified from cDNA of ESCs or MEFs by PCR, The amplified products were cloned into the pMXs expression vector. The shRNA targeting *Tfap2c, Esrrb, Rsk1,* and the control shRNA (shLuc) were cloned into the pSuper expression vector.

retrovirus was generated from Plat-E cells with the PEI-method. Briefly, $7.5 \times 10^6$ Plat-E Cells were seeded onto 10 cm dish and cultured in 10% FBS medium for 12–18 h to perform PEI transfection. Mixed 20ug plasmid and 40ul PEI reagent (#MW40000, YEASEN) into DMEM incubate for 15 mins at room temperature. Then, add the mixture into the Plat-E cell and refresh the medium after 10 hours. 48 hours post-transfection, harvested the retroviral supernatants.

## Generation of iPSCs from MEFs and induction of OCT4-GFP⁺ cells from TTFs
$3 \times 10^4$ cells were plated onto each well of 24 well plates, after 12 h later, attached MEFs (or TTFs) were prepared for retrovirus infection. In the first rounds of the infection process, 250ul 10% FBS medium and an equal volume of each retrovirus supernatant were added in a well of 24 well plates, added polybrene (#TR-1003, Sigma-Aldrich) to the mixture to a final concentration of 5 mg/ml and infected for 24 h, After 2 rounds of infection, the medium was changed with iCD4 medium. The medium was refreshed daily. For TTF induction, the medium was added with iCD4 medium plus 2% FBS to reduce cell death.

## Gene overexpression
All overexpression experiments performed in this study utilized the retroviral infection method. The cell and retroviral preparation were the same as aforementioned. For the retroviral infection, 250µl of 10% FBS medium and an equal volume of each retrovirus supernatant were added to a well in a 24-well plate. Polybrene (#TR-1003, Sigma-Aldrich) was then added to the mixture to achieve a final concentration of 5 mg/ml, and the cells were infected for 24 h. Following two rounds of infection, the medium was replaced with an iCD4 medium.

## Gene knockdown
The cell preparation and retroviral infection were the same as aforementioned. After infection, the medium was changed with iCD4 medium plus 1 µg/ml puromycin (#A1113803, GIBCO) to select for

3 days. The GFP-positive colonies were statistic at day10, and the cells were collected for RT-qPCR to confirm the mRNA expression level of the genes after knockdown.

## Image scanning and iPSCs Induction efficiency statistics
Image scanning for a whole well was performed with Nikon BioStation CT on day 10 after iPSCs generation. For iPSCs induction efficiency statistics, GFP-positive Induced pluripotent stem cell colonies were counted in the local computer on the images obtained.

## Induction medium preparation
**iCDx medium.** DMEM (#SH30022.01, HyClone), 0.5 X N2 (#17502048, Gibco), 0.5 X B27 (#17504044, Gibco), 1% Sodium Pyruvate (#11360070, Gibco), 1% GlutMax (#25300120, Gibco), 1% nonessential amino acids (NEAA, #11140076, Gibco), 0.1 mM 2-mercaptoethanol (#M6250-500 ML, Sigma), Vitamin C (50 µg/ml, #49752, Sigma-Aldrich), TV (T: Thiamine HCL, 9 µg/ml, #V-014,Sigma; V: Vitamin B12,1.4 µg/ml, #V6629, Sigma), bFGF (10 ng/ml, #100-18B, PeproTech), LIF (1000 µ/ml, #A35933, ThermoFisher), Y27632 (5 µM, TargetMol), GSK-LSD1 (1 µM, #T22822, TargetMol), SGC0946 (2.5 µM, #S7079,Selleck Chemicals), CHIR99021 (3 µM, Synthesized in GIBH).

**iCD4 medium.** DMEM, 0.5 X N2, 1 X B27, 1% Sodium Pyruvate, 1% GlutMax, 1% nonessential amino acids, 0.1 mM 2-mercaptoethanol, Vitamin C, TV (1000X), bFGF (10 ng/ml), LIF (1000 µ/ml), Y27632 (5 µM), GSK-LSD1 (1 µM), SGC0946 (2.5 µM), CHIR99021 (3 µM), and Repsox (5 µM, CAS No. 446859-33-2, ChemBest).

## Generation of Chimeric Mice form SALL4-iPSCs
The method for the Generation of Chimeric Mice was performed as Guo et al. reported previously[64]. In brief, *Sall4*-iPSCs induced from OG2-MEFs were injected into ICR 8-cell stage embryos, The injected embryos were cultured to blastocyst stage and then implanted to the uteri of pseudopregnant ICR mice at 2.5 dpc.Chimeras could be confirmed by the coat color. To confirm the germline transmission of the chimeric mice, female ICR mice were crossed with male chimeric mice, F2 pups with gray coat color were derived from chimeric mice.

## Quantitative RT-PCR and RNA-seq
Total RNAs were isolated with TRIzol. For quantitative PCR, cDNA was prepared with HiScript II Q RT SuperMix for QPCR (#R222-01, Vazyme), and then qPCR with ChamQ SYBR qPCR Master Mix (#Q311-02, Vazyme). The construction of the RNA library was performed with VAHTS mRNA-seq V3 Library Prep Kit for Illumina (#NR611, Vazyme), and sequencing was performed on the illumina NovaSeq 6000 platform with NovaSeq 6000 S4 Reagent kitV1.5. The qPCR primers for pluripotent genes and relative genes used in this research can be found in Supplementary Table 1.

## ATAC-seq
ATAC-seq was performed as the protocol of TruePrepTM DNA Library Prep Kit V2 for Illumina (#TD501, Vazyme). In brief, approximately 50,000 cells were used per sample. After centrifuging at $500 \times g$ at 4 °C, suspended the cells with 50 µl lysis buffer (10 mM Tris-HCl pH 7.4, 10 mM NaCl, 3 mM MgCl2, 0.2% (v/v) IGEPAL CA-630) and put it in the ice for 10 mins. Then centrifuged at $500 \times g$ for 10 mins at 4 °C, followed by the addition of 50 µl transposition reaction mix (10 µl 5 × TTBL, 5ul TTE Mix V50, 35 µl ddH2O). After 37 °C for 30 mins, the DNA from the cells was extracted and purified. Then, the puried fragment product is amplified by PCR for an appropriate number of cycles. ATAC-seq Libraries were dual-indexed and amplified with TruePrep Index Kit V3 for illumina (#TD203, Vazyme). All ATAC-seq libraries were sequenced on the illumina NovaSeq 6000 platform, and 200–1000 bp paired-end reads were generated.

## Cut&Tag

The MEFs were dissociated into single cells using 0.25% trypsin after two rounds of retroviral infection. The method for obtaining samples for Cut&Tag analysis followed the protocol of Vazyme TD903. In brief, approximately 100,000 cells were used per sample and bound to Concanavalin A-coated beads. The cells were then suspended and incubated with an anti-FLAG antibody (#F1804, Sigma) diluted in antibody buffer at 4 °C overnight. For the negative control data collection, the cells were incubated with IgG instead of the anti-FLAG antibody. After washing, the cells were incubated with secondary antibodies (Rabbit anti-mouse) for 1 h and then incubated with pA-Tn5 transposase in order at room temperature for 1 h. Following transposon activation and tagmentation, DNA from the cells was extracted and purified. Libraries were dual-indexed and amplified with TruePrep Index Kit V3 for Illumina (#TD203, Vazyme). All Cut&Tag libraries were sequenced on the Illumina NovaSeq 6000 platform and generated paired-end reads of 200–1000 bp.

## IP-MS and protein enrichment analysis

**Immunoprecipitation and on-bead digestion.** MEFs overexpressing WT-SALL4 or SALL4-mutants were prepared by using lysis buffer (50 mM Tris pH 8.0, 150 mM NaCl, 10% Glycerol, 0.5% NP40) supplemented with freshly added 1 Complete Protease inhibitors (Sigma, 1187358001). The cells were then incubated for 2 h at 4 °C on a rotation wheel. Soluble cell lysates were collected after centrifugation at a maximum speed of 4 °C for 15 min. Subsequently, 1 mg of nuclear lysates was incubated with Flag-M2 (M8823, Sigma) magnetic agarose beads for 1.5 h and washed three times with cell lysis buffer followed by one wash with PBS. After complete removal of the PBS, immunoprecipitated proteins were digested using the on-bead digestion protocol as previously described[65]. Briefly, beads were incubated with 100 µl of elution buffer (2 M urea, 10 mM DTT, and 100 mM Tris pH 8.5) for 20 min. Subsequently, iodoacetamide (Sigma, I1149) was added to a final concentration of 50 mM for 10 min in the dark, followed by the addition of 250 ng of trypsin (Promega, V5280) for partial digestion over a period of 2 h. The supernatant was collected in a separate tube after incubation. The beads were then subjected to another round of incubation with an additional 100 µl of elution buffer for a further duration of 5 min, and the resulting supernatant was collected in the same tube. All these procedures were carried out at room temperature on a thermoshaker set at 1500 rpm. Combined eluates underwent digestion with an overnight exposure to 100 ng of trypsin at room temperature. Finally, tryptic peptides were acidified to pH < 2 through the addition of 10 ul of 10% TFA (Sigma, #1002641000) and subsequently desalted using C18 Stagetips (Sigma, #66883-U) prior to MS analyses. Each experiment was conducted in technical triplicate.

**Mass spectrometry analysis.** Tryptic peptides were separated using a nanoEase M/Z Peptide BEH C18 column (Waters, 186008795) with a total data collection time of 60 min for peptide separation, followed by two steps of washing with an Easy-nLC 1200 connected online to a Fusion Lumos mass spectrometer (Thermo). Scans were collected in data-independent acquisition mode with dynamic exclusion set at 90 seconds. Raw data was analyzed using DIA-NN search against the Mouse Fasta database, incorporating label-free quantification and match between runs functions. The output protein group was further analyzed and visualized using the DEP2 package as previously described[66,67].

## Single-cell RNA-seq and analysis

The scRNA-seq libraries for SALL4 reprogramming and ESCs were prepared using the Chromium Next GEM Single Cell 3' Kit v3.1 (10× Genomics, PN-1000268) protocol. The O + S reprogramming samples were prepared with the SeekOne® Digital Droplet Single Cell 3'

library preparation kit (SeekGene Catalog No.K00202) according to the manufacturer's instructions.

The method for Single-cell RNA-seq data processing was refered to previously reported methods[68]. The MEFs samples we used in the UMAP plot were from previously reported article[69]. The Monocle (v.2.12)[70] software was utilized to infer the developmental trajectory of reprogramming processes. A new CellDataSet object was constructed from a cluster-annotated Seurat object using the newCellDataSet function. The differentialGeneTest function was employed to identify DEGs within each cluster, and genes with $q < 1 \times 10^{-5}$ were used to order the cells in pseudotime. Dimension reduction was carried out using the DDRTree algorithm, followed by the ordering of cells along the trajectory.

## RNA-seq and gene expression analysis

We conducted RNA-seq analysis using the mm10 reference genome and vM23 annotation. Initially, sequence data underwent adapter trimming with trim_galore. Subsequently, alignment was performed using Hisat2, followed by quantification with featureCounts. we employed DESeq2 for data normalization[71], and functional enrichment analysis was conducted using clusterProfiler[72].

In the context of the SALL4 system, differential gene expression analysis was carried out using DESeq2. Initially, differential expression genes relative to the DsRed system were determined. Genes exhibiting both upregulation and downregulation across different time points within the same system were excluded, and the remaining genes were integrated to identify differentially expressed genes within each system. Subsequently, these genes were categorized into distinct clusters based on their relative expression levels (fold change = 2) in ESCs and MEFs. Finally, genes were ranked according to their highest expression levels (or lowest in the case of downregulated genes) at specific time points. The results were then presented. In comparing differential gene expression among the SALL4, OCT4, and O + S systems, "common up/specific up" indicates genes that were upregulated in the given reprogramming system and displayed no differential expression or downregulation in the other reprogramming systems.

## ATAC-seq and chromatin accessibility analysis

Sequencing data was mapped to the mm10 mouse genome assembly using Bowtie2[73] with the '--very-sensitive' option. Duplicated mapped reads were removed with Sambamba using '--overflow-list-size 600000,' and mitochondrial sequences were filtered out using the 'grep -v 'chrM' command. Subsequently, BigWig files were generated from BAM files using bedtools and bamCoverage, with RPKM normalization. Motif analysis was performed using HOMER2. Peak annotation utilized the Chipseeker package, with a focus on the 3 kb region around the transcription start site (TSS).

Chromatin Accessibility Analysis referred to previously established methods with some improvements[74]. Peaks were called with MACS2[75] using the '-f BEDPE --keep-dup all -g mm --nomodel --shift −100 −extsize 200' settings and cut-off by the default $q$-value of 0.05. Peaks meeting this threshold were considered 'open' peaks, while the rest were categorized as 'close' regions. An integrated background file containing all peaks from every sample was generated using bedtools. Each sample's peaks were then compared to this background file, and regions displaying transitions from 'close-open,' 'open-close,' or permanently open between the reprogramming starting point (MEFs) and the desired endpoint (ESCs) were selected. These regions were considered indicative of chromatin accessibility changes associated with pluripotency. 'CO' denoted chromatin accessibility regions that successfully opened during the reprogramming process, with 'CO1' indicating a region that was closed in MEFs and permanently open from Day 0 onwards. Similarly, 'OC'

represented regions that were successfully closed, and 'PO' indicated permanently open regions.

In the analysis of chromatin accessibility and motif identification within the S4/O4/O + S system across various time points (D0, D4, D7, and D10), as depicted in Figs. 3d and 5b, peak calling was performed with MACS2, employing the DsRed samples from corresponding time points as controls to delineate the impact of individual transcription factors on chromatin accessibility. Subsequent motif enrichment analysis of the identified peaks was conducted using the HOMER2 software suite.

### Cut&Tag analysis

Sequencing data was mapped to the mm10 mouse genome assembly using Bowtie2 with the following options: '--end-to-end --very-sensitive --no-mixed --no-discordant.' Duplicated PCR reads were removed using Sambamba with the option '--overflow-list-size 600000.' BigWig files were generated from BAM files using bedtools and bamCoverage, with normalization based on RPKM (Reads Per Kilobase per Million mapped reads). Peaks were called using MACS2 with the following options: '-t -c -g mm -f BAMPE -q 0.1 --keep-dup.' Motif analysis was performed using HOMER2 with its default parameters. Differential peaks analysis for different SALL4 mutant variants was performed using MACS2 bdgdiff. GO analysis was performed using Chipseeker with a focus on regions within 3 kb of the transcription start site (TSS).

### Direct and indirect effects, and predicted binding sites

In our analysis, we distinguish between the direct and indirect effects of SALL4 binding on chromatin accessibility. 'Direct effects' pertain to regions where SALL4 binding directly influences chromatin accessibility. This category includes 'Direct close,' wherein SALL4 binds to an initially open chromatin region at Day 0, resulting in the transition of ATAC peaks from open to close. On the other hand, 'Direct open' describes instances where SALL4 binds to closed regions, leading to a transition of ATAC peaks from close to open. In contrast, 'Indirect effects' encompass regions where SALL4 doesn't bind directly but still instigates changes in chromatin accessibility.

In addition, we examined the co-occupancy of SALL4 and OCT4 at specific binding sites. The 'SALL4 and OCT4 predicted sites' are regions where Cut&Tag binding sites in the SALL4 and OCT4 systems intersect, indicating the potential co-occupancy of SALL4 and OCT4 at these locations. 'Actual binding sites' were defined as regions present in both the SALL4-FLAG and OCT4-FLAG CutTag peaks within the O + S system.

### Statistics and reproducibility

No statistical method was used to predetermine the sample size. The experiments were not randomized. The investigators were not blinded to allocation during the experiment and outcome assessment. Data are presented as mean ± s.d. The $P$-Value was calculated using a two-tailed unpaired t-test or one-way ANOVA with GraphPad Prism 8. A $P$-Value < 0.05 was considered statistically. The statistical test and precise $P$-Value, the exact number of sample sizes, and independent experiments were indicated in the relevant figure legends. Data for RNA-seq was performed twice, data for scRNA-seq, ATAC-seq, and Cut&Tag was performed once, and data for IP-MS was performed three times in this study.

### Reporting summary

Further information on research design is available in the Nature Portfolio Reporting Summary linked to this article.

## Data availability

There are no restrictions on data availability. All relevant data used in this paper are available. The bulk RNA-seq, scRNA-seq, Cut&Tag, and ATAC-seq data presented in this study have been deposited in the Genome Sequence Archive at the National Genomics Data Center, China National Center for Bioinformation / Beijing Institute of Genomics, Chinese Academy of Sciences under Bioproject ID PRJCA017942. The mass spectrometry proteomics data have been deposited to the ProteomeXchange Consortium (https://proteomecentral.proteomexchange.org) via the iProX partner repository[76,77] with the dataset identifier PXD057516. Source data are provided in this paper.

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

## Acknowledgements
We thank the lab members in GIBH for their kind help. We thank the members of the Analytical Instrumentation Core and Laboratory Animal Core in GIBH for their help in data collection. We thank Dr. Austin Smith and Dr. Ge Guo for their helpful discussions and valuable insights on the manuscript. We thank Dr. Xiaofei Zhang for their assistance in IP-MS data collection and analysis. This research was supported by grants from the National Key Research and Development Program of China (2018YFE0204800 [J.L.]), National Natural Science Foundation of China (U20A2013 [T.W.], 32370791 [J.L.]), Guangdong Basic and Applied Basic Research Foundation (2020A1515110122 [L.W.]), Science and Technology Projects in Guangzhou, China (Grant No. (202201010510[Z.Z])), Science and Technology Planning Project of Guangdong Province (2023B1212060050 [J.L.], 2023B1212120009 [J.L.], 2022B1212010010 [Y.L. and P.Z]), Basic Research Project of Guangzhou Institutes of Biomedicine and Health, Chinese Academy of Sciences (GIBHBRP23-02[J.L.]), Health@InnoHK Program launched by the Innovation Technology Commission of the Hong Kong SAR, P.R. China, the Postdoctoral Fellowship Program (Grade C) of China Postdoctoral Science Foundation (No.GZC20232689[L.Z.].), and Grants from Guangdong Province (2024A1515013168 [B.L.], 2024ZDZX2055 [B.L.]).

## Author contributions
J.L. and L.X. designed the project, L.X. and Z.H. performed the experiments, Z.W., Z.Z., JX.L., L.L., and RX.L. analyzed the data. Y.Y. performed the blastocyst injection, L.X. and Z.H. performed the Cut&Tag, ATAC-seq, RNA-seq, and scRNA-seq experiments. R.L. isolated the OG2-MEFs cell lines. L.W. supervised the data analysis. B.L., L.W., and J.L. supervised the whole study. J.L., L.X., Z.H., and Z.W. wrote the manuscript with the critical suggestions from B.L., L.W., Z.Z., M.K., H.W., H.M., J.G., Y.L., L.Z., and P.Z. J.L.conceived the whole study, and approved the final version.

## Competing interests
The authors declare no competing interests.
