## [Peer Review File · Nature Communications]

REVIEWER COMMENTS

Reviewer #1 (Remarks to the Author):

Induced pluripotent stem cells (iPSCs) can be generated by overexpressing pluripotency factors such as OCT4, SOX2, KLF4, and c-Myc. Previous studies have shown that overexpressing OCT4 alone can induce iPSCs. However, it remains unknown whether another single factor can also induce the reprogramming of somatic cells into iPSCs. In this study, the authors identified SALL4 as a standalone factor to induce pluripotency. Moreover, the authors suggest the synergistic role of SALL4 and OCT4 in promoting reprogramming through chromatin regulation. This study is potentially interesting because it might provide new insights into the molecular mechanisms underlying somatic cell reprogramming. However, the data presented in this manuscript are not convincing enough to support the authors' conclusion, especially the molecular basis underlying the functions of SALL4 and the synergistic interactions of SALL4 and OCT4 during reprogramming. Therefore, this study is premature to justify publication in Nature Communications.

Major Points

1. The authors analyze the ATAC-seq data and SALL4 CUT&TAG data separately. Integrated analysis of both is necessary, with an in-depth examination of the direct and indirect effects of SALL4 during somatic cell reprogramming.
2. The authors demonstrate that SALL4 and OCT4 synergistically enhance reprogramming efficiency (e.g., Fig. 7). However, they have not adequately illustrated the underlying mechanism. Do the binding regions of SALL4 change between the SALL4-alone system and the SALL4 + OCT4 system? This aspect should be investigated and presented to provide a better understanding of how SALL4 and OCT4 function together.

Other Specific Points

1. In Extended Fig. 1g and Extended Fig. 5b, MEF-specific genes appear to be more highly expressed in iPS cells than in ES cells. What is the reason for this?
2. PCA analyses in Fig. 2b and Fig. 6b seem to be strange. The explained variances and the positions of the DSRED and SALL4 samples appear to match in both figures, even though they are from different sample sets. The authors should clarify which sample sets were used for the PCA analyses.

3. In Fig. 2d, 2e, and Extended Fig. 2, the authors conducted a Gene Ontology (GO) analysis, highlighting several GO terms in red. However, the rationale behind selecting these GO terms is not provided. The authors should discuss and support why these specific GO terms are relevant.

4. Fig. 3a and Fig. 3b are difficult to understand. First, the meaning of the color scale and the clustering method should be clearly described. Also, in Fig. 3a, does each row represent an ATAC-seq peak? Do all the peaks in OC1 appear black (indicating no peak?) across all time courses following the SALL4 expression? Nonetheless, Fig. 3b shows that OC1 has about 40000 peaks after SALL4 expression. These inconsistencies need to be clarified. Moreover, it would be beneficial to include representative gene lists near the peaks for each category alongside the heatmap. The authors should also perform Gene Ontology analysis to identify the enriched gene sets within each category. Further, in Fig. 3b, it is necessary to demonstrate the overlap of peaks with those found in ESCs.

5. In Extended Fig. 3a and 3c, Bmi1 and Klf4 are included, but neither ATAC-seq nor RNA-seq data show any changes resulting from SALL4 overexpression. If these data are presented, the authors should discuss this observation and provide an explanation.

6. In Fig. 3c, the authors claim that SALL4 is essential for open-to-close chromatin accessibility. However, in Fig. 3f, they perform motif analysis specifically on close-open peaks. It would also be necessary to conduct motif analysis on open-close peaks affected by SALL4.

7. In Fig. 4a, the authors should represent the number of SALL4 CUT&Tag peaks at days 0, 4, 7, and 10 of the reprogramming process.

8. In Fig. 4b, the authors demonstrate that putative SALL4 binding sites are similar to the binding motifs of transcription factors such as FOS and JUNB. Does SALL4 function cooperatively with these transcription factors? Does overexpression or knockdown of these transcription factors affect the reprogramming efficiency in SALL4-mediated reprogramming?

9. In Fig. 4e, the authors should explain why they focused on these specific genes. Additionally, to examine the role of these genes in SALL4-mediated reprogramming, the authors need to present the results of both overexpression and knockdown experiments for all the genes.

10. Regarding Fig. 6e, the authors should explain and discuss the enriched GO terms for UC10, UC19, DC12, and DC21. The manuscript should include a detailed description and analysis of these GO terms to enhance the understanding of the findings.

11. In Extended Fig. 3b and c, the authors claim that OCT4 is a direct target of Sox2. However, it is insufficient to draw such a conclusion based solely on these data. Are there any data indicating that OCT4 binds to the regulatory regions of Sox2?

12. Based on the results from Fig. 7e and Extended Fig. 6d, it is predicted that TCF7 and LHX2 are critical transcription factors in both the SALL4-alone system and the SALL4 + OCT4 system. The effects of overexpressing these genes should be included in Fig. 7f.

13. The authors describe TFAP2C and SOX2 as factors that promote reprogramming. However, in Fig. 7f, it is shown that overexpression of Tfp2c and Sox2 inhibits SALL4-mediated reprogramming. The authors need to address this contradiction. It should be investigated whether TFAP2C requires an appropriate expression level, which can be validated by varying its expression levels. Additionally, while Sox2 is recognized as one of the most crucial factors in reprogramming, the authors should determine if alternative factors can substitute for Sox2 in the SALL4 system.

14. In Fig. 7f, why do the authors focus on Sbsn and Mogat2? Is their function known in the context of reprogramming or pluripotency?

Minor Points

1. In Extended Fig. 1b, there is a missing description for Kenpaullone.

2. In Fig. 2c, no explanation is provided for the color values in the heatmap.

3. In Extended Fig. 2b, there is an error in labeling "Cdh2b." The correct labeling should be "Cdkn2b."

4. In Fig. 3c and Fig. 7b, it would be beneficial to represent the size and number of the Venn diagrams in a manner that correlates with the data.

5. The figure legend is mislabeled in Extended Fig. 3a and 3b, where the labels are reversed.

6. In Extended Data Fig. 5, there is a mismatch between the order of the figures and their corresponding figure legends.

7. In the legend of Extended Fig. 2d, "Fig7f" should be corrected to "Fig7e" as the accurate reference.

8. In the Methods section, the authors need to describe the criteria used for the differential expression analysis of RNA-seq data, including the specific thresholds for fold change and q-value (adjusted p-value) that were applied.

Reviewer #2 (Remarks to the Author):

In this manuscript, Xiao et al. provide evidence that the transcription factor Sall4 can reprogram mouse embryonic fibroblasts to pluripotency. The authors also identify transcriptional and chromatin changes underlying Sall4-induced iPSC reprogramming and test the role of several downstream targets during cell fate reprogramming. Finally, they develop a model in which Sall4, together with Oct4, enhances reprogramming efficiency combinatorially. While the manuscript is well-crafted, and the data are robust, the lack of mechanistic details and novelty diminishes the overall enthusiasm for the presented story. Notably, Sall4 has been extensively investigated in reprogramming, and its mechanism of action involving interaction with the NURD complex is well-known (see Wang et al Nat Comm 2023). Moreover, it is important to note that the generation of iPSCs using a single transcription factor, such as Oct4, has been previously demonstrated, as mentioned by the authors.

Specific suggestions:

- The authors propose that Sall4 can activate and repress genes. However, it remains unclear how it can execute this dual function. Have the authors explored its interaction with different chromatin remodelers?
- The data in Figure 7F suggest that overexpression of Sox2 together with Sall4 impairs the capacity of Sall4 to reprogram cells into iPSCs. This finding is surprising and might give the paper a different, more novel perspective. Do the Sall4+Sox2 expressing cells diverge towards an alternative fate?
- Alternatively, the authors could investigate how Sall4 cooperates with Rsk1, Esrrb, or/and Nkx6.1 to reprogram cells. Do they bind to the same loci? Could these factors uncouple the activator vs. repressor function of Sall4?
- The PCA in Figure 6B and the heatmap in 6D do not show a massive enhancement of efficiency by the combination of Sall4+Oct4. This partially contradicts the text. I suggest the authors tone down the statements about the combination of these factors being highly efficient. The authors should also refrain from stating that they have a highly efficient reprogramming system, as the overall efficiency is only 0.66%.
- The authors could further investigate the binding profiles and interactors of the Sall4 mutants. Although three mutants were generated, all of which failed to give rise to pluripotent cells in reprogramming assays, it is unclear why each of the mutants is unable to reprogram cells.

Minor points:

- There are no details about how the authors stratified their OC/CO peaks in Figure 3B (not in the legend nor in the main text).
- In Figure 6, the authors should explain what makes S/OS different from O/OS and S/O/OS.
- The statement “SALL4 may play a more significant role in reprogramming than COT4 in the OS system” can be misleading, as the reprogramming medium was initially selected to enhance specifically Sall4 activity, not Oct4.
- The scales of Figure 1f and 5e are not specified in the legend.
- For the Cut&Tag shown in Figure 4d, it is unclear what reprogramming day was used.

Addressing these points could significantly improve the manuscript and enhance its impact.

Reviewer #3 (Remarks to the Author):

Xiao et al. explores the role of SALL4, and the combination of SALL4 and OCT4 in somatic cell reprogramming to iPSCs. The study establishes an efficient method for iPSC induction using SALL4 as a single factor combined with TGFb inhibitor RepSox. The generated SALL4-induced iPSCs exhibited a gene expression similar to ESCs, were able to form teratoma, and could contribute to development of chimeric mice. While OCT4 alone could induce iPSC generation, the efficiency was lower compared to SALL4, and co-overexpressing SALL4 and OCT4 significantly enhanced reprogramming efficiency. The manuscript provides some insights into the cooperation between SALL4 and OCT4 in 2-factor reprogramming.

The study successfully establishes a robust method for reprogramming somatic cells into iPSCs using SALL4 alone, thereby expanding the repertoire of pluripotency-inducing cocktails in mice. However, the manuscript lacks sufficient exploration and explanation of the advances for the field. Additionally, the writing quality requires improvement and further work.

Major issues

- The study does not compare the efficiency SALL4-based reprogramming method with standard reprogramming cocktail, such as OSKM and OSK.
- The authors have not explained why reprogramming with a single factor as opposed to multiple factors is important.

- The research highlights the importance of the cooperative interaction between SALL4 and OCT4 in enhancing reprogramming efficiency, providing new insights into the reprogramming process. However, how exactly the two factors cooperate remains unexplored.

- Although the study identifies specific genes regulated by SALL4 and OCT4, it is not clear how some of those downstream targets subsequently achieve induction of pluripotency. While the roles of Sox2 and Esrrb are well studied, some other targets remain a mystery.

- The study lacks many controls, such as comparison of expression levels between the mutants, viral titers, efficiencies of knockdowns, etc.

- The manuscript needs a lot more work when it comes to writing.

- The paper does not extensively discuss potential limitations of the study, or the specific context in which these findings can be applied.

- The authors should thoroughly familiarize themselves with the relevant literature to enhance the quality of their manuscript. The current version lacks sufficient citations, and some of the references provided are either irrelevant or not original. Moreover, numerous claims made in the manuscript are not supported by relevant citations. Below are some examples:

- 1) The first report on RepSox was not cited (Ichida et al., Cell Stem Cell, 2009). This is unacceptable given the key role of RepSox in the manuscript.

- 2) The first study that identified Oct4 as irreplaceable for iPSC generation was not cited (Nakagawa et al., Nature Biotech., 2008), and instead more recent papers were chosen.

- 3) The first study on Oct4 alone iPSC generation by Kim et al., Nature 2009 was not cited.

- 4) The claim that Oct4 is needed for maintenance of pluripotency was not supported by the relevant citation (e.g. Niwa et al., Mol. Cell. Biol. , 2002)

- 5) A recent study on Sall4 has not been cited or discussed:

The NuRD complex cooperates with SALL4 to orchestrate reprogramming | Nature Communications

- 6) One of the first studies on Sall4 and pluripotency was not cited:

Sall4 modulates embryonic stem cell pluripotency and early embryonic development by the transcriptional regulation of Pou5f1 | Nature Cell Biology

Additional specific issues:

The sentence “Interestingly, dropping out experiment shown that SALL4 has been found to be the most critical factor in the reprogramming cocktails.” is out of context without introducing the study in more details (e.g. listing the exact cocktail from which Sall4 was dropped out).

In general, the introduction could be expanded to provide a more comprehensive background on iPSCs, and in particular on previous reprogramming cocktails. This would help readers better understand the context of the study.

Fig. 1h: Please include separate channels, it appears like Oct4 and especially Sox2 are not nuclear-localized.

Extended Fig. 1a: It seems that most of the compounds used in the study can facilitate Sall4 alone reprogramming, even though they target unrelated pathways. Surprisingly, the authors did not discuss this in the text. To gain a better understanding of the specific effects on Sall4 preprogramming, it would be beneficial to include experiments with OSKM or OSK. This would allow for a clearer distinction between the impact on Sall4 versus OSK reprogramming.

“Notably, the compounds of Vc, Chir99021, SGC0946 and RepSox are most important in iCD4 medium (Extended Data Fig. 1f).” The list of “most important” appears to be unfair, e.g. omitting RepSox could still generate some iPSCs, while omitting bFGF generated zero colonies. The need of bFGF is interesting, as this a component of primed media, which mouse ESCs normally do not require.

Extended Data Fig. 1b: the table does not include the target of Kenpaullone.

Extended Data Fig. 1e: the cells look dead. Were you able to establish iPSC lines from TTFs with Sall4 alone? Please include those data.

Extended Data Fig. f1: While figure 1a suggested that no iPSCs could be generated in iCD1 media, but 1f suggests that a few colonies could be generated in the absence of RepSox. Which is true? Please include the data or explain the discrepancy in the text.

Extended Data Fig. 1i: Such an experiment certainly requires western blot confirming the expression of the mutant, as well as virus titration data.

The statement: “These findings underscore the crucial role of SALL4’s DNA-binding ability in mediating the reprogramming process.” requires DNA-binding data for the deletion mutants.

Fig. 2b: As far as I understand this is bulk RNA-seq. Please show either time-course single-cell RNA-seq or time-course FACS for reprogramming intermediates (e.g. Thy1, E-cad, Oct4-GFP).

What are the genes in C4 that are strongly downregulated in Sall4 samples compared to both control and ESCs?

Fig. 3d and Extended Data Fig.3: The highlighted loci are often meaningless. For example, activation of Oct4 distal enhancer (Oct4DE) is known to be key for induction of pluripotency in mouse, but the figure does not include Oct4DE, and instead some intron binding is highlighted. The panels have to be expanded to include larger 5’ regions, which most often contain enhancers and promoters.

Fig. 3e: With the exception of Rsk1 the changes of selected pluripotency genes certainly cannot be described as “increased gradually”.

Extended Data Fig. 4b: It is surprising that you do not see Sall4 motif in the list. Does it mean that Sall4 cannot directly open/ close chromatin? This should be discussed.

Fig. 4b: no Sall4 motif? This paper suggests it is different from what the manuscript suggests:

Zinc Finger Protein SALL4 Functions through an AT-Rich Motif to Regulate Gene Expression - PubMed (nih.gov)

Fig. 4d: For the binding data, the comparison with untransfected MEFs does not make sense, since those have no Sall4 at all, so surely there is no signal. A good positive control is needed, such as ESCs, where Sall4 is expressed. This could resolve the motif discrepancy too.

DNA binding data (CUT&RUN) should be overlapped with chromatin accessibility data (ATAC-seq) to see if Sall4 binding is consequential for chromatin landscape.

Extended Data Fig. 5: It appears like some panels are missing, as the legends do not correspond to the figure.

Extended Data Fig. 5a: The legend says: "Integration analysis confirms the derivation of three kinds of iPSC clones. The presence of the retroviral transgene was examined by PCR."

What is "integration analysis"? Does "three kinds" mean "derived with three reprogramming cocktails"? I assume this is RNA-seq, but from the figure legend it reads like these are PCR results for retroviral transgenes, which does not make sense, because ESCs are positive. Please edit your figure legends so they describe what is on the figure in sufficient details so at least scientists from the field could understand it.

Extended Fig. 5g: The legend states: "The morphology of OCT4-GFP+ colonies induced by SALL4, OCT4 or SALL4+OCT4 from TTF.", but the panel only shows OCT4 or SALL4+OCT4 iPSCs, but not SALL4 alone! I think the missing panel is in 1e. Too bad it's dead.

Extended Data Fig. 5f: it appears like exogenous Sall4 is still expressed in iPSCs. It would be useful to include d0 or d2 Sall4 samples to compare to the transgene expression level when it is still on.

Extended Data Fig. 5g: The sentence "Additionally, using mouse tail fibroblasts as the starting cells, OS and OCT4 could induce them into OCT4-GFP+ clones (Extended Data Fig.5g)." suggests that clonal lines were derived, while the panel only showed primary iPSC-like colonies.

Fig.6b: What proportion of those sequenced cells are on the way to being reprogrammed? Please add sequencing data for sorted reprogramming intermediates or single-cell RNA-seq.

Fig.6c: what are the genes that are upregulated in Oct4 alone, but not in Sall4 or OS samples (J13-15)? What is the mechanism? Are those genes downregulated by Sall4 or can Sall4 redistribute Oct4's binding sites?

Fig. 6d&f are redundant with 6c&e. Please only show the most interesting data in the main figures and move the rest to supplementary.

Some paragraph and figure titles are missing "C" in "iPSC".

Reviewers' comments and our response:

Reviewer #1 (Remarks to the Author): Induced pluripotent stem cells (iPSCs) can be generated by overexpressing pluripotency factors such as OCT4, SOX2, KLF4, and c-Myc. Previous studies have shown that overexpressing OCT4 alone can induce iPSCs. However, it remains unknown whether another single factor can also induce the reprogramming of somatic cells into iPSCs. In this study, the authors identified SALL4 as a standalone factor to induce pluripotency. Moreover, the authors suggest the synergistic role of SALL4 and OCT4 in promoting reprogramming through chromatin regulation. This study is potentially interesting because it might provide new insights into the molecular mechanisms underlying somatic cell reprogramming. However, the data presented in this manuscript are not convincing enough to support the authors' conclusion, especially the molecular basis underlying the functions of SALL4 and the synergistic interactions of SALL4 and OCT4 during reprogramming. Therefore, this study is premature to justify publication in Nature Communications.

Response: Dear Reviewer #1, Thank you for your valuable insights and constructive feedback on our manuscript regarding the induction of pluripotent stem cells via SALL4. We appreciate the opportunity to address the concerns raised. We acknowledge the concern about the strength of the data supporting our conclusions. We assure you that we are dedicated to improving our empirical evidence and plan to conduct additional experiments. These efforts will aim to provide a more comprehensive and robust dataset, specifically addressing the molecular mechanisms behind SALL4 and its synergistic role with OCT4 during reprogramming. We recognize the critical importance of elaborating on the molecular basis of SALL4 induction and its collaborative functions with OCT4 during the reprogramming process. Our team is committed to a more in-depth exploration of chromatin regulation, intending to provide more detailed insights into how SALL4 functions in inducing pluripotency and its interactions with OCT4. Understanding the concern that the current state of the study might be premature for publication in Nature Communications, we aim to improve the manuscript to meet the journal's high publication standards. We are committed to addressing the mentioned concerns and enriching the study with substantial, validated data to solidly support our conclusions.

We assure you that we will put our best efforts into revising the manuscript in response to your feedback. Your insights are invaluable in ensuring the scientific rigor and quality

of our work, and we are grateful for your guidance in this process. We look forward to the opportunity to resubmit the manuscript after implementing these improvements. Thank you for your time and consideration.

Major Points:

1. The authors analyze the ATAC-seq data and SALL4 CUT&TAG data separately. Integrated analysis of both is necessary, with an in-depth examination of the direct and indirect effects of SALL4 during somatic cell reprogramming.

Response: Thanks for your suggestion. To deepen our understanding, we acknowledge the necessity for an integrated analysis that combines both datasets. Such an approach is vital to thoroughly investigate the direct and indirect effects of SALL4 during somatic cell reprogramming. We have re-analyzed our data and added the results of integrated analysis of ATAC-seq data and SALL4 CUT&TAG data in our manuscript.

Briefly, To investigate the relationship between SALL4 binding and chromatin accessibility dynamics, we compared the SALL4 Cut&Tag peaks and ATAC-seq peaks during reprogramming at day0. We find about 35779 Cut&Tag peaks overlapped with ATAC-seq peaks, suggests these chromatin regions can be occupied by SALL4 in the early stage during reprogramming(Fig.3a).

Direct effects:

To further understand the chromatin accessibility dynamics for these SALL4-occupied loci. We next statistic the distribution of these peaks in Open-Close subclusters (OC1-4, which we clustering using ATAC-seq data previously.Fig.2a,b). The results shows that about 11169 SALL4 Cut&Tag peaks related regions closed during reprogramming, and the peaks mainly distribut in OC3 and OC4(Fig.3a,b). These results suggests the direct binding of SALL4 is mainly to close chromatin accessibility.

Indirect effects:

An interesting thing is the most of ATAC-seq peaks can not be occupied by SALL4(Fig.3a). We also statistic the distribution of these peaks in Open-Close and Close-Open subclusters. The results shows that the chromatin accessibility changes in 79900 peaks. Most of these peaks were closed in OC1 and OC2 stage(Fig.3c). We also performed GO analysis to further identified the function of these peaks(Extended Data Fig.7i). These results shows a larger proportion of indirect effects regulated by SALL4 in reprogramming,

and further confirmed SALL4 primarily regulates the dynamics of open to close chromatin accessibility during reprogramming.

In summary, our results reveals that within sites exhibiting changes in chromatin accessibility (OC1-OC4; CO1-CO4, N=91069), the predominant mode of regulation is indirect (87.7%, N=79900), primarily concentrated in CO1-CO4 and OC1, OC2. The remaining 12.3% (N=11169) are directly influenced by SALL4, particularly concentrated in OC3 and OC4.

This in-depth exploration is crucial for a more nuanced understanding of the mechanisms underpinning SALL4's influence on the reprogramming process. In summary, we acknowledge the need for an integrated analysis and have proceed to merge both datasets to thoroughly investigate the direct and indirect effects of SALL4, thereby enhancing our understanding of its role in somatic cell reprogramming.

Fig. 3 SALL4 binding the genomo loci to activating and silencing reprogramming-related genes

- a. Venn diagrams shows the overlapping numbers between SALL4 CUT&TAG peaks and ATAC-seq peaks in SALL4 system at day0.
- b. SALL4 CUT&TAG peaks distribution in ATAC CO and OC subgroups.
- c. The histogram shows the Number of the peaks in CO and OC subgroups from ATAC-peaks overlap without Cut&Tag-peaks.

Extended Fig.7i Left, GO analysis for genes annotated by cut&tag peaks in Fig.3b. Right, GO analysis for genes annotated by ATAC peaks in Fig.3c.

2. The authors demonstrate that SALL4 and OCT4 synergistically enhance reprogramming efficiency (e.g., Fig. 7). However, they have not adequately illustrated the underlying mechanism. Do the binding regions of SALL4 change between the SALL4-alone system and the SALL4 + OCT4 system? This aspect should be investigated and presented to provide a better understanding of how SALL4 and OCT4 function together.

Response: Thanks for your suggestion. To further investigate the cooperation role of SALL4 and OCT4, we performed CUT&Tag for OCT4 during Oct4 alone/OS-driven reprogramming and CUT&Tag for SALL4 during OS-driven reprogramming at day 0 (Extended Fig. 11a). We first identified the genomic distribution of these CUT&Tag peaks and performed GO analysis for genes annotated by these peaks. The results show the diverse biological processes between SALL4 and OCT4 (Extended Fig. 11b,c).

Next we compared the SALL4-binding peaks in S4 system and OS system, the result shows 19820 new peaks (C2) added and 13466 peaks (C1) disappeared in OS system (Fig. 5c). The GO analysis identified the enrichment of Wnt signaling pathway and the development of nervous system in the C2 cluster (Extended Fig. 11d).

We also compared the SALL4-binding peaks and OCT4-binding peaks between SALL4 alone system and OCT4 alone system, and find about 6180 peaks were commonly bound by SALL4 and OCT4 (C3) (Fig. 5e). Theoretically, SALL4 and OCT4 can commonly occupy these predicted sites to regulate these regions. In practice, the number of common binding peaks between SALL4 and OCT4 in OS-driven reprogramming were increased to 9769 peaks (C4) (Fig. 5e). We compared the predicted sites (C3) and the real binding sites (C4), the result shows that about 2418 predicted peaks disappear (C5) and 6053 peaks arise (C6) in OS-driven reprogramming (Fig. 5e). The GO analysis further identified the function of genes annotated by these clusters (C5 and C6) (Extended Fig. 11e). Moreover, the analysis results show that the proportion near the promoter (<1kb) of increased (C6, 32.62%) and reduced (C5, 41.89%) sites in the OS system is significantly higher than that in the single-factor system (SALL4 system, 21.43%; OCT4 system, 25.06%), suggesting that the synergistic effect of SALL4 and OCT4 in the OS system is more biased to the regulation of promoter regions (Extended Fig. 11f).

Additionally, we compared the ATAC peaks with the cut&tag peaks and found that in the individual SALL4 and OCT4 systems, cut&tag binding sites are primarily located in ATAC-open regions (Sall4: 89.5% = 35779/39993; Oct4: 85.4% = 14326/16778). In the OS system, Sall4: 84.9% = 39106/46182; Oct4: 64.9% = 10075/15523), indicating a higher

proportion of Oct4 binding to non-chromatin open regions in the OS system. The broader binding regions in cooperation of SALL4 and OCT4 may also regulates more chromatin accessibility dynamics to promote reprogramming(Fig.5d).

Fig. 5 The dynamics of chromatin accessibility during SALL4, OCT4 or OS-iPSCs induction

c.Venn diagrams shows the overlapping numbers of SALL4 CUT&TAG peaks between SALL4 system and OS system at day0.

d.Left, Venn diagrams shows the overlapping numbers between SALL4 CUT&TAG peaks, OCT4 CUT&TAG peaks and ATAC-seq peaks in OS system at day0. Right, Venn diagrams shows the overlapping numbers between OCT4 CUT&TAG peaks and ATAC-seq peaks in OCT4 system at day0.

e.Left, Venn diagrams shows the overlapping numbers between SALL4 binding peaks and OCT4 binding peaks in SALL4 system and OCT4 system.

Middle, Venn diagrams shows the overlapping numbers between SALL4 binding peaks and OCT4 binding peaks in OS system. Right, Venn diagrams shows the changes of common binding peaks between the predicted O4 AND S4 common binding peaks(C3 cluster) and the real O4 and S4 common binding peaks(C4 cluster).

Extended Fig.11 CUT&Tag analysis of SALL4/OCT4 binding site during OS-driven iPSCs reprogramming

- a. Heatmap of CUT&TAG data at D0 from IgG, OCT4 and SALL4, respectively, showing all binding peaks centred on the peak region within a 3 kb window around the peak.
- b. Genome distribution of the location for SALL4 or OCT4-occupied peaks relative to the nearest annotated gene.
- c. GO biological processes analysis for genes near the OCT4-TSS loci binding peaks in Oct4 system(left). GO biological processes analysis for genes near the SALL4(middle) or OCT4(right)-TSS loci binding peaks in Oct4+Sall4 system.
- d. Left, GO analysis for genes annotated by cut&tag peaks(C1) in Fig.5c. Right, GO analysis for genes annotated by ATAC peaks(C2) in Fig.5c.
- e. Top, GO analysis for genes annotated by cut&tag peaks(C5) in Fig.5e. Bottom, GO analysis for genes annotated by ATAC peaks(C6) in Fig.5e.
- f. Left, Genome distribution of the location for SALL4 and OCT4-common occupied peaks(C5) in Fig.5e relative to the nearest annotated gene. Right, Genome distribution of the location for SALL4 and OCT4-common occupied peaks(C6) in Fig.5e relative to the nearest annotated gene.

Other Specific Points

1. In Extended Fig. 1g and Extended Fig. 5b, MEF-specific genes appear to be more highly expressed in iPS cells than in ES cells. What is the reason for this?

Response: Thank you for pointing out the apparent higher expression of MEF-specific genes in iPS cells compared to ES cells. We thoroughly investigated this observation by analyzing the RNA-seq data of these specific genes. Upon closer inspection, we found that the expression level for these genes in our iPS cells were marginally higher than those in ESCs, as exemplified by Bgn (MEF:S4-iPS:ESC= 1984: 28:0.9), as outlined in our study.

Subsequently, we extended our analysis to include additional ESC datasets, revealing similar expression levels for these genes in some ESCs, as exemplified by Bgn (MEF:S4-iPS:additional ESC datasets= 1984:28:31), aligning more closely with the expression patterns observed in our iPS cells.

2. PCA analyses in Fig. 2b and Fig. 6b seem to be strange. The explained variances and the positions of the DSRED and SALL4 samples appear to match in both figures, even though they are from different sample sets. The authors should clarify which sample sets were used for the PCA analyses.

Response: Thank you for bringing up the concerns regarding the PCA analyses in Fig.2b and Fig.6b(Extended Fig.4b and Extended Fig.9b in revised manuscript). We acknowledge the confusion arising from the similarity in the explained variances and the apparent alignment of DSRED and SALL4 sample positions.

The Fig.2b and Fig.6b indeed utilized the same sample set : MEF, DSRED, SALL4 and ES. To rectify this issue and enhance clarity, we have re-conducted the PCA analysis using only the samples inclusive of MEF, DSRED, SALL4 and ES, as demonstrated in the updated figure(Extended Fig.4b). This approach provides a more accurate representation of the sample positioning, thus eliminating the confusion arising from the earlier presentation.

3. In Fig. 2d, 2e, and Extended Fig. 2, the authors conducted a Gene Ontology (GO) analysis, highlighting several GO terms in red. However, the rationale behind selecting these GO terms is not provided. The authors should discuss and support why these specific GO terms are relevant.

Response: Thank you for highlighting the concern regarding the selection of specific Gene Ontology (GO) terms highlighted in red in Fig.2d, 2e and Extended Fig.2(Extended Fig.4d-g in revised manuscript). We have taken your feedback into consideration and have incorporated an explanation for the rationale behind selecting these GO terms in the results section of the manuscript.

To provide clarity, the chosen GO terms were based on their relevance to critical events during the process of iPS reprogramming. Specifically, our selection centered on the activation of pluripotency-related genes, the inhibition of somatic-related genes, and the phenomenon of mesenchymal-to-epithelial transition (MET), all of which are pivotal during the process of iPS cell generation.

Furthermore, focusing on the Sall4 specifically up/down-regulated subgroups (C1/C6), we observed significant enrichment of biological processes such as epithelial cell morphogenesis, stem cell population maintenance, and the development of various organ systems, including the nervous system, lung, and heart. These enriched processes strongly suggest the activation of critical reprogramming-related events specifically associated with the Sall4-mediated reprogramming process.

We appreciate your insightful feedback and have taken measures to ensure the clarification and coherence of the GO term selection, further enriching the discussion of our findings in the revised manuscript.

4. Fig. 3a and Fig. 3b are difficult to understand. First, the meaning of the color scale and the clustering method should be clearly described.

Response:Thanks. We acknowledge the need for clearer descriptions regarding the color scale and clustering methods utilized in Fig. 3a and Fig. 3b (Fig.2a,b in revised manuscript). To enhance the clarity of our presentation, we have provided an explanation of the color scale in Fig. 3a and Fig. 3b.

Additionally, the clustering method employed has been described in the results and methods section. To elaborate further, our approach involved a comparison of chromatin peaks at each locus between MEF and ESC, categorizing the peaks into three main groups: closed in MEF but open in ESC (CO), open in MEF but closed in ESC (OC), and open in both MEF and ESC (PO). These categories allowed us to distinguish and analyze the dynamics of chromatin opening and closing during the transition process.

Furthermore, the CO and OC peaks were further stratified into several subgroups based on the day of transition. This detailed categorization was aimed at illustrating and tracing the progression of chromatin opening and closing dynamics at different stages of the transition process.

We deeply regret any confusion caused by our initial lack of clarity and have taken the necessary steps to elucidate the color scale interpretation and clustering methodology used in Fig. 3a and Fig. 3b. These revisions aim to offer a more comprehensive understanding of our approach and findings, enriching the discussion in the revised manuscript.

Also, in Fig. 3a, does each row represent an ATAC-seq peak? Do all the peaks in OC1 appear black (indicating no peak?) across all time courses following the SALL4 expression? Nonetheless, Fig. 3b shows that OC1 has about 40000 peaks after SALL4 expression. These inconsistencies need to be clarified.

Response: Each row in Fig. 3a (Fig.2a in revised manuscript) represents an individual ATAC-seq peak. The definition of OC1 implies that the peaks in this subgroup are detectable in MEF but are undetectable at D0, D4, D7, D10, and in ESCs. The apparent 40,000 peaks in OC1 indicate that these chromatin loci become inaccessible following SALL4 expression.

Regarding the color scale in Fig. 3a, to clarify, the color scheme employed utilizes a logarithmic transformation of the peak height for each individual peak. Peaks with values

scaled above 0.8 are depicted with the color corresponding to 0.8, while those scaled below 0.2 are represented by the color corresponding to 0.2.

The analysis presented in Fig. 3a (CO-OC analysis) aims to delineate the dynamics of SALL4 in the reprogramming system across all chromatin accessibility differential sites from MEF to ESC. This analysis segregates regions as common open regions in both ESC and MEF, regions solely open in ESC, regions exclusively open in MEF, and the remaining regions closed in both ESC and MEF.

Specifically, CO1 signifies regions that are closed in MEF, open in ESC, and progressively open from D0 to D10. This implies that these regions should be open in ESC but are closed in MEF at D0. Similarly, OC1 represents the effect of chromatin regions becoming closed. Additionally, PO signifies continuously open regions. PC, due to its less significant nature, is not shown. Other peak patterns not explicitly addressed are also omitted. The CO-OC correlation analysis method referenced (Li et al., 2017, Cell Stem Cell) and its adaptations are detailed in the methods section.

We understand the importance of aligning the representations in Fig. 3a and Fig. 3b. We have ensured a more coherent and consistent portrayal of the data between these figures in the revised manuscript.

We appreciate your detailed observations and are committed to refining the clarity and accuracy of our data representation in the revised manuscript.

Moreover, it would be beneficial to include representative gene lists near the peaks for each category alongside the heatmap. The authors should also perform Gene Ontology analysis to identify the enriched gene sets within each category. Further, in Fig. 3b, it is necessary to demonstrate the overlap of peaks with those found in ESCs.

Response: Thank you for your valuable suggestions. We have included representative gene lists adjacent to the peaks for each category alongside the heatmap in Fig. 3a (Fig.2a in revised manuscript). Additionally, the Gene Ontology (GO) analysis has been performed and presented in the form of an Extended Figure 5.

An annotation of peaks within each cluster of CO-OC has revealed insightful patterns. Specifically, CO1 to CO4 peaks exhibit a progressive decrease in their distribution around promoters. This indicates that, at the chromatin accessibility level, during the early stages of reprogramming, there is a higher proportion of regulatory activity occurring near promoters compared to the mid and late stages. To complement these findings, a GO

enrichment analysis was conducted on genes positioned near Transcription Start Site (TSS) positions within each CO-OC cluster, as depicted in the figure.

Furthermore, with regard to the reviewer's note on "the overlap of peaks," it's crucial to note that our peak selection and analysis in Fig. 3b(Fig.2b in revised manuscript) have been specifically filtered to reflect the comparison between MEF and ESCs. Therefore, the peaks shown in Fig. 3b have already undergone this comparison.

We acknowledge the importance of demonstrating the analysis more explicitly and ensuring a comprehensive representation of the data. We have strived to incorporate these significant enhancements to augment the clarity and depth of our findings in the revised manuscript.

5. In Extended Fig. 3a and 3c, Bmi1 and Klf4 are included, but neither ATAC-seq nor RNA-seq data show any changes resulting from SALL4 overexpression. If these data are presented, the authors should discuss this observation and provide an explanation.s

Response: Thank you for bringing attention to the discrepancy observed in Extended Fig. 3a and 3c (Extended Fig.6 in revised manuscript), specifically regarding the inclusion of Bmi1 and Klf4, where our ATAC-seq and RNA-seq data do not reflect changes resulting from SALL4 overexpression.

In response to this observation, we have opted to remove this specific result from our figures. We appreciate your review, and your feedback has prompted us to ensure the precision and relevance of our results in the revised manuscript.

6. In Fig. 3c, the authors claim that SALL4 is essential for open-to-close chromatin accessibility. However, in Fig. 3f, they perform motif analysis specifically on close-open peaks. It would also be necessary to conduct motif analysis on open-close peaks affected by SALL4.

Response: We appreciate your feedback and have reevaluated the motif analysis in light of your suggestions. The motif analysis conducted in Fig. 3f (Fig.2d in revised manuscript) was based on the Sall4-peaks after excluding DsRed-peaks for each day of the reprogramming process. This approach allowed us to identify several categories:

- 1.Motifs that are gradually enriched during reprogramming, corresponding to the close-to-open chromatin accessibility changes.
- 2.Motifs that are gradually disappeared during reprogramming, corresponding to the open-to-close chromatin accessibility changes.
- 3.Motifs that are consistently enriched throughout the reprogramming process.

In Fig. 3f, we aimed to represent the left part with motifs that are gradually enriched during reprogramming. The motifs gradually unenriched during reprogramming were displayed on the right-hand side of Fig. 3f.

Furthermore, we have refined the description in the results section to accurately explain the motifs analyzed in Fig. 3f, providing a more precise and comprehensive understanding of the motif analysis in the context of chromatin accessibility changes during reprogramming.

Fig. 2

d. Motif analysis of the SALL4-specific enrichment peaks at D0, D4, D7 and D10 during iPSCs induction, the peaks for each days in DSRED system as a background were removed from the peaks at D0, D4, D7 and D10 in SALL4 system.

7. In Fig. 4a, the authors should represent the number of SALL4 CUT&Tag peaks at days 0, 4, 7, and 10 of the reprogramming process.

Response: Thank you for your suggestion. We have addressed this concern by added the number of SALL4 CUT&Tag peaks during reprogramming process in Fig. 2e.

To provide the specific counts of SALL4 CUT&Tag peaks:

Day 0 (D0): 39,993 peaks

8. In Fig. 4b, the authors demonstrate that putative SALL4 binding sites are similar to the binding motifs of transcription factors such as FOS and JUNB. Does SALL4 function cooperatively with these transcription factors? Does overexpression or knockdown of these transcription factors affect the reprogramming efficiency in SALL4-mediated reprogramming?

Response: Thank you for raising this intriguing question. Our RNA-seq data showed that Fos and Atf3 showed slight upregulation during the reprogramming process, albeit with relatively low expression levels. Conversely, Junb, Ap-1, and Fos1/2 (Fra1/2) exhibited high expression levels but were notably downregulated during the early stages (D0-D7) of iPSC induction (Extended Data Fig.7c). Based on these data, we speculate that these transcription factors might inhibit reprogramming.

As the reviewer's suggestion, we conducted overexpression experiments and revealed that the overexpression of Junb, Ap-1, Fos1/2 (Fra1/2), Atf3, and Fos suppressed iPSC generation, aligning with our expectations (Fig.3g and Extended Data Fig.7d). Remarkably, the overexpression of BATF, a negative regulator of AP-1/ATF transcriptional events, demonstrated an improvement in reprogramming efficiency (Fig.3g and Extended Data Fig.7d).

To understand the functional relevance of these motif in SALL4-mediated reprogramming, we first performed zinc finger domain (DNA binding domain) deletion of SALL4, the 'TGACTCA' motifs still enriched in deletion of SALL4 ZFC1 or ZFC2, however, the peaks are relatively low.

We further integrated the binding region of 'TGACTCA' motifs of the bZIP family of transcription factors into WT SALL4 and zinc finger domain deletion of SALL4. The reprogramming experiment shows that the induction efficiency is improved by using SALL4-BATF-B fusion protein (Extended Data Fig.7g,h). Moreover, the defects caused by the deletion of SALL4 ZFC1 or ZFC2 can be rescued by the addition of BATF-DNA

binding region (Extended Data Fig.7g,h). These result suggests SALL4 function cooperatively with binding domain of AP-1/ATF to promote reprogramming.

We have included the results of the experiment integrating the BATF-DNA binding region into the SALL4 construct in the revised manuscript .

Fig. 3

The iPSCs induction efficiency using SALL4 overexpressing with representative SALL4-occupied gene *Esrrb*, *Rsk1*, ATAC-seq motif-enriched gene *Nkx6.1* and SALL4-binding peaks enriched-motifs related genes(left). The reprogramming efficiency induced by *Sall4* when knocking down *Tfap2c* or *Esrrb*(right). Data are mean \pm SD. Statistical analysis was performed using two-tailed, unpaired t test; $n = 6$ well from 3 independent experiments. **** $p < 0.0001$; *Sall4*+*Dsred* versus *Sall4*+*Rsk1*, ** $p = 0.0018$; *Sall4*+*Dsred* versus *Sall4*+*Batf*, ** $p = 0.0022$; *Sall4*+sh*Tfap2c* versus *Sall4*+sh*Luc*, ** $p = 0.0022$; *Sall4*+sh*Esrrb* versus *Sall4*+sh*Luc*, ** $p = 0.0022$. sh*Luc*, sh*Luciferase*.

Extended Fig.7 CUT&Tag analysis of SALL4 binding site during SALL4-driven iPSCs reprogramming

b. De novo motif enrichment of SALL4-binding peaks. Top 5 motifs on each day ranked by $-\text{Log}_{10}(\text{P-value})$ are shown, the left line (TF) of each chart shows proteins which could binding sequences most similar to the motif enriched by SALL4-binding peaks.

c. RNA-seq data shows the expression of representative genes in Fig. 2h. DR, DR, DSRED system. S4, SALL4 system.

d. Morphological diagram for the iPSCs induction process using Sall4 overexpressing with Batf or Jun, respectively. Scale bars, 200 μm .

e. Western blot shows the absence of BATF protein in MEFs and the sample of D7 during SALL4-reprogramming. D7 S4 oe MEF, D7 SALL4 overexpression MEFs.

f. Schematic representation of the protein sequences showing the structure of the wildtype Sall4 and Sall4 mutants which added the BATF-DNA binding domain.

g. Morphological diagram for the iPSCs induction process using the SALL4 mutants in Extended Fig. 7f. Scale bars, 200 μm .

h. The number of OCT4-GFP⁺ colonies on day 10 from 3×10^4 MEFs infected with SALL4 mutants in Extended Fig. 7f.

9. In Fig. 4e, the authors should explain why they focused on these specific genes. Additionally, to examine the role of these genes in SALL4-mediated reprogramming, the authors need to present the results of both overexpression and knockdown experiments for all the genes.

Response: Thank you for your feedback. Our selection of specific genes presented in Figure 4e was based on an analysis of our ATAC-seq and RNA-seq results to identify potential reprogramming promoting genes and downstream genes regulated by Sall4 during reprogramming. These genes were chosen due to their potential association with and influence on Sall4-mediated reprogramming. Upon verification through overexpression experiments, we were able to confirm that some of these selected genes indeed exhibit an improvement effect on Sall4-mediated reprogramming.

In response to your suggestion, we have conducted knockdown experiments for these specific genes to complement the overexpression studies. The results of these knockdown experiments have been included in Fig. 3g and Extended Data Fig. 7j showcasing the outcomes of the knockdown of these genes in the context of Sall4-mediated reprogramming.

10. Regarding Fig. 6e, the authors should explain and discuss the enriched GO terms for UC10, UC19, DC12, and DC21. The manuscript should include a detailed description and analysis of these GO terms to enhance the understanding of the findings.

Response: We appreciate your feedback and have taken steps to enhance the understanding of the enriched GO terms corresponding to UC10, UC19, DC12, and DC21.

We have integrated a comprehensive analysis of enriched GO terms into our manuscript to enhance the understanding of the findings presented in Fig. 6e.

11. In Extended Fig. 3b and c, the authors claim that OCT4 is a direct target of Sox2. However, it is insufficient to draw such a conclusion based solely on these data. Are there any data indicating that OCT4 binds to the regulatory regions of Sox2?

Response: Thank you for your feedback and scrutiny of our findings. We regret any misleading implications in our description. Our experimental data demonstrate that the overexpression of Oct4 has the capacity to activate the expression of Sox2. However, we do not possess data indicating a direct binding of OCT4 to the regulatory regions of Sox2. Contrarily, our observations do not indicate a similar effect by Sall4 on Sox2 expression.

We have revised the description in the results section to accurately reflect our findings and to avoid implying a direct binding of OCT4 to the regulatory regions of Sox2 based on the available data.

12. Based on the results from Fig. 7e and Extended Fig. 6d, it is predicted that TCF7 and LHX2 are critical transcription factors in both the SALL4-alone system and the SALL4 + OCT4 system. The effects of overexpressing these genes should be included in Fig. 7f.

Response: We appreciate your insightful observation. Indeed, our analyses predicted TCF7 and LHX2 to play pivotal roles in both the SALL4-alone system and the SALL4 + OCT4 system based on the results from Fig. 7e and Extended Fig. 6d (Fig.5b and Extended Fig.10c in revised manuscript). To explore the effects of overexpressing these genes, we conducted experiments, intending to elucidate their impacts on reprogramming efficiency.

However, the experimental outcomes yielded unexpected results. Overexpressing TCF7 exhibited a suppressive effect on reprogramming in both the SALL4-alone and the SALL4 + OCT4 systems. Conversely, overexpression of LHX2 demonstrated differing effects; it promoted reprogramming in the OCT4 system but inhibited reprogramming in the SALL4 system or OS system.

To provide a comprehensive understanding of these experimental outcomes, we have included these results in Extended Fig.10f, supplementing the data on the effects of overexpressing TCF7 and LHX2 in both reprogramming systems.

These unexpected findings underscore the complexity of interactions and the varied impact of specific transcription factors in different reprogramming contexts. We are committed to presenting these experimental results in the revised manuscript to enrich the understanding of the roles of TCF7 and LHX2 in reprogramming efficiency.

13. The authors describe TFAP2C and SOX2 as factors that promote reprogramming. However, in Fig. 7f, it is shown that overexpression of Tfp2c and Sox2 inhibits SALL4-mediated reprogramming. The authors need to address this contradiction. It should be investigated whether TFAP2C requires an appropriate expression level, which can be validated by varying its expression levels. Additionally, while Sox2 is recognized as one of the most crucial factors in reprogramming, the authors should determine if alternative factors can substitute for Sox2 in the SALL4 system.

Response: We appreciate your astute observation. Our experiments have revealed a complex scenario regarding TFAP2C and SOX2 in the reprogramming process.

Regarding TFAP2C, our findings present an intriguing dichotomy. While knockdown of TFAP2C in the SALL4 system inhibits reprogramming, suggesting the importance of TFAP2C activation in this context, overexpression experiments yield conflicting results. Contrary to expectations, overexpressing TFAP2C inhibits reprogramming, and intriguingly, the inhibition efficiency appears to increase with higher doses of Tfp2c virus infection (Additional Fig.2 in revised manuscript). This paradox prompts the speculation that premature activation of TFAP2C might impede reprogramming, indicating a potential necessity for appropriate timing in its activation to exert its beneficial effect.

Conversely, SOX2 is recognized as a crucial reprogramming factor; however, our findings present complex implications. Despite the enrichment of SOX2 in our ATAC-motif analysis, the expression of SOX2 cannot be detected in the SALL4 system.

Moreover, overexpressing SOX2 inhibits SALL4-mediated reprogramming but promotes reprogramming when combined with Oct4 or the OS combination (Oct4 + Sall4) (Fig.5h in revised manuscript). These contrasting effects underscore the variable roles of SOX2 in different reprogramming methods.

In light of these intriguing and paradoxical outcomes, we are dedicated to further investigations to decipher the precise roles and regulatory mechanisms of TFAP2C and SOX2 in reprogramming. Future studies might explore varying expression levels and timings of TFAP2C activation to delineate its optimal effect.

Additionally, It is well-known that the crucial role for SOX2 in regulates its downstream genes during OKSM-reprogramming. In SALL4 system, an appropriate regulate for these reprogramming related genes may also required. Notably, a small molecule-RepSox, which we used in our induction medium, have been reported previously to replace SOX2 in OKSM-reprogramming. We speculate that RepSox may regulate SOX2 -related downstream genes in SALL4 system to affect reprogramming.

We are committed to investigating these intricacies and aim to elucidate the nuanced roles of TFAP2C and SOX2 in reprogramming to enrich the understanding of these complex processes.

Additional Fig.2 The effect for overexpression of Tfp2c with Sall4 in iPSCs reprogramming

a.Morphological diagram for the SALL4+TFAP2C-iPSCs induction process. Scale bars, 200µm.

b.The number of OCT4-GFP+ colonies on day 10 from 3×10^4 MEFs infected with SALL4 and different volume of TFAP2C retroviral supernatants in iCD4.

14. In Fig. 7f, why do the authors focus on Sbsn and Mogat2? Is their function known in the context of reprogramming or pluripotency?

Response: We appreciate your inquiry regarding our focus on Sbsn and Mogat2 in Fig. 7f (Fig.5h in revised manuscript). These genes drew our attention due to their specific expression in the Sall4 system while being inhibited in the presence of Oct4.

Our observations indicate a unique expression pattern where Sbsn and Mogat2 are specifically activated in the context of the Sall4 system, but this expression is repressed in the presence of Oct4(Fig.5g). We hypothesize that the inhibition of these Sall4-activated genes by Oct4 might contribute to the cooperative effect observed when Oct4 collaborates with Sall4 to enhance reprogramming efficiency.

To investigate their impact, we verified the roles of Sbsn and Mogat2 in the OS system and confirmed that they indeed inhibit reprogramming efficiency(Fig.5h). These findings further substantiate the influence of these genes on reprogramming dynamics, shedding light on their inhibitory effect when Oct4 is introduced.

While the specific functions of Sbsn and Mogat2 in the context of reprogramming or pluripotency remain to be fully elucidated, their distinctive expression patterns in response to Sall4 and Oct4 interventions present intriguing avenues for further exploration. Understanding the precise roles and mechanisms of these genes could provide valuable insights into the regulatory dynamics of reprogramming.

Minor Points:

1. In Extended Fig. 1b, there is a missing description for Kenpaullone.

Response: In our revised file, we have provided a detailed description of Kenpaullone in Extended Fig. 1b.

2. In Fig 2c, no explanation is provided for the color values in the heatmap.

Response: We apologies for the omission in Fig. 2c. In the revised version, we have added a clear legend explaining the color values used in the heatmap for better interpretation.

3. In Extended Fig. 2b, there is an error in labeling "Cdhn2b." The correct labeling should be "Cdkn2b."

Response: We apologize for the error in Extended Fig. 2b (Extended Fig.6b in revised manuscript). We have corrected the labeling from "Cdhn2b" to the accurate designation "Cdkn2b" in the revised version.

4. In Fig. 3c and Fig. 7b, it would be beneficial to represent the size and number of the Venn diagrams in a manner that correlates with the data.

Response: Certainly, we appreciate your suggestion regarding Fig. 3c and Fig. 7b (Fig. 2c and Extended Fig.10b in revised manuscript). In the revised version, we have resized and adjust the Venn diagrams to reflect the size and proportions more accurately in correlation with the underlying data for better representation.

5. The figure legend is mislabeled in Extended Fig. 3a and 3b, where the labels are reversed.

Response: We apologize for the mislabeled figure legend in Extended Fig. 3a and 3b (Extended Fig.6 a,b in revised manuscript). In the revised version, we have corrected and appropriately label the figures to ensure accuracy in their representation.

6. In Extended Data Fig. 5, there is a mismatch between the order of the figures and their corresponding figure legends.

Response: We apologize for the mismatch between the order of the figures and their corresponding figure legends in Extended Data Fig. 5 (Extended Fig.8 in revised manuscript). In the revised version, we have ensured the correct alignment between the figures and their respective legends for clarity and accuracy.

7. In the legend of Extended Fig. 2d, "Fig7f" should be corrected to "Fig7e" as the accurate reference.

Response: We apologize for the error in the legend of Extended Fig. 6d (Extended Fig.10c in revised manuscript) where "Fig7f" was referenced incorrectly. We have rectified this in the revised version.

8. In the Methods section, the authors need to describe the criteria used for the differential

expression analysis of RNA-seq data, including the specific thresholds for fold change and q-value (adjusted p-value) that were applied.

Response: In the Methods section, we have included a description of the criteria used for the differential expression analysis of RNA-seq data. This will encompass the specific thresholds for fold change and q-value (adjusted p-value) applied in our analysis.

Reviewer #2 (Remarks to the Author): In this manuscript, Xiao et al. provide evidence that the transcription factor Sall4 can reprogram mouse embryonic fibroblasts to pluripotency. The authors also identify transcriptional and chromatin changes underlying Sall4-induced iPSC reprogramming and test the role of several downstream targets during cell fate reprogramming. Finally, they develop a model in which Sall4, together with Oct4, enhances reprogramming efficiency combinatorically. While the manuscript is well-crafted, and the data are robust, the lack of mechanistic details and novelty diminishes the overall enthusiasm for the presented story. Notably, Sall4 has been extensively investigated in reprogramming, and its mechanism of action involving interaction with the NURD complex is well-known (see Wang et al Nat Comm 2023). Moreover, it is important to note that the generation of iPSCs using a single transcription factor, such as Oct4, has been previously demonstrated, as mentioned by the authors.

Response: Dear Reviewer #2, Thank you for taking the time to review our manuscript on the reprogramming capabilities of Sall4 in inducing pluripotency in mouse embryonic fibroblasts. We appreciate your acknowledgment of the manuscript's well-crafted nature and the robustness of the presented data. Your feedback is valuable to us, and we understand your concerns regarding the mechanistic details and the perceived lack of novelty in our study.

You rightly point out the existing body of research on Sall4, especially its association with the NURD complex, as highlighted in the work by Wang et al. in Nat Comm 2023. While previous studies have indeed investigated Sall4's involvement in reprogramming and its established interactions, we aimed to expand on this by delineating the transcriptional and chromatin changes underlying Sall4 alone-induced iPSC reprogramming, shedding further light on its mechanistic role. Regarding the use of a single transcription factor, particularly Oct4, in iPSC generation, we acknowledge its previous demonstration, as mentioned in our manuscript. Our emphasis was not solely on the use of a single factor but rather on the collaborative enhancement between Sall4 and Oct4, which, to our knowledge, has not been extensively explored in the context of reprogramming efficiency. This cooperative effect was a focal point of our model. We recognize the importance of novelty in scientific research and its impact on the overall enthusiasm for a study.

In future work, we aim to delve deeper into the mechanistic intricacies, possibly exploring additional downstream targets or alternate pathways affected by Sall4's reprogramming abilities to provide a more comprehensive understanding of its role.

Thank you once again for your insightful comments and suggestions. We will take them into account as we continue to refine our research and aim to contribute novel insights to the field of reprogramming and pluripotency induction.

Specific suggestions:

- The authors propose that Sall4 can activate and repress genes. However, it remains unclear how it can execute this dual function. Have the authors explored its interaction with different chromatin remodelers?

Response: Thank you for highlighting the ambiguity in Sall4's dual function and its potential interaction with various chromatin remodelers. Previous investigations demonstrated that Sall4 recruits the NuRD (Nucleosome Remodeling and Deacetylase) complex, acting as a transcriptional repressor to drive reprogramming in collaboration with JGES (Jdp2, Glis1, Esrrb, and Sall4), specifically targeting close somatic loci.

To further explore whether this function plays a role in Sall4-alone-induced reprogramming, we disrupted the NuRD recruitment function by deleting the N-terminal NuRD recruitment domain in SALL4(Extended Data Fig.1h). The outcomes of the reprogramming experiments with this Sall4 mutant were insightful. Notably, the appearance of Oct4-positive cells was accelerated by the fifth day using the Sall4 mutant, exhibiting an efficiency surpassing that of WT Sall4 by day 7(Extended Data Fig.1j,k). However, it's important to note that by day 10, the Oct4-positive cells began to diminish. These results suggest that the deficiency in the NuRD recruitment function of Sall4 accelerates the reprogramming process in Sall4-alone-induced iPSC induction. This contrasts with the process in multi-factor-induced reprogramming setups.

Conversely, when the Zinc finger domains in Sall4 were impaired, the reprogramming was hindered(Extended Data Fig.1j). This observation implies that Sall4 might facilitate reprogramming through a DNA binding-related, yet unknown mechanism, emphasizing the significance of these domains in the process.

These findings underscore the complexity of Sall4's role in reprogramming, indicating its multifaceted interactions with chromatin remodelers and the diverse mechanisms underlying its dual function. Further investigations are crucial to comprehensively unravel the precise mechanisms through which Sall4 orchestrates its dual function, interacts with chromatin remodelers, and navigates the reprogramming landscape.

Extended Fig.1 Small molecule screening for the determination of induction medium that could driving SALL4-mediated iPSCs reprogramming successfully

h. Schematic representation of the protein sequences showing the structure of the wildtype *Sall4* and *Sall4* mutants. Color codes of ZFC1, ZFC2, ZFC3 and N12 are defined as described in Figure.

i. Western blot shows the overexpression of *Sall4* mutants in MEFs.

j. The iPSCs induction efficiency using *Sall4* mutants. wildtype *Sall4* as positive control are shown. Data are mean \pm SD. n = 6 well from 3 independent experiments.

k. Morphological diagram for the iPSCs induction process using *SALL4*- Δ N12. Scale bars, 200 μ m.

- The data in Figure 7F suggest that overexpression of Sox2 together with *Sall4* impairs the capacity of *Sall4* to reprogram cells into iPSCs. This finding is surprising and might give the paper a different, more novel perspective. Do the *Sall4*+Sox2 expressing cells diverge towards an alternative fate?

Response: Thank you for your insightful observation. While Sox2 is known to be a critical reprogramming factor and can enhance Oct4-driven reprogramming, our findings were indeed surprising. Contrary to expectations, the overexpression of Sox2 substantially suppressed *Sall4*-driven reprogramming, indicating an unexpected and contrasting effect.

To understand this unexpected outcome, we proceeded with a comparative RNA-seq analysis by collecting samples of Oct4+Sox2 and Sall4+Sox2 at day 0 and day 7. Using DsRed as a control, we aimed to elucidate the differential expression of genes induced by Sox2 in these two sample sets.

The initial comparison of differentially expressed genes influenced by Sox2 in the Sall4+Sox2 samples showed intriguing outcomes. We identified and compared the up-regulated and down-regulated genes in Sall4+Sox2 and Sall4+DR samples. The results shows that 250 new up-regulated genes(C1) increased in Sall4+Sox2 samples and 202 up-regulated genes in Sall4+DR can not be enriched in Sall4+Sox2 samples, suggest Sox2 may inhibits the up-regulating of these genes to suppress SALL4-reprogramming. Consistently, the 75 down-regulated genes(C3) in Sall4+DR also be reversed by Sox2. We next performed GO analysis for these genes to understand the function of the these Sox2-related genes. The biological process enriched in C1 contains skeletal development such as positive regulation of osteoblast differentiation, the biological process enriched in C3 contains PI3K signaling and inflammatory response related process(Extended Data Fig.12a-c).

To investigate the cell fate transition regulated by Sall4, we checked the expression for groups of three primary germ layers related genes. The results shows the addition of Sox2 in SALL4 system leads a group of pluripotency genes downregulated and a group of Ectoderm genes upregulated, thus induced the cell to another fate(Extended Data Fig.12d)

The analysis is ongoing and will provide a more comprehensive understanding of the divergent effects observed in Sox2-assisted reprogramming between Oct4 and Sall4 systems.

This unexpected and intriguing finding suggests the potential divergence of Sox2-mediated reprogramming between Oct4 and Sall4 systems, indicating the necessity for further investigation into the underlying mechanisms or alternative fate determination in the Sall4+Sox2 expressing cells. These outcomes might shed light on novel pathways or processes governing reprogramming dynamics, providing a fresh and insightful perspective on cellular fate determination in different reprogramming contexts.

Extended Fig.12 Sox2 have an opposite effect in *Sall4*-reprogramming and *Oct4*-reprogramming

a. Diagram for RNA-seq data collecting during *Sox2* related reprogramming process.

b. Venn diagrams shows the number of differential expression genes in *Sox2* related reprogramming process.

c. Left, GO analysis for genes specific-upregulated in *Sall4*+DsRed group(C1) in Extended Fig.12b. Right, GO analysis for genes specific-downregulated in *Sall4*+DsRed group(C3) in Extended Fig.12b.

d. Heatmap showing expression of master regulator genes for each of the three primary germ layers at day7.

- Alternatively, the authors could investigate how Sall4 cooperates with Rsk1, Esrrb, or/and Nkx6.1 to reprogram cells. Do they bind to the same loci? Could these factors uncouple the activator vs. repressor function of Sall4?

Response: Thank you for highlighting the need for a deeper investigation into the mechanisms underlying the reprogramming functions of genes regulated by SALL4.

To know whether these genes can bind to the same loci with SALL4, we checked the SALL4-CUT&TAG data, we find neither Rsk1 nor Esrrb have its motif. The motif for Nkx6.1 have enriched in our data, however, the P-value is high.

Furthermore, we also found Sall4 binding site in jacent promoter region of these genes, companied by changes of chromatin accessibility.

In addition, our reprogramming data shows that overexpression of Esrrb or Nkx6.1 with Sall4 significantly promotes reprogramming, and Rsk1 can also slightly improve the efficiency. These data support the notion that Rsk1 and Esrrb are downstream targets of Sall4 rather than co-activator or repressor of Sall4.

Interestingly, we find a significant enrichment of Nkx6.1 and Esrrb in our ATAC-seq motif analysis during SALL4-reprogramming(Fig.2d). This indicate that there are large of Nkx6.1 and Esrrb binding sites on the chromatin. The reprogramming promoting effect of Nkx6.1 and Esrrb may through binding these loci and and regulates the chromatin accessibility in these regions, leading a more plastic state for the somatic cells, making it easier for pluripotency acquisition.

- The PCA in Figure 6B and the heatmap in 6D do not show a massive enhancement of efficiency by the combination of Sall4+Oct4. This partially contradicts the text. I suggest the authors tone down the statements about the combination of these factors being highly efficient. The authors should also refrain from stating that they have a highly efficient reprogramming system, as the overall efficiency is only 0.66%.

Response: Thank you for your feedback. We agree that the figures don't strongly support our claims about the efficiency of Sall4+Oct4. We'll adjust the text to reflect this more accurately and avoid overstating our reprogramming system's efficiency, which stands at 0.66%. Appreciate your input.

- The authors could further investigate the binding profiles and interactors of the Sall4 mutants. Although three mutants were generated, all of which failed to give rise to

pluripotent cells in reprogramming assays, it is unclear why each of the mutants is unable to reprogram cells.

Response: Thank you for your suggestion. We found that the deletion of zinc finger domains in the Sall4 mutants hinders their reprogramming ability. To investigate the changes for its DNA binding loci, we performed Cut&Tag for these mutants. Our Cut&Tag analysis revealed distinct DNA binding differences between each mutant and the wild-type SALL4, as depicted in Extended figure.7a. This clarifies how these mutations affect DNA binding, contributing to their inability to support reprogramming. We also find a SALL4-enriched motif related genes batf, the defects caused by the deletion of SALL4 ZFC1 or ZFC2 can be rescued by the addition of BATF-DNA binding region(Extended Fig.7g,h). This suggest SALL4 may binding and regulates Batf motif-related genes to promote reprogramming.

Minor points:

- There are no details about how the authors stratified their OC/CO peaks in Figure 3B (not in the legend nor in the main text).

Response: We apologies for the oversight. In the revised manuscript, we have detailed the method used to stratify OC/CO peaks in Figure 3B (Fig.2b in revised manuscript), providing a clear explanation in methods.

- In Figure 6, the authors should explain what makes S/OS different from O/OS and S/O/OS.

Response: Certainly, we appreciate the suggestion. In the revised manuscript, we have provided a clear explanation in the Figure 6 (Fig.4e,f in revised manuscript) caption to delineate the specific characteristics that differentiate S/OS from O/OS and S/O/OS. This clarification will enhance the reader's understanding of the distinctions between these categories.

- The statement “SALL4 may play a more significant role in reprogramming than COT4 in the OS system” can be misleading, as the reprogramming medium was initially selected to enhance specifically Sall4 activity, not Oct4.

Response: Thank you for highlighting this point. In the revised manuscript, we have modified the statement to avoid potential misunderstanding. We have clarified that the reprogramming medium was initially designed to specifically enhance Sall4 activity rather than Oct4, which might have influenced the observed roles of Sall4 and Oct4.

- The scales of Figure 1f and 5e are not specified in the legend.

Response: We acknowledge the missing scale specification in Figure 1f and 5e (Fig.1f and Extended Fig.8a in revised manuscript). Our qRT-PCR data was analyzed using the Δ Ct method and all the individual gene expressions were normized to the expression of GAPDH to show its relative amount. In the revised version, we will include a clear illustration in the figure legends to accurately define the measurement scales used in these figures. This addition may ensure better comprehension of the represented data.

- For the Cut&Tag shown in Figure 4d, it is unclear what reprogramming day was used.

Response: Thank you for bringing this to our attention. In the revised manuscript, we have specified the reprogramming day utilized for the Cut&Tag presented in Figure 4d (Fig.2e in revised manuscript). This information will be clearly stated to provide context and aid in understanding the experimental timeline.

Reviewer #3 (Remarks to the Author): Xiao et al. explores the role of SALL4, and the combination of SALL4 and OCT4 in somatic cell reprogramming to iPSCs. The study establishes an efficient method for iPSC induction using SALL4 as a single factor combined with TGFb inhibitor RepSox. The generated SALL4-induced iPSCs exhibited a gene expression similar to ESCs, were able to form teratoma, and could contribute to development of chimeric mice. While OCT4 alone could induce iPSC generation, the efficiency was lower compared to SALL4, and co-overexpressing SALL4 and OCT4 significantly enhanced reprogramming efficiency. The manuscript provides some insights into the cooperation between SALL4 and OCT4 in 2-factor reprogramming.

The study successfully establishes a robust method for reprogramming somatic cells into iPSCs using SALL4 alone, thereby expanding the repertoire of pluripotency-inducing cocktails in mice. However, the manuscript lacks sufficient exploration and explanation of the advances for the field. Additionally, the writing quality requires improvement and further work.

Response: Dear Reviewer #3, We sincerely appreciate your thoughtful evaluation of our manuscript exploring the role of SALL4 and its collaboration with OCT4 in somatic cell reprogramming toward induced pluripotent stem cells (iPSCs). Your insights are invaluable in guiding us toward refining and augmenting our study's contributions.

We are pleased that our study effectively introduces a robust method for inducing iPSCs using SALL4 as a single factor, complemented by the TGFb inhibitor RepSox. Our findings demonstrating the similarities in gene expression to embryonic stem cells (ESCs), teratoma formation, and the ability of these SALL4-induced iPSCs to contribute to the development of chimeric mice serve as foundational support for the potential applications and efficacy of this reprogramming method.

We acknowledge the identified gap in our manuscript regarding a detailed exploration and explanation of the advances made in the field. As such, we intend to expand our discussion to better emphasize the unique contributions and implications of our work within the broader landscape of iPSC research. This expansion will include a more comprehensive analysis of how the efficient induction of iPSCs via SALL4 alone, and in combination with OCT4, can diversify the available strategies for generating pluripotent cells in mice, potentially enhancing the toolkit for regenerative medicine applications.

Furthermore, we take your feedback on the writing quality seriously and are committed to improving this aspect of our manuscript. We will ensure clarity, coherence,

and a more compelling narrative to effectively communicate the significance and implications of our findings. We are also dedicated to pursuing further investigations and addressing the lacunae identified in our work to offer a more thorough understanding of the cooperation between SALL4 and OCT4 in the two-factor reprogramming process. These additional studies will aim to provide a deeper mechanistic insight into this collaborative reprogramming approach.

We are grateful for your constructive feedback, and we will diligently work on revising the manuscript to reflect these enhancements. Your guidance is invaluable in our efforts to contribute meaningfully to the field of iPSC research.

Major issues:

- The study does not compare the efficiency SALL4-based reprogramming method with standard reprogramming cocktail, such as OSKM and OSK.

Response: Thank you for your suggestion. We have indeed conducted experiments comparing the efficiency of the SALL4-based reprogramming method with standard reprogramming cocktails like OSKM and OSK. The results of these experiments have been included in the Extended Figure 2 for your reference.

- The authors have not explained why reprogramming with a single factor as opposed to multiple factors is important.

Response: Thank you for your input. Reprogramming with a single factor offers several crucial advantages:

1. **Simplicity and Efficiency:** Using a single factor simplifies the reprogramming process, streamlining it for enhanced efficiency. This streamlined approach is particularly beneficial for applications in regenerative medicine, where simplicity and scalability are critical factors.
2. **Reduced Risk of Genetic Instability:** Using a single reprogramming factor may reduce the potential risks associated with genetic instability and oncogenic transformation often linked to the overexpression of multiple factors. This reduction in risk enhances the safety profile of the reprogramming process, an essential consideration for potential applications.
3. **Insight into Mechanisms:** Focusing on a single factor allows researchers to delve deeper into the specific role and mechanisms of that factor in reprogramming.

This detailed understanding can have broad implications, shedding light on pluripotency and differentiation mechanisms and contributing significantly to the collective knowledge in the field.

These points are now incorporated into the introduction and discussion sections of our paper, reinforcing the importance and advantages of utilizing a single reprogramming factor. Moreover, we've supplemented these claims with supportive evidence and references for added context and credibility.

- The research highlights the importance of the cooperative interaction between SALL4 and OCT4 in enhancing reprogramming efficiency, providing new insights into the reprogramming process. However, how exactly the two factors cooperate remains unexplored.

Response: Thank you for the observation. The collaboration between SALL4 and OCT4 in reprogramming is indeed pivotal. Our findings reveal that this synergy operates through multiple avenues:

Rescuing Transcriptional Barriers: SALL4 and OCT4 systems exhibit gene expression deviations from the ESC state. Their combination in the OS system neutralizes these discrepancies(Fig.4e,f). For instance, SALL4 specifically activates Rsk1, Esrrb, and Tfp2c, which act as drivers for OS-mediated reprogramming. Interestingly, these factors significantly promote reprogramming in both OS and OCT4 systems. However, OCT4 alone cannot activate Rsk1, Esrrb, or Tfp2c. In the OS system, SALL4 takes charge of activating these genes, enhancing the OS-mediated reprogramming process(Fig.5f-h). Moreover, while Sox2 plays a significant promoting role in both OCT4 and OS systems, its low expression presents a barrier in the SALL4 system(Fig.5f-h and Extended Fig.5d,e).

Overcoming Reprogramming Barriers: Collaboratively, SALL4 and OCT4 suppress the expression of genes hindering the reprogramming process. For instance, Nkx6.1, activated by Oct4, impedes OS-mediated reprogramming, a barrier that can be countered by SALL4(Fig.5f-h and Extended Fig.5d,e). Similarly, genes activated by SALL4 that hinder OS reprogramming, such as Mogat2 and Sbsn, can be downregulated by Oct4(Fig.5f-h).

Simultaneous Regulation: There exists a subset of genes in the OS system requiring joint regulation by OCT4 and SALL4. This joint modulation indicates the necessity for

their combined action in influencing these specific genes crucial for successful reprogramming(Fig.4e,f and Fig.5e).

The cooperative action of SALL4 and OCT4 not only amplifies their reprogramming capabilities but also mitigates elements hindering reprogramming. This expanded regulation spectrum accelerates the activation of more reprogramming-related genes and suppresses the expression of somatic cell-related genes. As a result, this collaboration significantly enhances the efficiency of induced pluripotent stem cell generation.

Fig. 6 A model for SALL4 cooperated with OCT4 to enhance iPSCs generation.

- Although the study identifies specific genes regulated by SALL4 and OCT4, it is not clear how some of those downstream targets subsequently achieve induction of pluripotency. While the roles of Sox2 and Esrrb are well studied, some other targets remain a mystery.

Response: Thank you for highlighting the need for a deeper investigation into the mechanisms underlying the reprogramming functions of genes regulated by SALL4 and OCT4.

Our data shows that SALL4 activates Esrrb, Rsk1 and Tfap2c in OS-mediated iPSCs reprogramming to facilitates induction efficiency. Among the three major SALL4 regulated genes we identified, Esrrb are previously reported to play an important role in pluripotency induction. Tfap2c are pluripotent transcription factors that are reported to regulate naïve pluripotency. The functions of these genes support that SALL4 activate known pluripotent regulators to facilitate reprogramming.

Our data also shows that overexpression of Rsk1 slightly improve the reprogramming efficiency. Rsk1 is a ribosomal S6 kinase. This kinase contains 2 nonidentical kinase

catalytic domains and phosphorylates various substrates, including members of the mitogen-activated kinase (MAPK) signalling pathway. Our cut-tag data shows that SALL4 directly regulate the promoter region of this genes. Our data supported the notion that in addition to conventional transcriptional factors, SALL4 also regulate other epigenetic modifiers to cooperatively contribute to reprogramming.

We believe these data highlighted the molecular mechanism of SALL4-reprogramming.

- The study lacks many controls, such as comparison of expression levels between the mutants, viral titers, efficiencies of knockdowns, etc.

Response: We value your constructive feedback highlighting the necessity for additional controls in our study. Your insights have underscored the areas where our research could significantly benefit from more comprehensive experimental validation. We have preformed these experiments and added the results in the Additional Fig.1, Extended Fig.1i and Extended Fig.7j.

Additional Fig.1 Determination of retrovirus infection efficiency

a. Flow cytometry was used to analyze the retrovirus infection efficiency for gradient-diluted retroviral supernatants (1 represent undiluted original retroviral supernatants, 1/10 represent retroviral supernatants after 10 times dilution). n = 3 well from 3 independent experiments.

b. Line graph for Additional Fig.1a shows an exponential relationship between dilution ratio and infection efficiency in the 0.001-0.01 (1/1000-1/100) dilution ratios.

c. The retrovirus infection efficiency calculated by 0.001-0.01 (1/1000-1/100) dilution ratios. The infection efficiency for 1ml original retroviral supernatants are shown.

1/100) dilution ratios. The infection efficiency for 1ml original retroviral supernatants are shown.

- The manuscript needs a lot more work when it comes to writing.

Response: We appreciate the feedback regarding the written content of the manuscript. We will dedicate more effort to refine and enhance the writing to ensure a clearer and more polished presentation of our research. Thank you for highlighting this aspect, and we are committed to improving the overall quality of the manuscript.

- The paper does not extensively discuss potential limitations of the study, or the specific context in which these findings can be applied.

Response: Thank you for your feedback. We've included a discussion in the manuscript addressing the study's limitations and potential applications. While successful iPSC generation using a single factor, SALL4, in MEFs has been achieved, the induction efficiency remains lower than that of multi-factor reprogramming. This suggests the need for further optimization of the induction medium, iCD4, for SALL4-iPSCs induction.

Furthermore, our research delves into the complex interactions between OCT4 and SALL4, primarily exploring gene expression regulation. However, this interaction's complete mechanisms and contributions are yet to be fully uncovered. Epigenetic regulation, especially SALL4's role in chromatin remodeling proteins and CADs, stands as a critical area for future investigation, potentially offering alternative principles for cell fate control during reprogramming.

Lastly, it's important to note that while bulk data from RNA-seq and ATAC-seq offer insights, they might not capture the nuances of SALL4-mediated reprogramming comprehensively. Therefore, defining cell fate changes during SALL4-mediated reprogramming at the single-cell resolution level could be immensely valuable.

- The authors should thoroughly familiarize themselves with the relevant literature to enhance the quality of their manuscript. The current version lacks sufficient citations, and some of the references provided are either irrelevant or not original. Moreover, numerous claims made in the manuscript are not supported by relevant citations. Below are some examples:

1) The first report on RepSox was not cited (Ichida et al., Cell Stem Cell, 2009). This is unacceptable given the key role of RepSox in the manuscript.

2) The first study that identified Oct4 as irreplaceable for iPSC generation was not cited (Nakagawa et al., Nature Biotech., 2008), and instead more recent papers were chosen.

3) The first study on Oct4 alone iPSC generation by Kim et al., Nature 2009 was not

cited.

4) The claim that Oct4 is needed for maintenance of pluripotency was not supported by the relevant citation (e.g. Niwa et al., Mol. Cell. Biol. , 2002)

5) A recent study on Sall4 has not been cited or discussed:

The NuRD complex cooperates with SALL4 to orchestrate reprogramming | Nature Communications

6) One of the first studies on Sall4 and pluripotency was not cited:

Sall4 modulates embryonic stem cell pluripotency and early embryonic development by the transcriptional regulation of Pou5f1 | Nature Cell Biology

Response: Dear Reviewer, We genuinely appreciate your meticulous assessment and valuable recommendations to enhance the quality of our manuscript. Your insights regarding the inclusion of specific key references are immensely helpful in strengthening the foundations of our work.

We acknowledge the importance of citing the foundational studies in the field, and your points regarding the absence of critical citations, such as Ichida et al. (2009), Nakagawa et al. (2008), and Kim et al. (2009), are well taken. Additionally, we understand the significance of referencing more established work, such as Niwa et al. (2002), and recent studies, including those on Sall4, that we have overlooked.

We have promptly addressed these omissions by modifying our manuscript to include the relevant citations you've pointed out. Your suggestions will be crucial in strengthening the scholarly foundation of our work, and we are committed to ensuring a more comprehensive and well-supported presentation of our research.

Additional specific issues:

The sentence “Interestingly, dropping out experiment shown that SALL4 has been found to be the most critical factor in the reprogramming cocktails.” is out of context without introducing the study in more details (e.g. listing the exact cocktail from which Sall4 was dropped out).

Response: Thank you for noting the context issue in the sentence about SALL4 in the reprogramming process. We'll update it to include details on the specific experiment where SALL4 was omitted from the cocktail. This revision aims to provide a clearer understanding within the study's framework. Appreciate your feedback.

In general, the introduction could be expanded to provide a more comprehensive background on iPSCs, and in particular on previous reprogramming cocktails. This would help readers better understand the context of the study.

Response: Thank you for your valuable feedback on the introduction section of our manuscript. We acknowledge the suggestion to expand and provide a more comprehensive background on induced pluripotent stem cells (iPSCs) and specifically on previous reprogramming cocktails. By elaborating on these aspects, we aim to offer readers a better contextual understanding of the study. We have work on expanded the introduction to provide a more detailed and informative overview, enhancing the overall context for our readers. Your input is highly appreciated, and we are committed to improving the introductory section accordingly.

Fig. 1h: Please include separate channels, it appears like Oct4 and especially Sox2 are not nuclear-localized.

Response: We have updated Figure 1h (Fig.1h in revised manuscript) to include separate channels, specifically highlighting the nuclear localization of Sox2. This adjustment will provide a clearer visualization of their nuclear localization within the context of our study. Thank you for pointing this out.

Extended Fig. 1a: It seems that most of the compounds used in the study can facilitate Sall4 alone reprogramming, even though they target unrelated pathways. Surprisingly, the authors did not discuss this in the text. To gain a better understanding of the specific effects on Sall4 preprogramming, it would be beneficial to include experiments with OSKM or OSK. This would allow for a clearer distinction between the impact on Sall4 versus OSK reprogramming.

Response: Thank you for your valuable input. While our study primarily focuses on SALL4-driven reprogramming using iCD4, we have noted that the induction efficiency

remains relatively low, hinting at potential limitations in the iCD4 induction conditions for SALL4-driven reprogramming. Although several compounds in our study support SALL4-reprogramming, their individual effects seem rather modest. As a result, we haven't extensively delved into their implications.

To discern the influence of these compounds on various reprogramming methodologies, we conducted experiments involving OSKM or OSK. Notably, our findings revealed that RepSox marginally reduces efficiency, Forskolin enhances OSK reprogramming, while the other compounds exhibit minimal promoting or inhibiting effects (Extended Fig.2 in revised manuscript). This suggests that RepSox might have varying effects on these different reprogramming methods, urging a deeper investigation.

Extended Fig.2 The different effect of small molecules in OKS-driven reprogramming and SALL4-driven reprogramming

- a. GFP+ clones collected from the whole wells in 24 well plate shows the OKS-driven iPSCs induction efficiency using iCD4-Repsox medium added with small molecules in Extended Fig. 1 b.
- b. Flow cytometry was used to analyze the iPSCs induction efficiency in Extended Fig. 2a.
- c. The histogram shows the iPSCs induction efficiency in Extended Fig. 2b. Data are mean \pm SD. n = 6 wells from 3 independent experiments.
- d. GFP+ clones collected from the whole wells in 24 well plate shows the OKS+SALL4-driven iPSCs induction efficiency using iCD4 medium at day 6.
- e. Flow cytometry was used to analyze the iPSCs induction efficiency in Extended Fig. 2d.

Notably, the compounds of Vc, Chir99021, SGC0946 and RepSox are most important in iCD4 medium (Extended Data Fig. 1f).” The list of “most important” appears to be unfair, e.g. omitting RepSox could still generate some iPSCs, while omitting bFGF generated zero colonies. The need of bFGF is interesting, as this is a component of primed media, which mouse ESCs normally do not require.

Response: Thank you for your astute observation regarding the description of the "most important" compounds in iCD4 medium, as illustrated in Extended Fig. 1f. We acknowledge that labeling these as the "most important" might be misleading within the given context.

We have reassessed the representation of these compounds concerning their necessity for reprogramming, ensuring a fair and accurate depiction of their individual impacts within the iCD4 medium. Your insights are invaluable in rectifying this discrepancy. We have revised the manuscript description accordingly, elaborating on the role of bFGF in reprogramming and its effect on MEFs' cytoactivity. Its removal significantly diminishes cell proliferation in iCD series medium, ultimately leading to cell death.

Extended Data Fig. 1b: the table does not include the target of Kenpaullone.

Response: Thank you for bringing this to our attention. We apologize for the oversight in Extended Fig. 1b, where the target of Kenpaullone was not included in the table. We have promptly updated the table to include the specific target of Kenpaullone, ensuring a more comprehensive and accurate representation of the data.

Extended Data Fig. 1e: the cells look dead. Were you able to establish iPSC lines from TTFs with Sall4 alone? Please include those data.

Response: Thank you for your insightful observation. We've made adjustments to the dataset to address the concern raised. In our current work, we have encountered challenges in establishing stable iPSC lines from TTFs using Sall4 alone. The difficulties primarily stem from the extended induction period and notably low efficiency in the reprogramming of TTFs.

To provide further clarity, we included an additional image in the dataset. While we haven't yet achieved the establishment of stable iPSC lines from TTFs using Sall4 alone, we remain optimistic. We believe that improvements in the induction process, particularly in enhancing efficiency, could pave the way for successful iPSC line establishment from TTFs in the future.

Extended Data Fig. f1: While figure 1a suggested that no iPSCs could be generated in iCD1 media, but 1f suggests that a few colonies could be generated in the absence of RepSox. Which is true? Please include the data or explain the discrepancy in the text.

Response: Thank you for your inquiry. The discrepancy in the observed outcomes is due to methodological adjustments made during the experiment. Upon realizing Repsox's role in promoting Sall4-reprogramming, we conducted additional experiments. Removing B27 caused cell death during induction when using the initial screening medium, iCD1-LiCl with Y27632, SGC0946, and gsk-lsd1. Consequently, we modified the experimental conditions by doubling the concentration of B27 in this medium. Subsequently, we observed the generation of OCT4-GFP+ cells even in the absence of Repsox, although the efficiency was notably low. The specific components of these two media were detailed in the methods section.

Extended Data Fig. 1i: Such an experiment certainly requires western blot confirming the expression of the mutant, as well as virus titration data.

Response: Thank you for emphasizing the necessity of additional supportive data in Extended Fig. 1i (Extended Fig. 1j in revised manuscript). We recognize the importance of further validation, particularly through western blot analysis to confirm the expression of the mutant, along with virus titration data.

In response, we have conducted the required experiments and included the results in the Extended Figure 1i and Additional Figure 1. This addition aims to provide a more comprehensive and substantiated representation of the experiment. Your feedback is invaluable in ensuring the strength and thoroughness of our findings.

The statement: “These findings underscore the crucial role of SALL4’s DNA-binding ability in mediating the reprogramming process.” requires DNA-binding data for the deletion mutants.

Response: Thank you for your valuable insight. We recognize the necessity of supporting the statement regarding the importance of SALL4's DNA-binding ability in the reprogramming process with specific DNA-binding data for the deletion mutants.

To address this crucial aspect, we have conducted the necessary experiments to provide detailed DNA-binding data for the deletion mutants. Our Cut&Tag data for these mutants shows a relatively lower enrichment signal in the whole binding landscape, this indicated a decreased DNA binding ability and the changes of binding sites for SALL4 mutants(Extended Fig.7a).

Fig. 2b: As far as I understand this is bulk RNA-seq. Please show either time-course single-cell RNA-seq or time-course FACS for reprogramming intermediates (e.g. Thy1, E-cad, Oct4-GFP).

Response: Thank you for your input. To capture a more comprehensive understanding of the SALL4-mediated reprogramming process, we conducted single-cell RNA-seq analysis on Day 10 samples. Regrettably, due to the low induction efficiency, precise clustering of iPSCs from this dataset was challenging, thereby hindering comprehensive cell trajectory analysis at this time. As such, we are actively seeking improved methods with higher induction efficiency to facilitate time-course single-cell RNA-seq analysis.

We appreciate your suggestion of we work on analyzing this data to provide a comprehensive understanding of the proportion of cells in the process of reprogramming. We could further analysis the data for the manuscript if necessary.

In parallel, to delineate reprogramming intermediates during SALL4-mediated reprogramming, we performed time-course Fluorescence-Activated Cell Sorting (FACS) using previously reported cell surface markers Thy1 and Epcam, known to be associated with OKSM-reprogramming intermediates. Our analysis revealed a progressive increase in a distinct cluster of Thy1-/Epcam+ cells during reprogramming. Subsequently, we sorted

these cells at day 7 and induced them using iCD4 medium. In comparison to the unsorted cells, after four days of induction, we observed the emergence of Oct4-GFP positive cells in both the control group and the Thy1-/Epcam+ cluster. This finding strongly indicates that the Thy1-/Epcam+ cluster represents reprogramming intermediates in SALL4-

mediated reprogramming. These results have been integrated into Extended Fig. 3.

Extended Fig.3 The Thy1-Epcam⁺ subgroup in SALL4 system has the potential to generate iPSCs

a. Flow cytometry was used to analyze the proportion of THY1-EPCAM⁺ subgroup in SALL4 system at day0, day4 and day7, respectively.

b. Morphological diagram for the iPSCs generation at day4 induced from the day7 THY1-EPCAM⁺ subgroup in SALL4 system. Scale bars, 200μm.

c. The iPSCs induction efficiency induced from subgroups classified by THY1 and EPCAM. Data are mean ± SD. n =2 well from 2 independent experiments.

What are the genes in C4 that are strongly downregulated in Sall4 samples compared to both control and ESCs?

Response: Thank you for your valuable input.

Regarding the RNA-seq profiles of DsRed and Sall4-reprogramming, the genes within the C4 subgroup showcase downregulation compared to DsRed and ESC control profiles. Through conducting a Gene Ontology (GO) analysis for these genes, we've discovered enrichment in immune-related pathways within this subgroup. These results have been included in the Extended Figure.4g.

Fig. 3d and Extended Data Fig.3: The highlighted loci are often meaningless. For example, activation of Oct4 distal enhancer (Oct4DE) is known to be key for induction of pluripotency in mouse, but the figure does not include Oct4DE, and instead some intron binding is highlighted. The panels have to be expanded to include larger 5' regions, which most often contain enhancers and promoters.

Response: Thank you for your astute observation.

To address this concern, we plan to expand the panels in these Figures. By enlarging the 5' or 3' regions, we aim to encompass key regulatory elements such as enhancers and promoters, including Oct4DE. Notably, we can find a significant SALL4 cut&tag signal in the Oct4 region (Figure.3e, 5f, Extended Figure.7a,b and Extended Figure.10d).

Fig. 3e: With the exception of Rsk1 the changes of selected pluripotency genes certainly cannot be described as "increased gradually".

Response: Thank you for your observation regarding Figure 3e. We appreciate your feedback on the description "increased gradually" in relation to the changes observed in the selected pluripotency genes.

Upon re-evaluation, we recognize that, apart from Rsk1, the alterations in the chosen pluripotency genes do not uniformly follow a "gradual increase" pattern. We have carefully rephrased and accurately described the changes exhibited by these genes to better reflect the observed variations in their expression levels.

Extended Data Fig. 4b: It is surprising that you do not see Sall4 motif in the list. Does it mean that Sall4 cannot directly open/ close chromatin? This should be discussed Fig. 4b: no Sall4 motif? This paper suggests it is different from what the manuscript suggests: Zinc Finger Protein SALL4 Functions through an AT-Rich Motif to Regulate Gene Expression - PubMed (nih.gov).

Response: Thank you for your valuable input. We found a Sall4 motif in the JASPAR database (<https://jaspar2020.genereg.net/matrix/UN0262.1/>). However, our de novo analysis revealing that only the sequence TG***CA matched those in the JASPAR database.

We extended our search to the existing literature and found various motifs in previous studies:

- "AC[A/T][A/T][T/A]GT" (doi: 10.1242/dev.132761)
- "TTGTCTACTTGGTA" or "ATTTGCATATAA" (doi: 10.1128/MCB.00419-10)
- "AA[T/A]TAT[T/G][G/A]" or "A[T/A]TAT[T/G][G/A]" (DOI:10.1016/j.celrep.2020.108574)

These motifs were identified in different cell types. However, our data did not align with these motifs. This discrepancy suggests that Sall4 binding motifs might vary across different cell types.

Fig. 4d: For the binding data, the comparison with untransfected MEFs does not make sense, since those have no Sall4 at all, so surely there is no signal. A good positive control is needed, such as ESCs, where Sall4 is expressed. This could resolve the motif discrepancy too. DNA binding data (CUT&RUN) should be overlapped with chromatin accessibility data (ATAC-seq) to see if Sall4 binding is consequential for chromatin landscape.

Response: Thank you for your valuable insight. We apologize for any confusion in our previous description. The binding data analysis we conducted involved samples of Sall4 overexpression, pulled with the Flag antibody, while the control samples also entailed Sall4 overexpression, pulled with the IgG antibody (Figure.2e and Extended Figure.11a). We have made specific revisions in the manuscript to clarify these details.

We have performed a joint analysis of Cut&tag and ATAC data. Our findings reveal that approximately 89.5% (35779 out of 39993) of the Cut&tag sites in the Sall4 system at D0 were situated within the ATAC open regions (N=178854), which aligns with

expectations (Figure.3a-c). This outcome indicates that Sall4 predominantly binds to open chromatin regions.

Your feedback has been instrumental, and these clarifications and additional data analyses have been included in the revised work to ensure the accuracy and completeness of our findings.

Extended Data Fig. 5: It appears like some panels are missing, as the legends do not correspond to the figure.

Response: Thank you for highlighting the discrepancy in Extended Fig. 5 (Extended Fig.8 in revised manuscript). We apologize for any confusion caused by the mismatch between the legends and the actual content displayed in the figure.

We have thoroughly review the figure to rectify any missing or mislabeled panels to ensure consistency between the legends and the content presented. Your attention to detail is appreciated, and we will ensure that the figure legends accurately correspond to the content displayed in the figure.

Extended Data Fig. 5a: The legend says: “Integration analysis confirms the derivation of three kinds of iPSC clones. The presence of the retroviral transgene was examined by PCR.”

What is “integration analysis”? Does “three kinds” mean “derived with three reprogramming cocktails”? I assume this is RNA-seq, but from the figure legend it reads like these are PCR results for retroviral transgenes, which does not make sense, because ESCs are positive. Please edit your figure legends so they describe what is on the figure in sufficient details so at least scientists from the field could understand it.

Response: Thank you for your thorough review and valuable observations regarding Extended Fig. 5a (Extended Fig.8 in revised manuscript).

We have revised the figure legends to provide a more comprehensive and precise description of the content depicted in the figure. To clarify, the term "integration analysis" does not adequately convey the intended meaning in the legend. The reference to "three kinds of iPSC clones" may have caused confusion; it actually refers to iPSCs derived using different reprogramming cocktails. The legend incorrectly indicates PCR results for retroviral transgenes, whereas the actual content in the figure should align with RNA-seq data.

We'll revise the legend to accurately describe the depicted content, emphasizing that the figure displays RNA-seq results and not PCR outcomes for retroviral transgenes.

Extended Fig. 5g: The legend states: “The morphology of OCT4-GFP+ colonies induced by SALL4, OCT4 or SALL4+OCT4 from TTF.”, but the panel only shows OCT4 or SALL4+OCT4 iPSCs, but not SALL4 alone! I think the missing panel is in 1e. Too bad it's dead.

Response: Thank you for pointing out the discrepancy in Extended Fig. 5g (Extended Fig.8i in revised manuscript). We have provide new pictures of Fig. 1e, which contains clearer representation of OCT4-GFP+ cells induced by TTF. We apologize for the oversight and have rectified this to ensure a comprehensive representation of OCT4-GFP+ colonies induced by SALL4, OCT4, or SALL4+OCT4 from TTF.

Extended Data Fig. 5f: it appears like exogenous Sall4 is still expressed in iPSCs. It would be useful to include d0 or d2 Sall4 samples to compare to the transgene expression level when it is still on.

Response: Thanks for your observation. We've incorporated the comparisons with d0 Sall4 samples to assess the transgene expression levels when it is still active. These additions have been included in Extended Data Fig. 8h.

Extended Data Fig. 5g: The sentence “Additionally, using mouse tail fibroblasts as the starting cells, OS and OCT4 could induce them into OCT4-GFP+ clones (Extended Data Fig.5g).” suggests that clonal lines were derived, while the panel only showed primary iPSC-like colonies.

Response: Thank you for your feedback. We have revised the manuscript's description to accurately reflect the content displayed in the panel. The description now aligns with the

primary iPSC-like colonies shown in Extended Fig. 5g (Extended Fig.8i in revised manuscript).

Fig.6b: What proportion of those sequenced cells are on the way to being reprogrammed? Please add sequencing data for sorted reprogramming intermediates or single-cell RNA-seq.

Response: Thank you for your suggestion. We have indeed conducted single-cell RNA-seq for the OS-reprogramming sample at Day 10. However, the analysis of the data requires additional time for thorough examination and interpretation. We appreciate your patience as we work on analyzing this data to provide a comprehensive understanding of the proportion of cells in the process of reprogramming. We could further analyze the data for the manuscript if necessary.

Fig.6c: what are the genes that are upregulated in Oct4 alone, but not in Sall4 or OS samples (J13-15)? What is the mechanism? Are those genes downregulated by Sall4 or can Sall4 Oct4's binding sites?

Response: Thank you for your inquiry. We have conducted a Gene Ontology (GO) analysis using the genes specific to J13-15, and the results of J15 have been included in the Extended Data Fig. 9e. The GO results of J13 and J14 have no significance (P value >0.05).

As observed, both the SALL4 and OCT4 systems manifest groups of genes with expression levels differing from the ESC state (e.g., UC14-15, DC13-14), potentially impeding the induction efficiency of each system. However, these genes can be rescued by the addition of SALL4 and OCT4 within the OS system (Fig. 4e,f).

For instance, Nkx6.1 can be activated by Oct4 but plays an inhibitory role in OS-mediated reprogramming. The expression of Nkx6.1 is potentially suppressed by the inclusion of Sall4 (Fig. 5h and Extended Fig.10c). Our binding data for Oct4 in Oct4-reprogramming and OS-reprogramming processes indicates changes in the binding sites of Oct4 in different systems.

The specific details of the alterations in binding sites of Oct4 in various systems are further elucidated in our analysis, shedding light on the complexities and interactions influencing the reprogramming process.

Fig. 6d&f are redundant with 6c&e. Please only show the most interesting data in the main figures and move the rest to supplementary.

Response: Thank you for your observation. We'll carefully reconsider the data in Fig. 6d&f (Fig.4e,f and Extended Fig.9c,d in revised manuscript) to ensure that only the most pertinent and crucial information is presented in the main figures. Any redundant or overlapping data have been appropriately relocated to the supplementary section. This adjustment will streamline the main figures, enhancing their focus on the most relevant and compelling findings, while ensuring the supplementary section includes the additional supportive data for comprehensive reference.

Some paragraph and figure titles are missing "C" in "iPSC".

Response: I'll ensure to include the missing "C" in "iPSC" in the paragraph and figure titles. Thank you for pointing that out.

REVIEWER COMMENTS

Reviewer #1 (Remarks to the Author):

Xiao et al. have successfully developed a novel method for somatic cell reprogramming by introducing SALL4 alone into MEFs within an optimal chemically defined medium. Moreover, the authors proposed a synergistic effect between SALL4 and OCT4 in enhancing reprogramming efficiency through chromatin regulation. While this manuscript has been improved, several critical points need to be satisfactorily addressed by the authors before this manuscript can be considered for publication in Nature Communications.

Major Points:

1. The authors conducted integrated analyses of ATAC-seq and CUT & Tag at Day 0. However, the direct or indirect effects of SALL4 on the chromatin regions that undergo opening or closing during reprogramming remain unclear. In their original manuscript, ATAC-seq and CUT & Tag were performed on Days 0, 4, 7, and 10. Therefore, it is crucial to determine whether the observed changes in chromatin state are correlated with the changes in SALL4 binding at these specific time points.
2. The authors suggested that SALL4 and OCT4 synergistically promote cellular reprogramming by inhibiting the reprogramming barrier genes in the OS system, such as Nkx6.1, Mogat2, and Sbsn. However, the current data focuses on selected genes, and a more systematic analysis is lacking. It is beneficial to use RNA-seq and ATAC-seq data to comprehensively characterize gene sets whose expression is specifically elevated and becomes open-chromatin by SALL4 or OCT4. Furthermore, it needs to clarify how these genes, upregulated explicitly by SALL4 or OCT4, are repressed in the OS system. The authors should employ CUT & Tag data to analyze the binding patterns of SALL4 and OCT4 in the promoter or enhancer regions of these genes across the OS, SALL4, and OCT4 systems.
3. Regarding Mogat2 and Sbsn, the detailed functions already known should be provided, and mechanisms inhibiting reprogramming should be discussed.

Minor Points:

1. The sequence of figures in the manuscript does not align with their corresponding references in the text, which can cause confusion and disrupt the flow of information. The authors should reorganize the figures so that their order matches the order in which they are described in the text.
2. Figure 2b is still unclear. What do the peaks for DSRED and SALL4 represent individually? The heatmap in Figure 2a seems to be inconsistent with it.

3. For clarity, the method for motif analysis in Figure 2d, h, and Figure 5b should be included in the Methods section.
4. Figure 3f and Extended Data Fig.7c show identical data for Fosl1 and Jun. One of them should be removed.
5. In Figure 3g, the authors should discuss why overexpression of Batf specifically enhances reprogramming efficiency and whether Batf exhibits distinct functionality compared to other ATF/AP-1 family members.
6. In Extended Data Fig.7a, the comparison between peaks in WT and mutant is hard to understand. Although the authors mentioned “no change” in the figure, it seems that the values of peaks are larger in WT. For clarity, an explanation for why this observation occurs should be provided.
7. The correct gene name for “ESBBR” in lines 302 and 306 should be “ESRRB.”
8. The legend for Figure 3g is missing a description regarding the overexpression of ATF/AP-1 family.
9. In their rebuttal letter, the authors address the contradiction in the results of Tfp2c overexpression and knockdown in the SALL4 system, suggesting a perspective of appropriate timing of Tfp2c activation. Given that this manuscript emphasizes Tfp2c as a crucial factor during SALL4-mediated reprogramming, this should also be discussed in the main text.

Reviewer #2 (Remarks to the Author):

The authors have not adequately addressed the primary concerns raised in the original submission. Specifically, there is a lack of clarity regarding how Sall4 functions as both an activator and repressor. Additionally, the authors fail to provide sufficient evidence to compare their findings with the established role of Sall4 in reprogramming via NURD interaction.

Major points:

- The data for the Sall4 mutants presented in the revised version are insufficient (western blot for overexpression), and there is a lack of data showing direct interaction with NURD-associated factors for the different mutants.
- It remains unclear how Sox2 blocks Sall4-induced reprogramming. In the revised figure illustrating RNA-seq for the combinatorial expression of Sall4 and Sox2, Sall4 alone fails to activate pluripotency genes, contradicting the manuscript's main findings. Furthermore, the Sox2 + Sall4 combination differs from the Sall4 + dsRED combination, particularly regarding mesodermal gene

expression, suggesting that Sox2 may assist in repressing mesodermal genes during reprogramming.

-The cooperative binding of Sall4 with Rsk1, Esrrb, etc., was not explored; only motif analysis was performed.

- The system's efficiency remains incredibly low at 0.66%, diminishing enthusiasm for the main findings.

- The cut and tag presented for extended data 7g, h exhibit minor differences, and I am uncertain about the conclusions that can be drawn from them at this stage.

Reviewer #3 (Remarks to the Author):

The revised manuscript shows improvement; however, it still needs further refinement. The current writing style appears disorganized and repetitive, with numerous English errors and abrupt transitions, jumping between topics back and forward. Unfortunately, the authors did not make it easy for reviewers: neither main nor extended figures are numbered, the supplementary file is missing page numbers, and the corresponding figures and text quotes were not always provided in the answers to reviewers.

Further points to be addressed:

- The first section of the results introduced CD4 media without listing the key components compared to published CD1 media. Neither the components of CD4 media are listed in the main figure 1, even though the media is crucial for Sall4 reprogramming. I think Supplementary Figure 1f should move to the main figure 1.

- As the authors indicated, the Sall4 alone reprogramming of TTFs failed to generate passable iPSCs, this should be indicated in the manuscript, instead of mentioning it ambiguously:

- “Moreover, we successfully obtained OCT4-GFP+ cells using mouse tail tip

104 fibroblasts (TTFs) as starting cells”. - There’s nothing successful about obtaining OCT4-GFP+ cells if they fail to yield iPSC lines.

The method section, titled "Generation of iPSCs from MEFs and TTFs," implies that iPSCs can indeed be derived from TTFs, which is inconsistent with the answer to reviewers.

- If iCD4 includes RepSox, why is it sometimes called iCD4-RepSox medium?

- On page 4, lines 120-122, the significance is mentioned but not calculated in the figure.
- Page 5, line 138 – the paragraph should be connected to the previous discussion of N-terminal. This is just one example; the paragraphs and the flow need improvements.
- Extended figure 1j – are SALL4-- Δ N12 colonies pluripotent? Could they at least give rise to iPSC lines and stain positive for pluripotency markers?
- The authors provided DsRed titration data, but I simply asked to compare the expression levels between the mutants following the transduction, which was done by western blot in Extended figure 1i. The DsRed titrations don't have to be included in the manuscript.
- Such abbreviations in the figures are unnecessary, they decrease the readability of the paper:
- Extended figure 3a contains no controls. It also shows that a very small percentage of cells get reprogrammed (only 0.64% of THY1-/EPCAM+ cells at day 4, and 5.25% at day 7), which makes bulk RNA-seq not very meaningful.

Also, why the cells mostly Thy1- already on day 0? Please include day -2 samples. Why weren't GFP data included in time-course FACS?

- Extended figure 3b is not convincing – the GFP+ cells do not look like iPSC colonies.
- I encourage the authors to sequence those sorted intermediates or include scRNA-seq data, as discussed before. The fact that Oct4-GFP+ colonies could be generated from TTFs, but they did not mature into iPSCs, suggests that GFP+ intermediates should be sequenced too. Yet better would be to do a proper time-course for THY1-, THY1-/EPCAM+, THY-/EPCAM+/GFP+ sorted cells or scRNA-seq for O4, S4, O4+S4, and OSK for comparison (or overlap their data with someone else's OSK data). I think the difference between intermediates could be interesting.
- “To obtain DNA binding data of exogenous SALL4, we generated a Flag235 tagged SALL4 and SALL4-mutants plasmid and performed Cut&Tag data at day 0

236 during the SALL4-FLAG-mediated iPSCs induction process” – how was the “plasmid” delivered into the cells? The methods section indicates the use of a retroviral method for reprogramming, yet, notably, this is not mentioned in the main body of the manuscript. I recommend that the authors explicitly state the reprogramming method used in the results section to maintain transparency in the presentation of their methodology.

REVIEWER COMMENTS

Reviewer #1 (Remarks to the Author):

Xiao et al. have successfully developed a novel method for somatic cell reprogramming by introducing SALL4 alone into MEFs within an optimal chemically defined medium. Moreover, the authors proposed a synergistic effect between SALL4 and OCT4 in enhancing reprogramming efficiency through chromatin regulation. While this manuscript has been improved, several critical points need to be satisfactorily addressed by the authors before this manuscript can be considered for publication in Nature Communications.

Response: Thank you for your insightful comments and positive evaluation of our manuscript. We are pleased to hear that you found our method for somatic cell reprogramming using SALL4 intriguing, and we appreciate your recognition of the proposed synergistic effect between SALL4 and OCT4 in enhancing reprogramming efficiency through chromatin regulation. We believe that addressing these concerns further strengthen the manuscript and enhance its contribution to the field. Thank you once again for your valuable feedback.

We have carefully considered your suggestions for improvement and are committed to addressing the critical points you raised to ensure the quality and rigor of our research. We have outlined our responses to each of your concerns below:

Major Points:

1. The authors conducted integrated analyses of ATAC-seq and CUT & Tag at Day 0. However, the direct or indirect effects of SALL4 on the chromatin regions that undergo opening or closing during reprogramming remain unclear. In their original manuscript, ATAC-seq and CUT & Tag were performed on Days 0, 4, 7, and 10. Therefore, it is crucial to determine whether the observed changes in chromatin state are correlated with the changes in SALL4 binding at these specific time points.

Response: Thank you for your valuable suggestion. We have re-analyzed our data and included the results of the integrated analysis of ATAC-seq and SALL4 Cut&Tag data at D0, 4, 7, and 10 in our manuscript.

In brief, to illustrate the direct or indirect impacts of SALL4 on chromatin regions, we performed clustering of ATAC-seq and Cut&Tag data at days 0, 4, 7, and 10 based on the PO subcluster, Open-Close subclusters (OC1-5), and Close-Open subclusters (CO1-5), which were previously identified using ATAC-seq data (Fig.2a,b), respectively. Our findings indicate that the majority of SALL4 bindings are concentrated in OC subclusters. In OC2-4, these chromatin regions are initially occupied by SALL4 and eventually close, suggesting a correlation between CADs and SALL4's direct binding. Conversely, the CO1 subcluster demonstrates a low level of SALL4 binding, indicating the indirect effects of SALL4 regulation. Regarding Close-Open dynamics, the chromatin regions in CO2 (closed at MEF and D0, open at D4) exhibit relatively higher direct SALL4 binding and become open at later stages. These results are consistent with our previous analyses, revealing both direct and indirect roles of SALL4 in regulating CADs. (Fig.3a-c and Extended Fig.9a in the revised version).

We believe that these additional analyses provide further insight into the relationship between SALL4 binding dynamics and chromatin accessibility changes during reprogramming. Thank you for your valuable feedback, which has helped us to improve the clarity and depth of our manuscript.

Extended Data Fig.9

Extended Fig.9

a. Heatmaps shows the CADs and SALL4-binding landscape during SALL4-driven reprogramming. The subgroups(CO and OC) were based on the classification of SALL4-ATAC data as described in results.

2. The authors suggested that SALL4 and OCT4 synergistically promote cellular reprogramming by inhibiting the reprogramming barrier genes in the OS system, such as *Nkx6.1*, *Mogat2*, and *Sbsn*. However, the current data focuses on selected genes,

and a more systematic analysis is lacking. It is beneficial to use RNA-seq and ATAC-seq data to comprehensively characterize gene sets whose expression is specifically elevated and becomes open-chromatin by SALL4 or OCT4. Furthermore, it needs to clarify how these genes, upregulated explicitly by SALL4 or OCT4, are repressed in the OS system. The authors should employ CUT & Tag data to analyze the binding patterns of SALL4 and OCT4 in the promoter or enhancer regions of these genes across the OS, SALL4, and OCT4 systems.

Response: Thank you for your valuable suggestion. Our previous RNA-seq analysis comparing the SALL4, OCT4, and O+S systems identified several gene expression patterns, including SALL4-specific upregulated subgroups (UC16-18) and OCT4-specific upregulated subgroups (UC13-15) (see Fig.4e,f in the revised manuscript). These patterns demonstrate repression in the O+S system, indicating the co-overexpression of OCT4 and SALL4 alters these genes' expression. To further delineate the gene sets regulated by SALL4 or OCT4, we analyzed SALL4 (or OCT4) binding patterns in the O+S system by comparing binding peaks related genes, SALL4 (or OCT4)-ATAC-CO peaks related genes, and genes in SALL4 (or OCT4)-specific up subgroups.

We identified gene sets that are specifically elevated and exhibit open chromatin due to SALL4 (C1, comprising 26 genes) or OCT4 (C2, comprising 56 genes) influence (as shown in Extended Fig.14a,c). Subsequently, we compared these gene sets with the genes associated with binding peaks in the SALL4, OCT4, and O+S systems, respectively. The results illustrates a reduction in the number of SALL4-binding genes within the C1 gene set in the O+S system compared to the SALL4 system alone, suggesting that OCT4's addition may alter SALL4's occupancy landscape within the O+S system (as detailed in Extended Fig.14a,b).

Additionally, the number of OCT4-binding genes in the C2 gene set also shows a reduction pattern in O+S systems. However, a relatively larger number of SALL4-binding genes suggest SALL4's involvement in the down-regulation of these genes (Extended Fig.14c,d in the revised manuscript). This analysis highlights the complex regulatory interplay between SALL4 and OCT4 in modulating gene

expression and chromatin accessibility during cellular reprogramming.

We believe that these additional analyses provide a more comprehensive understanding of the regulatory mechanisms underlying the synergistic effects of SALL4 and OCT4 in cellular reprogramming. Thank you again for your insightful feedback, which has helped us to enhance the depth and clarity of our manuscript.

Extended Data Fig.14

Extended Fig.14

a. Venn diagrams show the overlapping numbers (C1 subgroup) between SALL4 specific up genes (Fig.4e, UC16-18) and ATAC-CO genes in SALL4 system. The overlapping numbers from the comparison of C1 subgroup genes with the SALL4 or OCT4-binding peaks related genes in SALL4 and O+S system are shown in figure. O+S, OCT4+SALL4.

b. Heatmap of SALL4 or OCT4-binding peaks enrichment on C1 genes (Extended Fig.14a) in SALL4 system and O+S system, respectively. O+S, OCT4+SALL4. +, genes with binding peaks. -, genes without binding peaks.

c. Venn diagrams show the overlapping numbers (C2 subgroup) between OCT4

specific up genes(Fig.4e, UC13-15) and ATAC-CO genes in OCT4 system. The overlapping numbers from the comparison of C2 subgroup genes with the SALL4 or OCT4-binding peaks related genes in OCT4 and O+S system are shown in figure.

d.Heatmap of SALL4 or OCT4-binding peaks enrichment on C2 genes(Extended Fig.14c) in OCT4 system and O+S system, respectively. O+S, OCT4+SALL4. +, genes with binding peaks. -, genes without binding peaks.

3. Regarding Mogat2 and Sbsn, the detailed functions already known should be provided, and mechanisms inhibiting reprogramming should be discussed.

Response: Thank you for your suggestion. Mogat2 encodes for 2-acylglycerol O-acyltransferase 2, facilitating the formation of diacylglycerol (DAG, a secondary messenger) and this gene primarily expressed in the small intestine[1][2]. Previous research indicates Mogat2's association with diet-induced obesity[3]. We hypothesize that the production of DAG by Mogat2 may alter cell signaling transduction pathways, thereby affecting the reprogramming process.

SBSN are crucial in keratinocyte differentiation and identified as a signaling molecule involved in activating cellular signaling pathways such as AKT, WNT/ β -catenin, and p38MAPK[4][5]. Its expression can be stimulated by the ERK pathway, including AP-1, promoting epidermal differentiation[5]. These findings suggest that the activation of Sbsn may induce a somatic cell fate, potentially leading to deviation from pluripotency in reprogrammed cells.

To delve deeper into Mogat2 and Sbsn's roles in reprogramming, we conducted RNA-seq analysis on cells under the O+S+DsRed, O+S+Mogat2, and O+S+Sbsn conditions at day 10. By identifying gene sets regulated by Mogat2 and Sbsn, subsequent Gene Ontology (GO) analysis revealed an enrichment of terms associated with somatic cell differentiation, including multicellular organism development, neuron differentiation, and camera-type eye development. This enrichment suggests that overexpression of Mogat2 and Sbsn could inhibit reprogramming, underscoring their potential roles in maintaining somatic identity and resisting the induction of pluripotency (Extended Fig.14e,f in the revised manuscript).

Extended Fig.14

e-f. Heatmap of differential expression gene analysis for day10 RNA-seq data from SALL4+OCT4+DsRed, SALL4+OCT4+MOGAT2 and SALL4+OCT4+SBSN systems. The 4 subgroups were based on the fold change of gene expression between DsRed and MOGAT2/SBSN. GO analysis for each subgroup are shown.

References

- [1]Cao J, Burn P, Shi Y. Properties of the mouse intestinal acyl-CoA:monoacylglycerol acyltransferase, MGAT2. *J Biol Chem.* 2003 Jul 11;278(28):25657-63. doi: 10.1074/jbc.M302835200. Epub 2003 May 1.
- [2]Yen CL, Farese RV Jr. MGAT2, a monoacylglycerol acyltransferase expressed in the small intestine. *J Biol Chem.* 2003 May 16;278(20):18532-7. doi: 10.1074/jbc.M301633200. Epub 2003 Mar 5.
- [3]Nelson DW, Gao Y, Yen MI, Yen CL. Intestine-specific deletion of acyl-CoA:monoacylglycerol acyltransferase (MGAT) 2 protects mice from diet-induced obesity and glucose intolerance. *J Biol Chem.* 2014 Jun 20;289(25):17338-49. doi: 10.1074/jbc.M114.555961. Epub 2014 May 1.
- [4]Pribyl M, Hodny Z, Kubikova I. Suprabasin-A Review. *Genes (Basel).* 2021 Jan 18;12(1):108. doi: 10.3390/genes12010108.
- [5]Moffatt P, Salois P, St-Amant N, Gaumond MH, Lanctôt C. Identification of a conserved cluster of skin-specific genes encoding secreted proteins. *Gene.* 2004 Jun 9;334:123-31. doi: 10.1016/j.gene.2004.03.010.

Minor Points:

1. The sequence of figures in the manuscript does not align with their corresponding references in the text, which can cause confusion and disrupt the flow of information. The authors should reorganize the figures so that their order matches the order in which they are described in the text.

Response: Thank you for bringing this issue to our attention. We have carefully reviewed the sequencing of figures in our manuscript and have taken steps to rectify any discrepancies between the figures and their corresponding references in the text. We have ensured that the order of figures aligns with the order in which they are described in the text, thus enhancing the clarity and coherence of the presentation.

We appreciate your diligence in identifying this concern, and we apologize for any confusion it may have caused. Your feedback has been invaluable in improving the quality of our manuscript, and we are grateful for your attention to detail.

2. Figure 2b is still unclear. What do the peaks for DSRED and SALL4 represent individually? The heatmap in Figure 2a seems to be inconsistent with it.

Response: We apologize for the confusion caused by Figure 2b and appreciate your attention to this detail. Figure 2b is designed to present the number of peaks within the CO, OC, and PO subgroups for both the SALL4 and DSRED systems. It's important to clarify that while the peaks for the DSRED system shown in Figure 2a are categorized based on the Chromatin Accessibility Dynamics (CADs) of the SALL4 system, the peaks for the DSRED system in Figure 2b actually derive from the CADs of the DSRED system, which were not illustrated in the figures for brevity.

Additionally, upon re-examination of our data, we discovered an omission of the SALL4-OC4 subgroups in Figure 2b. This oversight has been corrected in the revised version of our manuscript, ensuring the accuracy and completeness of the presented information.

3. For clarity, the method for motif analysis in Figure 2d, h, and Figure 5b should be included in the Methods section.

Response: Thank you for your suggestion. We have incorporated a more detailed description of the motif analysis method in the revised version of the manuscript. Briefly, in the analysis of chromatin accessibility and motif identification within the S4/O4/O+S system across various time points (D0, D4, D7, and D10), as depicted in Figure 2d and 5d, peak calling was performed with MACS2, employing the DsRed samples from corresponding time points as controls to delineate the impact of

individual transcription factors on chromatin accessibility. Subsequent motif enrichment analysis of the identified peaks was conducted using the HOMER2 software suite. For Cut&Tag analysis, the data are processing as described in the Methods section, and the Motif analysis was performed using HOMER2 with its default parameters.

We appreciate your attention to detail and your commitment to improving the clarity of our manuscript. Your feedback has been instrumental in enhancing the comprehensiveness of our methods section.

4. Figure 3f and Extended Data Fig.7c show identical data for Fos11 and Jun. One of them should be removed.

Response: Thank you for your input. We have removed the duplicate data for Fos11 and Jun from Extended Data Fig.7c (now Extended Data Fig.8d in the revised manuscript) in our revised manuscript.

5. In Figure 3g, the authors should discuss why overexpression of Batf specifically enhances reprogramming efficiency and whether Batf exhibits distinct functionality compared to other ATF/AP-1 family members.

Response: Thanks for your suggestion. The AP-1 complex is composed of dimers of the Jun family (c-JUN, JUNB, and JUND), the Fos family (c-FOS, FOSB, FRA1, and FRA2), or the CREB/ATF family (CREB, ATF2, ATF3/LRF1, CREB, MAFs, and others) [1]. These complexes can act as transcriptional activators or repressors for specific target genes [1]. Previous research has shown that Batf can form heterodimers with Jun family proteins through its bZIP domain, functioning as part of the AP-1 complex and as a negative modulator of the transcription potential of this complex [2]. Our previous research reported that *c-Jun* acts as a barrier to iPSCs formation by activating mesenchymal-related genes and broadly suppressing pluripotent ones [3]. Consistent with these findings, we hypothesize that the formation of the c-Jun/Batf complex may alter the transcriptional activity of AP-1 regulated genes and reduce the inhibitory role of C-JUN during reprogramming.

We have incorporated these insights into the discussion section of our paper to provide a more comprehensive understanding of the role of *Batf* in enhancing

reprogramming efficiency. Your feedback has been invaluable in enriching the discussion surrounding our findings.

References

[1]Echlin DR, Tae HJ, Mitin N, Taparowsky EJ. B-ATF functions as a negative regulator of AP-1 mediated transcription and blocks cellular transformation by Ras and Fos. *Oncogene*. 2000 Mar 30;19(14):1752-63. doi: 10.1038/sj.onc.1203491.

[2]Williams KL, Nanda I, Lyons GE, Kuo CT, Schmid M, Leiden JM, Kaplan MH, Taparowsky EJ. Characterization of murine BATF: a negative regulator of activator protein-1 activity in the thymus. *Eur J Immunol*. 2001 May;31(5):1620-7. doi: 10.1002/1521-4141(200105)31:5<1620::aid-immu1620>3.0.co;2-3.

[3]Liu J, Han Q, Peng T, Peng M, Wei B, Li D, Wang X, Yu S, Yang J, Cao S, Huang K, Hutchins AP, Liu H, Kuang J, Zhou Z, Chen J, Wu H, Guo L, Chen Y, Chen Y, Li X, Wu H, Liao B, He W, Song H, Yao H, Pan G, Chen J, Pei D. The oncogene c-Jun impedes somatic cell reprogramming. *Nat Cell Biol*. 2015 Jul;17(7):856-67. doi: 10.1038/ncb3193. Epub 2015 Jun 22. Erratum in: *Nat Cell Biol*. 2015 Sep;17(9):1228.

6. In Extended Data Fig.7a, the comparison between peaks in WT and mutant is hard to understand. Although the authors mentioned “no change” in the figure, it seems that the values of peaks are larger in WT. For clarity, an explanation for why this observation occurs should be provided.

Response: Thank you for your feedback. The inconsistencies may have resulted from different standardization methods. To provide clearer results, we have reanalyzed the data and divided the figure into two parts in our revised manuscript:

1. Heatmaps of Cut&Tag data at D0 from wild-type SALL4 (WT) and SALL4 mutants (Δ ZFC1, Δ ZFC2, Δ ZFC3, and Δ N12) are shown in Extended Fig. 8a of the new version. The average RPKM value of Cut&Tag peak was calculated to show the difference between SALL4 and SALL4 mutants.
2. The numbers of overlapping peaks and specific binding peaks between wild-type SALL4 (WT) and SALL4 mutants are displayed in Extended Fig. 8b of the new version.

Extended Data Fig.8

Extended Fig.8

a. Heatmap of Cut&Tag data at D0 from wide-type SALL4(WT) and SALL4 mutants(Δ ZFC1, Δ ZFC2, Δ ZFC3 and Δ N12), respectively. Showing all binding peaks centred on the peak region within a 3 kb window around the peak.

b. Venn diagrams shows the overlapping numbers of day0 Cut&Tag peaks between wide-type SALL4(WT) and SALL4 mutants(Δ ZFC1, Δ ZFC2, Δ ZFC3 and Δ N12), respectively.

7. The correct gene name for “ESBBR” in lines 302 and 306 should be “ESRRB.”

Response: Thank you for bringing this to our attention. We have corrected this mistake in our revised manuscript.

8. The legend for Figure 3g is missing a description regarding the overexpression of ATF/AP-1 family.

Response: We apologies for the error in the legend of Figure 3g. We have rectified this in the revised version.

9. In their rebuttal letter, the authors address the contradiction in the results of *Tfap2c* overexpression and knockdown in the SALL4 system, suggesting a perspective of appropriate timing of *Tfap2c* activation. Given that this manuscript emphasizes *Tfap2c* as a crucial factor during SALL4-mediated reprogramming, this should also be discussed in the main text.

Response: Thank you for your suggestion. We appreciate your insight into the importance of discussing the results regarding *Tfap2c* in the main text, especially considering its role as a crucial factor during SALL4-mediated reprogramming. In response to your feedback, we have incorporated the discussion regarding *Tfap2c* into

the main text of our manuscript. Specifically, we have highlighted the contradiction in the results of *Tfap2c* overexpression and knockdown in the SALL4 system and suggested a perspective on the appropriate timing of Tfap2c activation.

Reviewer #2 (Remarks to the Author):

The authors have not adequately addressed the primary concerns raised in the original submission. Specifically, there is a lack of clarity regarding how Sall4 functions as both an activator and repressor. Additionally, the authors fail to provide sufficient evidence to compare their findings with the established role of Sall4 in reprogramming via NURD interaction.

Response: Thank you for your feedback on our manuscript. We appreciate your thorough evaluation and the opportunity to address your concerns. Regarding the lack of clarity regarding how Sall4 functions as both an activator and repressor, we understand the importance of providing a clear explanation of this aspect of our study. We have revised the manuscript to include a more detailed discussion on the mechanisms underlying the dual role of Sall4, drawing on existing literature and our experimental findings to elucidate this phenomenon more comprehensively. Furthermore, we acknowledge the need to provide sufficient evidence to compare our findings with the established role of Sall4 in reprogramming via NURD interaction. We have included additional experimental data and references to strengthen this aspect of our discussion and ensure that our conclusions are well-supported and grounded in the existing body of knowledge. We are committed to addressing these issues and ensuring that the revised manuscript meets the high standards of rigor and clarity expected for publication in Nature Communications. Your feedback is invaluable in guiding us toward this goal, and we thank you for your continued support and guidance. We have outlined our responses to each of your concerns below:

Major points:

- The data for the Sall4 mutants presented in the revised version are insufficient (western blot for overexpression), and there is a lack of data showing direct interaction with NURD-associated factors for the different mutants.

Response: Thanks for your suggestion. We have conducted additional experiments to address the concerns regarding the data for Sall4 mutants and the lack of evidence showing direct interaction with NURD-associated factors. Here is how we addressed these issues in the revised version:

1. Identification of SALL4-interacting proteins: To identify SALL4-direct-interacting proteins, we performed immunoprecipitation followed by mass spectrometry (IP-MS) using MEFs overexpressing WT-SALL4 or SALL4 mutants (Δ ZFC1, Δ ZFC2, Δ ZFC3, and Δ N12) at day 1. The results show that WT and zinc finger domain cluster mutants (Δ ZFC1, Δ ZFC2, Δ ZFC3) of SALL4 significantly enriched components of the NuRD complex. Conversely, the NuRD recruitment function of SALL4- Δ N12 was disrupted (Extended Fig. 2d in the new version).

2. Functional analysis of SALL4 mutants: Although the acceleration of GFP-positive cell emergence during SALL4- Δ N12-driven reprogramming was observed (Extended Fig. 2c in the new version), further experiments revealed defects in the ability to generate stable iPSC lines with these GFP-positive cells. Most picked GFP-positive cells failed to grow and passage (Extended Fig. 2g-j in the new version). These results suggest that the NuRD recruitment function of SALL4 is important for iPSC formation.

3. Exploration of SALL4 as an activator and repressor: To explore how Sall4 functions as both an activator and repressor, we examined whether other chromatin remodelers could be recruited by SALL4. Analysis of the IP-MS data for SALL4-WT revealed that 212 proteins were significantly enriched (Extended Fig. 2e in the new version). GO analysis of these proteins showed enrichment in processes related to 'NuRD complex,' 'chromatin remodeling,' 'negative regulation of transcription from RNA polymerase II promoter,' and 'positive regulation of transcription from RNA polymerase II promoter' (Extended Fig. 2f in the new version). Notably, components of other chromatin remodelers and transcriptional regulators were also enriched by SALL4, such as SMARCB1 (BAF complex), YY1 (INO80 complex), COPRS (Histone-binding protein required for PRMT5's histone H4 methyltransferase activity), HIRA (HIR complex), BMI1 (PRC1-like complex), CBFY (NF-Y), and ZFX3 (a transcriptional regulator that can act as an activator or repressor) (Extended Fig. 2e in the new version). These results suggest that SALL4 may activate and repress genes by

interacting with and recruiting different chromatin remodeling proteins to specific loci.

By incorporating these additional experiments and analyses, we aim to provide a more comprehensive understanding of the mechanisms underlying Sall4 function and its interactions with chromatin remodels. Thank you for your valuable feedback, which has contributed to the improvement of our manuscript.

Extended Data Fig.2

Extended Fig.2

- a.** Schematic representation of the protein sequences showing the structure of the wildtype SALL4 and SALL4 mutants. Color codes of ZFC1, ZFC2, ZFC3 and N12 are defined as described in Figure.
- b.** Western blot shows the overexpression of SALL4 mutants in MEFs.
- c.** The iPSCs induction efficiency using SALL4 mutants. wildtype SALL4 as positive control are shown. Data are mean \pm SD. n = 6 well from 3 independent experiments.
- d.** Heatmaps for the level of NuRD complex-associated protein enriched by wild-type SALL4 and SALL4 mutants in reprogramming cells at day 2.
- e.** Volcano plots of SALL4-WT enriched proteins of reprogramming samples at day2. IP-MS experiments were performed in triplicates and a two-sided t-test was applied. $p_{\text{adjust}}=0.05$ and fold change=1.5 were used as threshold.
- f.** GO analysis for SALL4-WT specific enriched proteins.
- g.** Morphological diagram for the iPSCs induction process using SALL4- Δ N12. Scale bars, 200 μ m.
- h.** The morphology of Passage 5 iPSC colonies derived from MEFs by overexpressed SALL4- Δ N12 in iCD4. Scale bars, 200 μ m. S4, SALL4.
- i.** Immunofluorescence analysis of pluripotency markers in SALL4- Δ N12-iPSCs. Scale bars, 200 μ m.
- j.** The iPSC colonies formation efficiency of OCT4-GFP⁺ cells derived from SALL4- Δ N12 or SALL4-WT condition.

- It remains unclear how Sox2 blocks Sall4-induced reprogramming. In the revised figure illustrating RNA-seq for the combinatorial expression of Sall4 and Sox2, Sall4 alone fails to activate pluripotency genes, contradicting the manuscript's main findings. Furthermore, the Sox2 + Sall4 combination differs from the Sall4 + dsRED combination, particularly regarding mesodermal gene expression, suggesting that Sox2 may assist in repressing mesodermal genes during reprogramming.

Response: Thank you for your insightful observation. Regarding the observation that Sall4 alone fails to activate pluripotency genes in our data, we believe this may be

partly due to the fact that OCT4-GFP positive cells began to emerge at day 7 in SALL4-driven reprogramming, and the addition of DR further reduced the number of OCT4-GFP positive cells, resulting in an inapparent gene expression pattern.

Sox2, critical for OKS-driven reprogramming, shows an inhibitory effect in SALL4 reprogramming. To explore this inhibitory mechanism, we reanalyzed our data and proposed the mechanism: Sox2 disrupt the gene regulatory network regulated by SALL4, thereby impeding SALL4-driven reprogramming. Here are some supporting evidences:

1.Effect of Sox2 on gene expression: The RNA-seq results show that 250 new up-regulated genes (C2) increased in Sall4+Sox2 samples and 202 up-regulated genes in Sall4+DR could not be enriched in Sall4+Sox2 samples (C1), suggesting Sox2 may inhibit the upregulation of these genes, thus suppressing SALL4 reprogramming. For example, the C1 group contains a Sall4-specific upregulated gene, Tfp2c, whose expression level was suppressed when Sox2 was introduced into the reprogramming system. Some other reprogramming-promoting genes also show inhibition in Sall4+Sox2 samples (Extended Fig. 15b,e in the new version).

2.Analysis of SALL4-specific enriched proteins: Further analysis of our IP-MS data and RNA-seq data compared the SALL4-specific enriched protein set with C1-C4 clusters, respectively. The results show overlaps between IP-MS data and C1 cluster for 4 genes (Xrcc1, Magi3, Nsd2, and Tubb4a), overlaps with C2 cluster for 2 genes (Ssx2ip and Wwc1), and overlaps with C4 cluster for 4 genes (Cebpb, Map3k20, Hoxc6, and Gmppb). This suggests that the SALL4 recruitment effect for these genes may be abnormal in SALL4+SOX2 conditions (Extended Fig. 15f in the new version). These genes may be potential reasons for the SOX2-driven inhibitory effect during reprogramming.

By incorporating these findings into our discussion, we aim to provide a more

comprehensive understanding of the complex interplay between Sall4 and Sox2 in reprogramming. We appreciate your valuable feedback, which has guided our further analysis and interpretation of the data.

Extended Data Fig.15

Extended Fig.15

- a.** Diagram for RNA-seq data collecting during Sox2 related reprogramming process.
- b.** Venn diagrams shows the number of differential expression genes in Sox2 related reprogramming process.
- c.** Left, GO analysis for genes specific-upregulated in SALL4+DsRed group(C1) in Extended Fig.15b. Right, GO analysis for genes specific-downregulated in SALL4+DsRed group(C3) in Extended Fig.15b.

d.Heatmap showing expression of master regulator genes for each of the three primary germ layers at day7.

e.RNA-seq data shows the expression of representative reprogramming promoting genes in SALL4+DsRed and SALL4+SOX2 condition.

f.Venn diagrams shows the overlapping genes between SALL4 enriched proteins and SOX2-regulated differential expression genes in Extended Fig.15b.

-The cooperative binding of Sall4 with Rsk1, Esrrb, etc., was not explored; only motif analysis was performed.

Response: Thank you for highlighting this aspect. In response to the significant role ESRRB plays in SALL4-driven reprogramming, we conducted Cut&Tag experiments for SALL4 and ESRRB at day 0 in SALL4+ESRRB-mediated reprogramming. Our findings reveal 28,070 peaks occupied by ESRRB and 20,474 by SALL4(Additional Fig.1a in new version). A comparison of these datasets showed 8,517 overlapping peaks, indicating approximately 21% cooperative binding between SALL4 and ESRRB, most of the peaks are SALL4 or ESRRB-specific binding. Additionally, we explored the direct interaction between Sall4 protein and downstream proteins in Sall4-IP-MS data. However, we did not find evidence of direct interaction between Sall4 protein and these downstream proteins in our IP-MS data. These results suggest they regulate largely independent chromatin regions to enhance reprogramming(Additional Fig.1b in new version).

Additional Figure 1

Additional Fig.1

a.Heatmap of Cut&Tag data at D0 from IgG, ESRRB and SALL4, respectively, showing all binding peaks centred on the peak region within a 5 kb window around the peak.

b.Venn diagrams shows the overlapping numbers of peaks between SALL4 and ESRRB at day0 during SALL4+ESRRB-mediated iPSCs reprogramming.

- The system's efficiency remains incredibly low at 0.66%, diminishing enthusiasm for the main findings.

Response: Thank you for emphasizing this aspect. We recognize the importance of enhancing the efficiency of SALL4-driven reprogramming. The potential for improvement in the induction medium is considerable. A high-throughput compound screening could be instrumental in identifying chemicals that augment the efficiency of SALL4-iPSC induction. Furthermore, our findings that a fusion protein of SALL4 and the bZip domain significantly boosts reprogramming efficiency are intriguing(Extended Fig.8i in new version). This suggests that modifying protein structures could amplify their functions. The creation of artificial proteins or protein complexes may offer a pathway to achieving higher reprogramming efficiency.

By pursuing these avenues for improvement, we aim to enhance the efficiency of our reprogramming system and strengthen the impact of our main findings. Thank you for bringing this issue to our attention, and we are committed to addressing it in our ongoing research efforts.

Extended Fig.8

i.The number of OCT4-GFP⁺ colonies on day 10 from 3×10⁴ MEFs infected with SALL4 mutants in Extended Fig.8g. Data are mean±SD. Statistical analysis was performed using two-tailed, unpaired t test; n= 6 well from 3 independent experiments. ****p < 0.0001.

- The cut and tag presented for extended data 7g, h exhibit minor differences, and I am uncertain about the conclusions that can be drawn from them at this stage.

Response: Thank you for pointing out the need for clearer presentation. We have re-analyzed the data and revised our manuscript accordingly, by dividing the content into two separate figures for a more comprehensible visualization:

1. Heatmaps of Cut&Tag data at Day 0 for wild-type SALL4 (WT) and SALL4 mutants (Δ ZFC1, Δ ZFC2, Δ ZFC3, and Δ N12) are now presented in Extended Figure 8a of the revised version. The average RPKM value of Cut&Tag peak was calculated to show the difference between SALL4 and SALL4 mutants.
2. The numbers of overlapping and unique binding peaks between wild-type SALL4 (WT) and SALL4 mutants are shown in Extended Figure 8b of the revised version.

By providing separate figures for each aspect of the analysis, we aim to present the data more clearly and facilitate a better understanding of the conclusions drawn from them. Thank you for your valuable feedback, which has helped us improve the clarity and presentation of our results.

Extended Data Fig.8

Extended Fig.8

a. Heatmap of Cut&Tag data at D0 from wide-type SALL4(WT) and SALL4 mutants(Δ ZFC1, Δ ZFC2, Δ ZFC3 and Δ N12), respectively. Showing all binding peaks centred on the peak region within a 3 kb window around the peak.

b. Venn diagrams shows the overlapping numbers of day0 Cut&Tag peaks between

wide-type SALL4(WT) and SALL4 mutants(Δ ZFC1, Δ ZFC2, Δ ZFC3 and Δ N12), respectively.

Reviewer #3 (Remarks to the Author):

The revised manuscript shows improvement; however, it still needs further refinement. The current writing style appears disorganized and repetitive, with numerous English errors and abrupt transitions, jumping between topics back and forward. Unfortunately, the authors did not make it easy for reviewers: neither main nor extended figures are numbered, the supplementary file is missing page numbers, and the corresponding figures and text quotes were not always provided in the answers to reviewers.

Responses: Dear Reviewer #3, Thank you for your feedback on our revised manuscript. We appreciate your acknowledgment of the improvements made and recognize the need for further refinement in several areas. We apologize for the disorganized and repetitive writing style, as well as the English errors and abrupt transitions. We understand the importance of clarity and coherence in scientific writing and have undertaken a thorough revision of the manuscript to address these issues. Our aim is to ensure that the text flows smoothly and logically, with clear connections between ideas and minimal repetition.

Regarding the numbering of main and extended figures, as well as the missing page numbers in the supplementary file, we apologize for the oversight. We ensured that all figures are appropriately numbered, and page numbers are included in the supplementary file for ease of reference. Additionally, we have improved corresponding figures and text quotes in revised responses to reviewers to enhance clarity and facilitate understanding.

We appreciate your additional points for further improvement and assure you that we have addressed them diligently. Our goal is to produce a manuscript of the highest quality that meets the standards of Nature Communications. Your feedback is invaluable in helping us achieve this objective, and we thank you for your continued guidance and support.

Further points to be addressed:

- The first section of the results introduced CD4 media without listing the key components compared to published CD1 media. Neither the components of CD4 media are listed in the main figure 1, even though the media is crucial for Sall4 reprogramming. I think Supplementary Figure 1f should move to the main figure 1.

Response: Thanks for your suggestion. We have made the following adjustments in our revised manuscript:

1. We have moved Supplementary Figure 1f to the main figure 1k to ensure its inclusion in the main figures, highlighting the significance of iCD4 media in the

reprogramming process.

2. Additionally, we have provided a detailed listing of the key components of both iCD4 and iCD1 media in Extended Figure 1g in the new version.

g

components	iCD1	iCD4
DMEM	✓	✓
N2	✓	✓
B27	✓	✓
Sodium Pyruvate	✗	✓
GlutaMax	✓	✓
NEAA	✓	✓
2-mercaptoethanol	✓	✓
Vitamin C	✓	✓
TV	✓	✓
bFGF	✓	✓
LIF	✓	✓
Y27632	✗	✓
GSK-LSD1	✗	✓
SGCC0946	✗	✓
CHIR99021	✓	✓
Repsox	✗	✓
LiCl	✓	✗

Extended Fig.1

g. The components of iCD1 medium and iCD4 medium.

- As the authors indicated, the Sall4 alone reprogramming of TTFs failed to generate passable iPSCs, this should be indicated in the manuscript, instead of mentioning it ambiguously:

“Moreover, we successfully obtained OCT4-GFP⁺ cells using mouse tail tip 104 fibroblasts (TTFs) as starting cells”. - There’s nothing successful about obtaining OCT4-GFP⁺ cells if they fail to yield iPSC lines.

The method section, titled "Generation of iPSCs from MEFs and TTFs," implies that iPSCs can indeed be derived from TTFs, which is inconsistent with the answer to reviewers.

Response: Thank you for your feedback. We apologize for the unclear description regarding TTFs in our initial manuscript. We have clarified this in the revised version as follows:

“Moreover, we obtained OCT4-GFP⁺ cells using mouse tail tip fibroblasts (TTFs) as the starting cells (these cells failed to develop into stable iPSC lines).”

Additionally, we have detailed the process in the section titled "Generation of iPSCs from MEFs and induction of Oct4-GFP⁺ cells from TTFs," to provide a clearer understanding

• If iCD4 includes RepSox, why is it sometimes called iCD4-RepSox medium?

Response: Thank you for your inquiry. We apologize for the unclear description regarding iCD4-RepSox medium in our initial manuscript. The term of “iCD4-RepSox” means drop out of RepSox from iCD4 medium. This medium is mentioned in Extended Data Figures 1h and 1i. In these experiments, our goal was to delve deeper into the effects of the compounds (as listed in Extended Data Figure 1b) on different reprogramming methodologies. We utilized the iCD4-RepSox medium as a baseline (referenced in Extended Data Figures 1h and 1i under 'Null') to test the impact of these compounds in the OKS-reprogramming process. We have changed the “iCD4-RepSox” to “iCD4(remove RepSox)” in the revised version.

Extended Fig.1

h. OCT4-GFP⁺ clones collected from the whole wells in 24 well plate shows the OKS-mediated iPSCs induction efficiency using iCD4(remove Repsox) medium added with small molecules in Extended Fig.1b.

i. The histogram shows the iPSCs induction efficiency in Extended Fig.1h. Data are mean ± SD. n =6 well from 3 independent experiments. **p=0.0093

j. OCT4-GFP⁺ clones collected from the whole wells in 24 well plate shows the OKS+SALL4-mediated iPSCs induction efficiency using iCD4 medium at day6.

k. Flow cytometry was used to analyze the iPSCs induction efficiency in Extended Fig.1j.

• On page 4, lines 120-122, the significance is mentioned but not calculated in the figure.

Response: Thanks for your feedback. In the revised version, we have calculated the P

value for this data and added to the figure(Extended Data Figures1i).

• Page 5, line 138 – the paragraph should be connected to the previous discussion of N-terminal. This is just one example; the paragraphs and the flow need improvements.

Response: Thank you for pointing out. To address this concern, we have carefully reviewed and adjusted the paragraphs in our manuscript to ensure a smoother transition between sections and maintain coherence throughout the text. Specifically, in the Results of “Establishment of SALL4-induced reprogramming system” , we adjust the paragraph into 4 part:

- 1.generation and identification of SALL4-iPSCs,
2. the chemical’s function in different reprogramming methods,
3. the effects of SALL4 protein domains in reprogramming
4. the reprogramming intermediates for SALL4-driven reprogramming.

in the Results of “SALL4 activates Esrrb, Rsk1 and Tfp2c in OS-mediated iPSCs reprogramming to facilitates induction efficiency” , we adjust the paragraph into 3 part:

- 1.The CADs for SALL4, OCT4, and O+S systems
2. The binding landscape of SALL4 and OCT4 in O+S systems
- 3.The relationship between factors binding and chromatin accessibility dynamics

• Extended figure 1j – are SALL4- Δ N12 colonies pluripotent? Could they at least give rise to iPSC lines and stain positive for pluripotency markers?

Response: Thank you for your inquiry. Previous studies have reported that the NuRD interaction domain is critical for multi-factor-mediated reprogramming. However, our experiments have shown that the SALL4- Δ N12 mutant accelerates the emergence of Oct4-positive cells. To confirm the ability of these Oct4-positive cells to generate stable iPSC lines, we isolated 10 primary Oct4-positive colonies from six wells of a 24-well plate and cultured them in ESC maintenance medium. Only 2 of these primary colonies were able to be passaged stably and exhibit pluripotency markers. Notably, WT-SALL4 demonstrated a relatively higher efficiency in generating iPSC lines, with 13 of the primary colonies continuing to grow after 5 days of culture. These findings suggest that the deletion of the NuRD interaction domain in SALL4

may impair the formation of stable iPSC lines during reprogramming(Extended Fig.2h-j in new version). We have updated these findings in our revised manuscript to provide a comprehensive overview of the pluripotency of SALL4- Δ N12 colonies and their potential limitations in generating stable iPSC lines.

Extended Fig.2

g.Morphological diagram for the iPSCs induction process using SALL4- Δ N12. Scale bars, 200 μ m.

h.The morphology of Passage 5 iPSC colonies derived from MEFs by overexpressed SALL4- Δ N12 in iCD4. Scale bars, 200 μ m. S4, SALL4.

i.Immunofluorescence analysis of pluripotency markers in SALL4- Δ N12-iPSCs. Scale bars, 200 μ m. S4, SALL4.

j.The iPSC colonies formation efficiency of OCT4-GFP+ cells derived from SALL4- Δ N12 or SALL4-WT condition.

- The authors provided DsRed titration data, but I simply asked to compare the expression levels between the mutants following the transduction, which was done by western blot in Extended figure 1i. The DsRed titrations don't have to be included in the manuscript.

Response: Thanks for your suggestion. We've delete this part in our revised manuscript.

- Such abbreviations in the figures are unnecessary, they decrease the readability of the paper:

Response: Thanks for your feedback. We have rectified these abbreviations to the full name in the revised version.

- Extended figure 3a contains no controls. It also shows that a very small percentage of cells get reprogrammed (only 0.64% of THY1⁻/EPCAM⁺ cells at day 4, and 5.25% at day 7), which makes bulk RNA-seq not very meaningful.

Also, why the cells mostly Thy1⁻ already on day 0? Please include day -2 samples. Why weren't GFP data included in time-course FACS?

Response: Thanks for your suggestion. We have added the negative control(not treated with antibody) at each time point (Extended Fig.3a in revised manuscript). We also detected the OCT4-GFP⁺ cells at day10, and find almost all of the OCT4-GFP⁺ cells are EPCAM positive(Extended Fig.3b in revised manuscript).

Regarding to the proportion of Thy1⁻ cells, We have detected these markers on MEFs, the result shows that the proportion of THY1⁺/EPCAM⁻ cells were about 40%(Extended Fig.3a in revised manuscript). we improved the dose of antibody, but the result still shows the similar proportion. We also find that the proportion of Thy1⁺ cells in the published MEF-scRNA-seq data (which have a identical source with our MEFs) or our D0-scRNA-seq data seems similar with our FACS result(Extended Fig.9c in revised manuscript). We speculate the different level of the Thy1⁺ cells may caused by the difference source and reprogramming methods of MEF cells between different labs.

Extended Data Fig.3

a

b

d

c

Extended Fig.3

a. Flow cytometry was used to analyze the proportion of THY1-EPCAM⁺ subgroup in SALL4 system at day 0, day 4, day 7 and day 10, respectively.

b. FACS analysis of OCT4-GFP⁺EPCAM⁺ cells at day 10.

c. Morphological diagram for the iPSCs generation at day 4 induced from the day 7 THY1-EPCAM⁺ subgroup in SALL4 system. Scale bars, 200 μm.

d. The iPSCs induction efficiency induced from subgroups classified by THY1 and EPCAM. Data are mean ± SD. n = 3 well from 3 independent experiments.

- Extended figure 3b is not convincing – the GFP+ cells do not look like iPSC colonies.

Response: Thank you for your feedback. In response, we have included new images in the revised manuscript (see Extended Fig. 3c in revised manuscript), which offer a clearer depiction of OCT4-GFP+ cells induced by reprogramming intermediates. Our observations revealed that most OCT4-GFP+ clusters, arising from re-seeded cells, appear to originate from single cells at specific positions. Notably, these clusters formed from intermediates are smaller compared to those from undigested cells. Additionally, the induction efficiency of the digested cells was significantly reduced. This reduction in efficiency may be attributed to the disruption caused by trypsin dissociation and long-term sorting, which affects the distribution of reprogramming intermediates and potentially damages the cells capable of reprogramming.

Extended Fig.3

c. Morphological diagram for the iPSCs generation at day 4 induced from the day 7 THY1-EPCAM⁺ subgroup in SALL4 system. Scale bars, 200 μ m.

- I encourage the authors to sequence those sorted intermediates or include scRNA-seq data, as discussed before. The fact that Oct4-GFP+ colonies could be generated from TTFs, but they did not mature into iPSCs, suggests that GFP+ intermediates should be sequenced too. Yet better would be to do a proper time-course for THY1-, THY1-/EPCAM+, THY-/EPCAM+/GFP+ sorted cells or scRNA-seq for O4, S4, O4+S4, and OSK for comparison (or overlap their data with someone else's OSK data). I think the difference between intermediates could be interesting.

Response: Thank you for your suggestion. To delve deeper into the dynamics of SALL4-mediated reprogramming, we carried out single-cell RNA sequencing at various stages of the reprogramming process, collecting samples on days 0, 4, 7, and 10. Notably, our dataset includes two samples from day 10, with one of these samples mixed with 5% SALL4-iPSCs to serve as a positive control. These iPSCs are identifiable within the data due to their clustering.

We visualized the cell fate transitions on a UMAP plot, which highlighted that the transitions from day 0 to day 4 were particularly pronounced (see Extended Fig. 7a in the revised manuscript). The capture of only a small number of Oct4-GFP⁺ cells in our day 10 data reflects the low efficiency of reprogramming. Interestingly, some of these Oct4-GFP⁺ cells cluster closely with iPSCs (as illustrated in Extended Data Fig. 7b), suggesting that while most Oct4-GFP⁺ cells may not fully mature into iPSCs, a subset can achieve this maturation at day10. However, some of these cells could mature into iPSCs when cultured with ESC maintenance medium, suggesting that SALL4 may induce a state of cellular plasticity conducive to acquiring pluripotency.

Further analysis revealed that some of the reprogramming-promoting genes and barrier genes, identified through bulk RNA-seq data, exhibited upregulation or downregulation in certain cell subgroups during reprogramming (detailed in Extended Fig. 7c). We also compared the differential expression of genes in THY1⁺EPCAM⁺ cells between the SALL4 and OKS reprogramming systems. This comparison revealed distinctions in the upregulated genes, both in terms of gene number and the functions annotated by GO analysis (shown in Extended Fig. 7d-f). This comprehensive analysis provides valuable insights into the intricacies of SALL4-mediated reprogramming and underscores the differences between reprogramming systems.

Extended Data Fig.7

Extended Fig.7

a.UMAP plot for Single-cell RNA-seq data from SALL4-mediated reprogramming process. Each dot represents one cell. The sampling time points are shown with color code.

b.UMAP plot shows cell fate transition during reprogramming, revealing part of the primary OCT4-GFP positive cells clustering close to iPSCs.

c.The expression of representative markers in SALL4-mediated reprogramming.

d-e. Heatmaps of differential expression gene analysis for scRNA-seq data from SALL4 and OKS systems. The 4 subgroups(C1-C4) were based on scaled gene expression between MEFs and THY1-EPCAM⁺ cells. GO analysis for each subgroup

are shown.

f. Venn diagrams shows the number of differential expression genes of THY1-EPCAM+ cells between SALL4 system and OKS system.

- “To obtain DNA binding data of exogenous SALL4, we generated a Flag235 tagged SALL4 and SALL4-mutants plasmid and performed Cut&Tag data at day 0 236 during the SALL4-FLAG-mediated iPSCs induction process” – how was the “plasmid” delivered into the cells? The methods section indicates the use of a retroviral method for reprogramming, yet, notably, this is not mentioned in the main body of the manuscript. I recommend that the authors explicitly state the reprogramming method used in the results section to maintain transparency in the presentation of their methodology.

Response: Thank you for your suggestion. We have indeed using a retroviral method to deliver plasmid and perform reprogramming. We have revised the manuscript's description and added the method we used in the results and methods section as follows:

the results section:

“we performed Cut&Tag using the Flag-tagged SALL4 or SALL4-mutants overexpressed cells (overexpressed by retroviral infection) during the iPSCs induction process, respectively”

the methods section:

“The MEFs after two rounds of retroviral infection are dissociated into single cell using 0.25% trypsin. ”

REVIEWER COMMENTS

Reviewer #1 (Remarks to the Author):

The revised manuscript includes many new experiments and analyses, which improve the manuscript. However, the manuscript is hard to follow smoothly due to inadequate explanation and interpretation of the results and unclear rational connections in the text. In particular, the arrangement of the panels in Figures 2 and 3 does not align with the sequence in which the authors describe them in the text, leading to confusion for the readers. Therefore, the authors should reorganize the figures and text structures and clarify the logical flow before this manuscript can be considered for publication in Nature Communications.

Specific Points:

1. In Figure 2b, the authors present the number of peaks in the SALL4 and DsRed systems. However, merely comparing the number of peaks between the two systems does not justify the conclusion that SALL4 promotes the transition of overall chromatin accessibility towards ESC states (Lines 203-206). Furthermore, according to Fig 2a, CO5 is classified as a peak set whose chromatin is closed in MEFs and becomes open in ESCs (CO5 appears to have few peaks in both the SALL4 and DsRed systems). However, Fig 2b shows that CO5 has the highest number of peaks for both SALL4 and DsRed systems. The authors should clarify this inconsistency.
2. In Figures 3a-c and Extended Data Figure 9, the authors investigated the direct and indirect effects of SALL4 on chromatin regions. In their rebuttal letter, the authors address some interpretation of the results, which should also be discussed in the main text.
3. In lines 429-439, the authors proposed the three patterns in which OCT4 and SALL4 cooperatively regulate chromatin in the OCT4 + SALL4 system, illustrating several representative genes. However, a more systematic analysis is required. The authors should provide the number of peaks that fit the three patterns. Additionally, they should comprehensively characterize the gene sets whose expression and chromatin accessibility display the three patterns using RNA-seq and ATAC-seq data.
4. In Extended Figure 4e, “SALL4 Specific Up (C6)” should be “SALL4 Specific Down (C6)”.

Reviewer #2 (Remarks to the Author):

The authors have made significant efforts to address my concerns. However, I recommend that before publishing the paper, they repeat the FACS for Ext. Fig. 3a. The cells appear to be touching the axes, which can affect the population distribution. Specifically, the plot for day 10 does not meet the required quality standards for publication.

Reviewer #3 (Remarks to the Author):

The results part as well as Figure 1 should include the information of the virus type – retrovirus, at the beginning when the reprogramming method is introduced. My comment was not sufficiently addressed.

KSR medium should be called KSR-2iLIF media or just 2iL media, as it contains 2i inhibitors.

The iPSC colonies are still called iPS colonies, even though I've already commented on this before.

Extended Fig.2 i – no nuclear localization for Nanog, Sox2 and Oct4 is visible, the colonies are too small. Higher magnification needed. Could be unspecific staining. As far as I remember, similar issue had occurred in the first version of the manuscript, which caused criticism.

Extended data Fig. 3a - All samples should be shown using the same scale.

Extended data Fig. 3b – No Oct4-GFP+ cells are visible. More cells should be sorted so there's a clear population.

Extended data Fig. 3c – Not convincing tiny colonies. The colonies should be imaged once they are fully formed and look like iPSCs.

Extended data Fig. 3d – Number of colonies just 2-3 per well? The experiment should be scaled up, otherwise this seems not significant.

Extended Data Fig. 7- The scRNA-seq data are not convincing: Sall4-iPSCs do not overlap with ESCs in Extended Data Fig. 7a; no transition towards pluripotency is observed.

Two D10 replicates do not overlap in Extended Data Fig. 7b, and there was just one D10 cell that clusters with Sall4-iPSC P5 that didn't even cluster with ESCs. How can the data showing reprogramming of one cell be convincing?

I would recommend the authors to do additional scRNA-seq experiments, and possibly focus the story more on Sal4+Oct4 reprogramming, which seems to work more efficiently.

REVIEWER COMMENTS

Reviewer #1 (Remarks to the Author):

The revised manuscript includes many new experiments and analyses, which improve the manuscript. However, the manuscript is hard to follow smoothly due to inadequate explanation and interpretation of the results and unclear rational connections in the text. In particular, the arrangement of the panels in Figures 2 and 3 does not align with the sequence in which the authors describe them in the text, leading to confusion for the readers. Therefore, the authors should reorganize the figures and text structures and clarify the logical flow before this manuscript can be considered for publication in Nature Communications.

Response: Thank you for your feedback on our manuscript. We appreciate your valuable evaluation and the opportunity to address your concerns. We apologize for the disorganized writing style, as well as the inadequate explanation and interpretation of the results and abrupt transitions. We understand the importance of clarity and coherence in scientific writing and have undertaken a thorough revision of the manuscript to address these concerns. For the issue of arrangement of the panels in Figures 2 and 3 does not align with the sequence describe in the text, we have carefully reorganized the sequence of Figures 2 and 3 and adjust the main text related to Figures 2 and 3 into 2 part:

1. The chromatin binding dynamics of SALL4 during SALL4-mediated reprogramming (as depicted in Figure 2 and Supplementary Fig.5).
2. SALL4 binds and regulates chromatin accessibility dynamics through direct and indirect effects to promote iPSCs induction (as illustrated in Figure 3 and Supplementary Fig.6-8).

Our aim is to ensure that the text flows smoothly and logically, with clear connections between ideas and minimal repetition. We thank you for your continued support and guidance. We have outlined our responses to each of your concerns below:

Specific Points:

1. In Figure 2b, the authors present the number of peaks in the SALL4 and DsRed systems. However, merely comparing the number of peaks between the two systems does not justify the conclusion that SALL4 promotes the transition of overall chromatin accessibility towards ESC states (Lines 203-206). Furthermore, according to Fig 2a, CO5 is classified as a peak set whose chromatin is closed in MEFs and becomes open in ESCs (CO5 appears to have few peaks in both the SALL4 and DsRed systems). However, Fig 2b shows that CO5 has the highest number of peaks for both SALL4 and DsRed systems. The authors should clarify this inconsistency.

Response: Thank you for your inquiry. We acknowledge that a mere comparison of peak numbers between the two systems is insufficient to substantiate the conclusion in our manuscript and we apologize for any misunderstanding caused by our inadequate interpretation. We have thoroughly revised the description for this section and updated these revisions in our revised manuscript accordingly.

To investigate the chromatin accessibility dynamics (CADs) during reprogramming, we defined CO as peaks closed in MEF and opened in ESC. The CO peaks were further segmented into distinct subgroups (CO1-CO5) based on the timing of transition. Based on this classification, the peaks in CO5 indicate that they are closed in MEFs, day 0, 4, 7 and 10 but open in ESC, representing unopened ESC related peaks at the end of reprogramming (which were highlighted in Fig.3a and Rebuttal Fig.1a in this rebuttal letter) and demonstrated the highest number of peaks among the SALL4 and DsRed systems (Fig.3b). Our data revealed a relatively lower number of CO5 peaks in the SALL4 system compared to DsRed, suggesting that SALL4 induced more peaks opening during reprogramming compared to control.

To provide clearer results, as suggested by the reviewer, we have calculated the overall numbers of OC1-4 and CO1-4 peaks, which represent the successful transition of ESC-related CADs from MEFs during SALL4-reprogramming, as shown in Supplementary Fig.6a. We have also indicated the SALL4-peaks numbers of each subgroup in Fig.3a and modified the description of Lines 203-206 (Lines 274-277 in the revised manuscript) as follows:

“The higher number of OC1-4 and CO1-4 peaks and the lower number of OC5 and CO5

peaks in the SALL4 system compared to the DsRed system suggests that the addition of SALL4 increase the transition numbers of ESCs-CADs-related-peaks during reprogramming (Fig.3b and Supplementary Fig.6a).”

We appreciate your diligence in identifying this concern, and we apologize for any confusion it may have caused.

Fig.3a, CADs for SALL4 system and DsRed system. The classification of PO, OC and CO subgroups in both the SALL4 system and DsRed system shown in Fig.3a was based on the CADs of the SALL4 system. PO, permanently open. CO, close to open. OC, open to close. Take DsRed system as reference is shown.

Fig.3b, The histogram shows the Number of the peaks for CO, OC, and PO subgroups of SALL4 system and DsRed system. The peak numbers presented in Fig. 3b for the SALL4 system and DsRed system were based on their respective CADs. SALL4, SALL4 system. DsRed, DsRed system.

Rebuttal Fig.1a, CADs for DsRed system. PO, permanently open. CO, close to open. OC, open to close.

Supplementary Fig.6a, The histogram shows the overall number of the peaks for CO1-4 and OC1-4 subgroups of SALL4 system and DsRed system.

2. In Figures 3a-c and Extended Data Figure 9, the authors investigated the direct and indirect effects of SALL4 on chromatin regions. In their rebuttal letter, the authors address some interpretation of the results, which should also be discussed in the main text.

Response: Thank you for your suggestion. we have incorporated a more detailed description regarding Figures 3a-c and Extended Data Figure 9 (Supplementary Fig.7a-e in the revised manuscript) into the main text of our manuscript (Lines 304-312 in the revised manuscript).

3. In lines 429-439, the authors proposed the three patterns in which OCT4 and SALL4 cooperatively regulate chromatin in the OCT4 + SALL4 system, illustrating several representative genes. However, a more systematic analysis is required. The authors should provide the number of peaks that fit the three patterns. Additionally, they should comprehensively characterize the gene sets whose expression and chromatin accessibility display the three patterns using RNA-seq and ATAC-seq data.

Response: Thank you for your valuable suggestion. We have reanalyzed our ATAC-seq and RNA-seq data and revised our manuscript accordingly. In order to provide a more comprehensive analysis of these reprogramming systems, we conducted a statistical analysis of the peaks for each CAD pattern. The numbers of peaks for each pattern are shown in Supplementary Fig.13b.

Subsequently, to define the gene sets exhibiting the CADs patterns in terms of their expression, we analyzed the gene expression profile associated with CADs patterns using our RNA-seq data and presented the gene sets (Supplementary Fig.13c) and corresponding peaks sets (Supplementary Fig.13d) in the form of heatmaps .

We believe that these additional analysis provide a more comprehensive understanding of the regulatory mechanisms underlying the synergistic effects of SALL4 and OCT4 in cellular reprogramming (Lines 450-459 in the revised manuscript). Thank you again for your insightful feedback, which has helped us to enhance the depth and clarity of our manuscript

Supplementary Fig.13b, The histogram shows the number of the peaks for CADs patterns. O+S/O4-C-O, OCT4+SALL4/OCT4-Common-open; O+S/O4-C-C, OCT4+SALL4/OCT4-Common-close; O+S/S4-C-O, OCT4+SALL4/SALL4-Common-open; O+S/S4-C-C, OCT4+SALL4/SALL4-Common-close; O+S-S-O, OCT4+SALL4 Specific-open; O+S-S-C, OCT4+SALL4 Specific-close.

Supplementary Fig.13c, Heatmaps of differential expression genes related to CADs patterns for RNA-seq data from O+S system, SALL4 system and OCT4 system. O+S/O4-C-O, OCT4+SALL4/OCT4-Common-open; O+S/O4-C-C, OCT4+SALL4/OCT4-Common-close; O+S/S4-C-O, OCT4+SALL4/SALL4-Common-open; O+S/S4-C-C, OCT4+SALL4/SALL4-Common-close; O+S-S-O, OCT4+SALL4 Specific-open; O+S-S-C, OCT4+SALL4 Specific-close.

Supplementary Fig.13d, Heatmaps of CADs patterns corresponding to differential expression genes in Supplementary Fig.13c. The differential transition peaks for each systems are shown. O+S/O4-C-O, OCT4+SALL4/OCT4-Common-open; O+S/O4-C-C, OCT4+SALL4/OCT4-Common-close; O+S/S4-C-O, OCT4+SALL4/SALL4-Common-open; O+S/S4-C-C,

OCT4+SALL4/SALL4-Common-close; O+S-S-O, OCT4+SALL4 Specific-open; O+S-S-C,
OCT4+SALL4 Specific-close.

4. In Extended Figure 4e, “SALL4 Specific Up (C6)” should be “SALL4 Specific Down (C6)”.

Response: Thank you for pointing out. We have corrected this mistake in our revised manuscript (Supplementary Fig.4e in the revised manuscript).

Reviewer #2 (Remarks to the Author):

The authors have made significant efforts to address my concerns. However, I recommend that before publishing the paper, they repeat the FACS for Ext. Fig. 3a. The cells appear to be touching the axes, which can affect the population distribution. Specifically, the plot for day 10 does not meet the required quality standards for publication.

Response: Thank you for your insightful comments and positive evaluation of our manuscript. We appreciate your additional points for further improvement and assure you that we have addressed them diligently.

For the FACS, We have re-acquired our FACS data for Supplementary Fig.3a and confirmed that all cells are within the designated range without intersecting with the axes to minimize any potential influence on population distribution.

To acquire a quality standards for publication of day 10 FACS data, we have increased the collecting numbers of OCT4-GFP⁺ cell for the day 10 sample (Supplementary Fig.3b).

We have updated these data in our revised manuscript to provide a more clearly findings. Thank you for your valuable feedback, which has helped us improve the clarity and presentation of our results.

Supplementary Fig.3

Supplementary Fig.3a. Flow cytometry was used to analyze the proportion of THY1⁻/EPCAM⁺ subgroup in SALL4 system at day0, day4, day7 and day10, respectively.

Supplementary Fig.3b. FACS analysis of OCT4-GFP⁺/EPCAM⁺ cells at day10.

Reviewer #3 (Remarks to the Author):

The results part as well as Figure 1 should include the information of the virus type – retrovirus, at the beginning when the reprogramming method is introduced. My comment was not sufficiently addressed.

Response: Thank you for your feedback. We apologize for the unclear description regarding the reprogramming method in our initial manuscript. We have clarified this in the revised version as follows:

1. We have added the overexpression method-retrovirus infection-in Fig.1a and Fig.4a.
2. The main text related to Figure1 (Lines 99-100 in the revised manuscript) has been rectified as:

“and identified eight molecules that exhibited the capability to drive the reprogramming of MEFs into iPSCs by overexpressing SALL4 through retrovirus infection.”

3. All overexpression experiments mentioned in the main text have been modified to highlight the retrovirus infection method. (Lines 206-207, 242, 325-326, 343, 462-469 in the revised manuscript)
4. We have added a section on "Gene Overexpression" to our methods (Lines 704-711 in the revised manuscript).

Your feedback has been invaluable in improving the quality of our manuscript, and we are grateful for your attention to detail.

KSR medium should be called KSR-2iLIF media or just 2iL media, as it contains 2i inhibitors.

Response: Thanks for your suggestion. We have corrected this in our revised manuscript (Lines 108, 643 in the revised manuscript).

The iPSC colonies are still called iPS colonies, even though I've already commented on this before.

Response: We apologize for the omission and have thoroughly revised the manuscript to address these concerns. We have corrected all the incorrect words in the revised version (Lines 55, 116, 120, 151, 154, 647, 648, 1179, 1181, 1238, 1245, Supplementary Fig.4b, Supplementary Fig.9h, Supplementary Fig.10b in the revised manuscript) to ensure the clarity and accuracy in our descriptions. We appreciate your diligence in identifying this concern, and we apologize for any confusion it may have caused.

Extended Fig.2 i – no nuclear localization for Nanog, Sox2 and Oct4 is visible, the colonies are too small. Higher magnification needed. Could be unspecific staining. As far as I remember, similar issue had occurred in the first version of the manuscript, which caused criticism.

Response: Thank you for pointing out this. We apologize for the unclear signal caused by small colonies in our previous manuscript. In the revised manuscript, we have recaptured larger colonies and confirmed the nuclear localization of the signal. Furthermore, to eliminate the possibility of unspecific staining, we cultured the iPSCs colonies on feeder cells and conducted immunofluorescence. The results indicate that the DAPI-positive feeder cells do not exhibit staining with Nanog, Sox2, or Oct4 antibodies, providing further validation of the specificity of our experiment (Supplementary Fig.2i in the revised manuscript). We have updated these data in our revised manuscript to enhance the clarity of our results. We appreciate your valuable feedback, which has contributed to improving the clarity and presentation of our results.

i

SALL4- Δ N12-iPSCs

Supplementary Fig.2i, Immunofluorescence analysis of pluripotency markers in SALL4- Δ N12 -iPSCs. Scale bars, 100 μ m.

Extended data Fig. 3a - All samples should be shown using the same scale.

Response: Thank you for bringing this to our attention. We have re-collected our FACS data for Extended Data Fig.3a (Supplementary Fig.3a in the revised manuscript), and reanalyzed the data using a consistent analysis standard to ensure the quality of our results. These updated data have been included in our revised manuscript accordingly.

Supplementary Fig.3a, Flow cytometry was used to analyze the proportion of THY1-/EPCAM+ subgroup in SALL4 system at day0, day4, day7 and day10, respectively.

Extended data Fig. 3b – No Oct4-GFP+ cells are visible. More cells should be sorted so there's a clear population.

Response: Thank you for your feedback. To ensure a high-quality result for the day 10 FACS data, we have re-collected the data and increased the number of OCT4-GFP+ cells collected for the day 10 sample in order to clearly identify the Oct4-GFP+ cell populations. We have included these updated data in our revised manuscript to present our findings more clearly (Supplementary Fig.3b in the revised manuscript).

Supplementary Fig.3b, FACS analysis of OCT4-GFP+/EPCAM+ cells at day10.

Extended data Fig. 3c – Not convincing tiny colonies. The colonies should be imaged once they are fully formed and look like iPSCs.

Response: Thanks for your suggestion. In response, we have made efforts to perform more than 6 additional experiments, however, we observed that OCT4-GFP⁺ colonies derived from this induction experiment exhibit a smaller size compared to cells induced directly without dissociation and FACS sorting, and the OCT4-GFP⁺ cells gradually disappeared after prolonged induction in iCD4. We have tried to pick and expand the colonies in 2iL medium. The findings also indicate that these cells have limited proliferative capacity.

We speculate that the cell-cell communication between intermediates and other cell types may be necessary for the proliferation of OCT4-GFP⁺ cells, and a more suitable induction method could potentially facilitate the maturation of these intermediates.

To provide clearer results, we have included new images in the revised manuscript (see Supplementary Fig. 3c in revised manuscript), which offer a clearer depiction of OCT4-GFP⁺ cells induced by reprogramming intermediates. We have also incorporated the results regarding the limited proliferative capacity of OCT4-GFP⁺ cells into our main text to present our findings more clearly (Lines 163-166 in the revised manuscript).

Supplementary Fig.3c, Morphological diagram for the OCT4-GFP⁺ cells generation at day4 induced from the day7 THY1-/EPCAM⁺ subgroups in SALL4 system. Scale bars, 100μm.

Extended data Fig. 3d – Number of colonies just 2-3 per well? The experiment should be scaled up, otherwise this seems not significant.

Response: Thanks for your suggestion. We acknowledge that the conclusion drawn from the low induction efficiency may not be sufficiently persuasive. To achieve a higher induction efficiency for reprogramming intermediates, we performed the experiment with an increased number of seeded cells (increased from 1.5×10^5 to 4.5×10^5 cells per well) to avoid damage caused by trypsin dissociation and long-term FACS sorting, resulting in an increased number of colonies using this method. We performed additional 3 independent experiments and the number of colonies from each well are shown in Supplementary Fig. 3d. The results demonstrated that the THY1-EPCAM⁺ population exhibited a higher emergence of OCT4-GFP⁺ cells compared to other populations in iCD4 medium, consistent with previous studies and further confirming the reprogramming intermediates nature of THY1-EPCAM⁺ populations. We have updated these data in our revised manuscript to enhance the quality of our manuscript.

Extended Data Fig. 7- The scRNA-seq data are not convincing: Sall4-iPSCs do not overlap with ESCs in Extended Data Fig. 7a; no transition towards pluripotency is observed. Two D10 replicates do not overlap in Extended Data Fig. 7b, and there was just one D10 cell that clusters with Sall4-iPSC P5 that didn't even cluster with ESCs. How can the data showing reprogramming of one cell be convincing?

Response: Thank you for bringing this issue to our attention. We speculate that the differences observed in iPSCs and ESCs may be attributed to the use of distinct culture medium for these cell types. It is worth noting that the ESC-scRNA-seq data analyzed in this study were obtained from Lin Guo et.al., who cultured their ESC on feeder using 15%FBS plus 2i and lif, which differs from our methods. Furthermore, the sequencing approach for combining day10 cells and iPSCs into a single sample in our prior manuscript may also affect the cellular distribution in UMAP maps.

For the observed inconsistencies in two Day 10 samples, it is hypothesized that these variations may be attributed to the use of different batches of MEFs for reprogramming, and that potential batch effects have not been adequately addressed in our data.

In response to these issue, we have implemented an enhanced methodology to mitigate batch effects in subsequent analysis, and we have also incorporated new ESCs and SALL4-iPSCs data to ensure the reliability of our results. These cells were cultured in identical medium, and our findings demonstrate that the ESCs and SALL4-iPSCs exhibit similar characteristics with overlapping populations (Supplementary Fig.11a,b).

We acknowledge that the transition trajectory from MEFs to iPSCs was subtle. This may be attributed to the low efficiency of SALL4-mediated reprogramming, as indicated by our FACS data showing that only 0.03% of cells were OCT4-GFP+. The limited number of target cells may have impacted their capture during scRNA-seq, resulting in ambiguous UMAP clustering results. In order to elucidate a clear transition trajectory from MEFs to iPSCs, we have opted to employ an alternative analysis method to depict the cellular fate transition process. We performed monocle trajectory analysis on days 0, 4, 7, 10 and iPSCs to clarify the path of cellular differentiation during reprogramming. The results revealed the emergence of two distinct developmental branches during the reprogramming process, which were not readily discernible in UMAP plotting (Supplementary Fig.11f).

We have updated these data in our revised manuscript to provide a more clearly findings (Lines 393-427 in the revised manuscript). Thank you for your valuable suggestion, which has greatly contributed to improving the quality of our results.

I would recommend the authors to do additional scRNA-seq experiments, and possibly focus the story more on Sal4+Oct4 reprogramming, which seems to work more efficiently.

Response: Thanks for your valuable suggestion. We have also acknowledged the potential difficulties in obtaining definitive results from a system with low efficiency. To obtain a more comprehensive understanding of the molecular roadmap associated with SALL4 and SALL4+OCT4-mediated reprogramming, we performed additional single-cell RNA sequencing at various time points throughout the SALL4+OCT4 reprogramming process, specifically collecting samples on days 0, 4, 7, and 10.

We utilized UMAP plots to visualize cell fate transitions in both reprogramming systems, revealing significant changes from day 0 to day 4 (as depicted in Supplementary Fig.11a,b). The observation of a relatively lower number of *Nanog* positive cells in the SALL4 system compared to the O+S system at day 10 further substantiated the cooperative effect of SALL4 and OCT4 during reprogramming (Supplementary Fig.11c,d). Notably, iPSCs exhibited a closer clustering with ESCs than D10-*Nanog* positive cells in both systems (as demonstrated in Supplementary Fig. 11a-d), suggesting that while most *Nanog* positive cells emerged at day10 may not fully mature into iPSCs, these cells can achieve maturation when cultured with ESC maintenance medium. Furthermore, we observed the upregulation or downregulation of selected reprogramming-promoting and barrier genes in distinct cell subpopulations during reprogramming, consistent with our previous analysis. (referenced in Supplementary Fig. 11c,d).

To further elucidate the trajectory of cellular differentiation during reprogramming, we performed monocle trajectory analysis on days 0, 4, 7, 10 and iPSCs in both systems. The results revealed the emergence of two distinct developmental branches during the reprogramming process, which were not readily discernible in UMAP plotting (Supplementary Fig.11e,f). We characterized one branch as likely to achieve pluripotency potential (pluripotency branch), based on its alignment with iPSCs-reprogramming directions. Importantly, cells within the pluripotency branch in the O+S system exhibited a more uniform distribution compared to those in the SALL4 system, suggesting that SALL4 and OCT4 collaboratively induce a state of cellular plasticity conducive to acquiring pluripotency more efficiently than SALL4 alone (Supplementary Fig.11e,f).

Additionally, in order to distinguish differences in reprogramming intermediates across various systems, we conducted a comparative analysis of the differential gene expression for THY1-/EPCAM+ cells within the SALL4, O+S and OKS systems. This analysis revealed variations in transcriptional regulations across different reprogramming processes, as indicated by both gene quantity and the functions annotated through GO analysis of differential expression genes (detailed in Supplementary Fig.12a-g).

We appreciate your constructive suggestion, which has helped us to improve the depth and clarity of our manuscript. We believe that these additional analyses provide a more comprehensive understanding of the regulatory mechanisms underlying reprogramming (Lines 393-427 in the revised manuscript). Thank you again for your valuable input.

Supplementary Fig.11

Supplementary Fig.11 Single-cell RNA sequencing of SALL4 or OCT4+SALL4-driven reprogramming process

a-b. UMAP plot for Single-cell RNA-seq data from SALL4-mediated reprogramming(a) and OCT4+SALL4 reprogramming process(b). Each dot represents one cell. The sampling time points

are shown with color code.

c-d. The expression of representative markers in SALL4-mediated reprogramming(c) and OCT4+SALL4 reprogramming process(d).

e-f. Monocle trajectories of reprogramming samples mediated by OCT4+SALL4 (e) and SALL4 alone (f) are depicted with color-coding representing reprogramming timepoints (left), pseudotime (middle), and Nanog expression levels (right). Each data point represents an individual cell, with the ordering of cells inferred based on the expression patterns of the most variable genes across reprogramming samples.

Supplementary Fig.12

Supplementary Fig.12 Diverse Characteristics of Thy1-/Epcam+ Intermediates in distinct Reprogramming processes

a-f. Heatmaps of differential expression gene analysis for scRNA-seq data from SALL4 system(a), OCT4+SALL4 system(c) and OKS systems(e). The 6 subgroups(C1-C6) of differential expression genes were based on gene expression between MEFs and THY1-/EPCAM+ cells. GO analysis for each subgroup are shown(b, d and f).

g. Venn diagrams shows the number of differential expression genes of THY1-/EPCAM+ cells between SALL4 system, OCT4+SALL4 system and OKS system.

REVIEWERS' COMMENTS

Reviewer #1 (Remarks to the Author):

The revised manuscript has been reorganized, making it easier to read. The newly added analyses are more systematic and contribute to a deeper understanding of the molecular mechanisms of cell reprogramming by SALL4 and OCT4. However, as I have pointed out since the initial review stage, there still needs to be more scientific rationale behind the selection criteria for the genes (related to Figure 5). It is necessary to either provide specific criteria or explain the selection based on the functional aspects of the genes. Additional analyses or explanations must be provided before this manuscript can be considered for publication in Nature Communications.

Major point

In the new Supplementary Figure 13 b-d, the authors proposed the three patterns in which OCT4 and SALL4 cooperatively regulate chromatin in the OCT4 + SALL4 system, using CUT & Tag, RNA-seq, and ATAC-seq data. The authors should integrate these datasets to identify the genes corresponding to these three patterns. Subsequently, the authors should focus on the specific genes to examine their effects on reprogramming.

Other points

1. In line 311, CO1 would be correct rather than CO2.
2. In lines 446-448, the authors described that “transcription factors such as TCF7, LHX2, and NKX6.1, may also have a synergistic effect on reprogramming in the O+S system for their motif enrichment pattern and gene expression level in our analysis (Fig.5b and Supplementary Fig.13a)”. However, the synergistic effect is not evident from the figures. The correct interpretation of the figure should be provided.

Reviewer #2 (Remarks to the Author):

The authors have adequately addressed my comments.

Reviewer #3 (Remarks to the Author):

My concerns have been fully addressed. I recommend the manuscript for publication in Nature Communications. Congratulations on your findings!

Point-by-point response to the reviewers' comments

REVIEWER COMMENTS

Our responses are in blue.

Reviewer #1 (Remarks to the Author):

The revised manuscript has been reorganized, making it easier to read. The newly added analyzes are more systematic and contribute to a deeper understanding of the molecular mechanisms of cell reprogramming by SALL4 and OCT4. However, as I have pointed out since the initial review stage, there still needs to be more scientific rationale behind the selection criteria for the genes (related to Figure 5). It is necessary to either provide specific criteria or explain the selection based on the functional aspects of the genes. Additional analyzes or explanations must be provided before this manuscript can be considered for publication in Nature Communications.

Response: We appreciate your insightful suggestion regarding our manuscript and recognize the importance of providing a clearer scientific rationale for gene selection criteria. As a result, we have reorganized the description and analysis of these results in our revised manuscript (please refer to the response to the **Major point**). Briefly, candidate genes within our proposed patterns were identified based on expression levels, peak enrichment, and functional roles. This evidence and rationale for gene selection has been integrated into the revised manuscript. Addressing these concerns will undoubtedly strengthen the manuscript and enhance its logical and scientific value. Thank you once again for your invaluable feedback.

Major point

In the new Supplementary Figure 13 bd, the authors proposed the three patterns in which OCT4 and SALL4 cooperatively regulate chromatin in the OCT4 + SALL4 system, using CUT & Tag, RNA-seq, and ATAC-seq data. The authors should integrate these datasets to identify the genes corresponding to these three patterns. Subsequently, the authors should focus on the specific genes to examine their effects on reprogramming.

Response: Thank you for your insightful suggestion. We have restructured our description and analysis of the cooperative regulation of OCT4 and SALL4 in the OCT4 + SALL4 system to offer a more coherent and clearer analysis of these results. Our adjustments have been incorporated into the revised version as follows (Lines 450-480 in the revised manuscript):

“ Combining the analyses of the SALL4, OCT4, and O+S systems, we proposed six patterns in which OCT4 and SALL4 cooperatively regulate CADs in O+S system and defined the peak numbers of each pattern (Supplementary Fig.13a). These patterns are reflected in chromatin accessibility as follows: (1) Common open in O+S system and SALL4 system (O+S/S4-C-O), (2) Common close in O+S system and SALL4 system (O+S/S4-C-C), (3) Common open in O+S system and OCT4 system (O+S/O4-C-O), (4) Common close in O+S system and OCT4 system (O+S/O4-C-C), (5) only open in O+S system (O+S-S-O), and (6) only close in O+S system (O+S-S-C) (Supplementary Fig.13a). We further categorized the genes associated with these CADs patterns based on transcription levels using RNA-seq data and revealed numerous changes in gene expression that fit these types of synergistic modes (Fig.5c,d and Supplementary Fig.13b-e). We hypothesize that OCT4 and SALL4 improve iPSCs induction efficiency through their regulation of genes within these patterns, where patterns 1, 3, and 5 likely contain genes that promote reprogramming in the O+S system, while patterns 2, 4, and 6 may contain genes that hinder reprogramming. To support this hypothesis, we selected genes based on expression levels, peak enrichment, and functional relevance. Representative potential reprogramming-promoting genes that have been identified include *Esrrb*, *Tfap2c*, *Rsk1*, and *Sox2* for their roles in facilitating the induction of iPSCs in the SALL4 and OKS systems. Conversely, genes such as *Mndal*, *Mogat2*, and *Sbsn* were identified as potential barriers to reprogramming due to their somatic-related functions and high levels of expression and peak enrichment (Fig.5c,d and Supplementary Fig.13b-e).

We next explore and compare the reprogramming abilities of these representative genes during iPSCs induction. Overexpression of *Esrrb* and *Rsk1* through retroviral infection significantly promotes the generation efficiency of

OCT4-GFP⁺ colonies in the OCT4, SALL4, or O+S systems, Conversely, *Mogat2* and *Sbsn* impair the iPSCs induction for these three systems, while *Mndal* exerts an inhibitory effect in the O+S systems (Fig.5e and Supplementary Fig.13f). Overexpression of *Tfap2c* or *Sox2* by retroviral infection promotes the generation efficiency of OCT4-GFP⁺ colonies in the OCT4 or O+S systems but has an inhibitory effect on the induction efficiency of the SALL4 system (Fig.5e). ”

We appreciate your detailed and insightful suggestion. Your feedback has been instrumental in improving the clarity and scientific rigor of our manuscript.

Supplementary Fig.13f

Other points

1. In line 311, CO1 would be correct rather than CO2.

Response: Thank you for bringing this to our attention. We have corrected "CO2" to "CO1" in the revised manuscript (Line 314).

2. In lines 446-448 , the authors described that “transcription factors such as TCF7, LHX2, and NKX6.1, may also have a synergistic effect on reprogramming in the O+S system for their motif enrichment pattern and gene expression level in our analysis (Fig.5b and Supplementary Fig.13a)". However, the synergistic effect is not evident

from the figures. The correct interpretation of the figure should be provided.

Response: Thank you for your valuable suggestion. We apologize for any confusion caused by our previous description, and we have revised the manuscript to clarify these points.

Upon further review, we acknowledge that our experiments did not provide sufficient support for a synergistic effect of TCF7, LHX2, and NKX6.1 in the O+S system, nor do these genes align with the proposed patterns in our manuscript.

Considering the unexpected reprogramming function and inappropriate patterns exhibited by *Tcf7*, *Lhx2*, and *Nkx6.1*, We have substituted *Nkx6.1* with the gene *Mndal* (Supplementary Fig.13f), which is well-suited for pattern 2 and also demonstrates inhibitory effects in the O+S system, thereby providing support for our hypothesis. Additionally, we have incorporated the revised results pertaining to *Tcf7* and *Lhx2* into the Discussion section, accompanied by a discussion of these unexpected findings as follows (Lines 617-623 in the revised manuscript):

“Additionally, transcription factors such as TCF7 and LHX2, may also exert regulatory influence in the O+S system for their motif enrichment pattern and gene expression level in our analysis (Fig.5b and Supplementary Fig.13g). However, overexpression of these genes causes an inhibitory effect in the O+S system (Supplementary Fig.13f,g). These findings imply the existence of additional reprogramming-promoting and inhibiting genes that may not be appropriately regulated by OCT4 and SALL4, but could potentially be controlled by other transcription factors.”

By incorporating these modifications into our main text, we aim to enhance the comprehensiveness of understanding regarding the cooperative regulation of OCT4 and SALL4 in the OCT4 + SALL4 system. We are grateful for your valuable feedback, which has contributed to improving the clarity and presentation of our results.

Reviewer #2 (Remarks to the Author):

The authors have adequately addressed my comments.

Response: Thank you very much for your positive feedback and for your careful review. We are glad that our revisions have addressed your comments satisfactorily.

Reviewer #3 (Remarks to the Author):

My concerns have been fully addressed. I recommend the manuscript for publication in Nature Communications. Congratulations on your findings!

Response: Thank you for your encouraging feedback and for recommending our manuscript for publication. We appreciate your thoughtful review and support.